# Safe Reinforcement Learning with ADRC Lagrangian Method

## Abstract

Safe reinforcement learning (Safe RL) seeks to maximize rewards while satisfying safety constraints, typically addressed through Lagrangian-based methods. However, existing approaches, including PID and classical Lagrangian methods, suffer from oscillations and frequent safety violations due to parameter sensitivity and inherent phase lag. To address these limitations, we propose ADRC-Lagrangian methods that leverage Active Disturbance Rejection Control (ADRC) for enhanced robustness and reduced oscillations. Our unified framework encompasses classical and PID Lagrangian methods as special cases while significantly improving safety performance. Extensive experiments demonstrate that our approach reduces safety violations by up to 74%, constraint violation magnitudes by 89%, and average costs by 67%, establishing superior effectiveness for Safe RL in complex environments.

## 1 Introduction

Reinforcement Learning (RL) aims to maximize rewards as agents interact with environments, finding applications in fields such as robotics (Sun et al., 2023; Luo et al., 2024; Li et al., 2024) and the post-training of Large Language Models (LLMs) (Bai et al., 2022; Lee et al., 2023; Rafailov et al., 2023). However, in real-world scenarios like autonomous driving (Muhammad et al., 2020), safety requirements are often of paramount importance. It is essential not only to maximize rewards but also to ensure compliance with safety constraints. To address this challenge, Safe RL (García & Fernández, 2015) has emerged as a paradigm dedicated to reliable and robust policy learning in complex and dynamic environments. Safe RL is typically formulated as a Constrained Markov Decision Process (CMDP). Among the various approaches for solving CMDPs, Lagrangian methods play a pivotal role by transforming constrained optimization problems into unconstrained ones through the introduction of a dual variable, the Lagrange multiplier. This transformation enables the adaptation of any RL algorithm into a Safe RL framework (Achiam et al., 2017; Chow et al., 2018b; 2019), leading to the development of numerous novel Safe RL algorithms (Liu et al., 2024; Chen et al., 2024).

Classical Lagrangian updates can be interpreted as pure integral controllers on the constraint violation signal. While simple, this mechanism reacts slowly to the rapid distributional shifts caused by policy updates and the stochasticity of cost estimates, leading to lag, overshoot, and persistent oscillations in safety performance. Attempts to mitigate these issues with PID-based extensions (Stooke et al., 2020) reduce oscillations by adding proportional and derivative terms, but their behavior remains fragile: performance is highly sensitive to the chosen gains and rarely transfers robustly across tasks (Panda, 2012; Åström & Hägglund, 1995). These limitations point to a deeper challenge—existing methods lack a way to explicitly counteract the drifting disturbances that underlie oscillatory training.

To overcome this challenge, we turn to the broader toolbox of *adaptive control*, whose central goal is to maintain stability and performance under unknown and time-varying dynamics. Among the many adaptive strategies, Active Disturbance Rejection Control (ADRC) (Han, 1998; Zhong et al., 2020a) is particularly well suited to the Safe RL setting. Unlike classical adaptive methods that often require a parametric model or extensive gain tuning, ADRC treats all uncertainty—including model error, noise, and nonstationarity—as a lumped disturbance, and employs a lightweight observer to estimate and cancel it online. This observer-based design makes ADRC both model-free and robust to changing dynamics, while its reliance only on observable signals (such as cost returns) makes it a natural match for reinforcement learning. In contrast, alternatives such as MRAC or Lyapunov-based

adaptive schemes typically assume access to richer state information or stronger structural knowledge, which is impractical in high-dimensional RL environments.

Building on this idea, we introduce ADRC Lagrangian methods, which augment the classical dual update with an observer that estimates and cancels the disturbance acting on the constraint return. By combining this observer with a smooth reference trajectory for the cost threshold, our update suppresses transient overshoot while directly compensating for nonstationarity and noise. The resulting method is simple to implement, model-free, and optimizer-agnostic, yet it fundamentally changes the dynamics of constraint regulation: we derive a theoretical lower bound on the observer gain that provides safe defaults across diverse environments, we show that classical and PID Lagrangian updates emerge as strict special cases of our formulation, and we establish through frequency-domain analysis that our approach achieves smaller disturbance-estimation error and reduced phase lag. Empirically, these properties translate into substantial improvements in safety throughout training: on OmniSafe benchmarks our method reduces violation rates by up to 74%, lowers violation magnitudes by 89%, and decreases average costs by 67%, all while maintaining competitive reward performance. These results demonstrate that bringing the ADRC perspective into Safe RL yields both principled theoretical guarantees and significant empirical gains.

In conclusion, our main contributions are:

- We are the first to introduce ADRC into Safe RL, dynamically adjusting the Lagrange multiplier to improve constraint satisfaction and training stability.

- We theoretically establish that both PID and classical Lagrangian methods are special cases of our ADRC Lagrangian methods. Moreover, through frequency-domain analysis, we demonstrate that our method significantly reduces phase lag compared to traditional approaches, leading to faster and more stable constraint satisfaction.

- Comprehensive experiments validate the effectiveness of our method, showing significant improvements in reducing oscillations during training across diverse benchmarks.

## 2 RELATED WORK

**Safe RL** Safe reinforcement learning (RL) aims to find optimal policies that maximize rewards while satisfying safety constraints (Garcıa & Fernández, 2015; Achiam et al., 2017; Wachi & Sui, 2020; Yang et al., 2020). Common approaches include safe exploration techniques to ensure safety during training (Sui et al., 2015; Wang et al., 2023) and the primal-dual framework, which employs Lagrangian multipliers to address constrained optimization (Ray et al., 2019; Ding et al., 2020; Chow et al., 2018b; 2019). Recent advances have improved the tuning of these multipliers through gradient-based methods (Lillicrap et al., 2019; Tessler et al., 2018; Zhang et al., 2020), PID-based updates (Stooke et al., 2020), adaptive primal-dual methods (Chen et al., 2024), and variational inference approaches (Liu et al., 2022; Huang et al., 2022), enhancing algorithm stability and performance (Yao et al., 2024). On-policy algorithms can be broadly categorized into Lagrangian methods, such as PDO (Chow et al., 2018b), and convex optimization methods like CPO (Achiam et al., 2017) and CVPO (Liu et al., 2022). Recent developments, including APPO (Dai et al., 2023) and CUP (Yang et al., 2022), specifically address oscillatory behaviors and improve constraint feasibility. Furthermore, accurate estimation of objective and constraint functions is crucial, as it significantly influences the efficiency and reliability of policy updates (Altman, 2021). Additionally, methods targeting training stability, such as policy inertia learning (Chen et al., 2021) and soft-switching gradient manipulation (Gu et al., 2024), have effectively reduced oscillations, highlighting their importance for Safe RL.

**Control Theory** Control theory includes traditional controllers such as Proportional-Integral-Derivative (PID) controllers (Åström & Hägglund, 2006; Ang et al., 2005), which, despite their widespread use, are sensitive to parameter variations and external disturbances (Panda, 2012; Åström & Hägglund, 1995). To overcome these limitations, Model Reference Adaptive Control (MRAC), which is a milestone of adaptive control, has been developed, enabling dynamic parameter adjustment to maintain performance under system uncertainties (Nguyen & Nguyen, 2018; Parks, 1966). By incorporating the MIT rule (Mareels et al., 1987), MRAC-PID controllers leverage backpropagation-inspired methods (Rumelhart et al., 1986) to adapt PID parameters in real time, addressing control gain sensitivity (Singh & Kumar, 2015; Kungwalrut et al., 2011). Furthermore, Zhang & Guo (2019)

it has been demonstrated that PID parameters could be selected within a specific manifold to ensure global system stabilization with exponential error convergence. In parallel, Active Disturbance Rejection Control (ADRC) has emerged as a robust alternative for managing uncertainties and disturbances. First introduced by Han (1998), ADRC has been further developed to enhance its applicability and theoretical underpinnings (Han, 2009). Recent advancements include applications in nonlinear systems (Guo & Zhao, 2017) and rigorous stability analysis using Lyapunov functions (Zhong et al., 2020b), solidifying ADRC as an effective and versatile control strategy.

## 3 Preliminaries

**Safe Reinforcement Learning**  A Markov Decision Process (MDP) $\mathcal{M}$ (Puterman, 2014) is defined by the tuple $(\mathcal{S}, \mathcal{A}, R, \mathbb{P}, \mu, \gamma)$, where $\mathcal{S}$ and $\mathcal{A}$ denote state and action spaces, $R$ is the reward function, $\mathbb{P}(s' \mid s, a)$ is the state transition probability, $\mu$ is the initial state distribution, and $\gamma \in (0, 1)$ is the discount factor. A parameterized stationary policy $\pi_\theta(a \mid s)$ specifies action probabilities given state $s$. The goal of reinforcement learning (RL) is to maximize the expected return:

$$J(\pi_\theta) = \mathbb{E}_{s \sim \mu} \left[ \sum_{t=0}^{\infty} \gamma^t r_{t+1} \right]. \tag{1}$$

Safe RL is typically formulated as a constrained MDP (CMDP) (Altman, 1999), which extends MDPs with constraints defined by cost functions $c_i : \mathcal{S} \times \mathcal{A} \to \mathbb{R}$ and thresholds $d_i$. The cost return under policy $\pi_\theta$ is:

$$J_{c_i}(\pi_\theta) = \mathbb{E}\pi_\theta \left[ \sum_{t=0}^{\infty} \gamma^t c_i(s_t, a_t) \right]. \tag{2}$$

Safe RL aims to find an optimal policy:

$$\pi^* = \arg \max_{\pi_\theta} J(\pi_\theta) \quad \text{s.t.} \quad J_{c_i}(\pi_\theta) \leq d_i, \forall i. \tag{3}$$

**Lagrangian Methods**  In constrained optimization problems such as those in Safe RL, the goal is to maximize the objective function while satisfying constraints. A common approach is to apply the Lagrangian method. Specifically, for a CMDP with a single cost constraint, denote $d$ as the cost threshold, and $\lambda \geq 0$ as the Lagrangian multiplier, we define the Lagrangian function as:

$$\mathcal{L}(\theta, \lambda) = J(\pi_\theta) - \lambda(J_c(\pi_\theta) - d), \tag{4}$$

The optimal solution aims to maximize the Lagrangian with respect to $\theta$ while minimizing it with respect to the multiplier $\lambda$. To achieve this, we apply a gradient-based approach to update $\lambda$ iteratively. Specifically, we define the constraint violation as $g(\pi_\theta) := J_c(\pi_\theta) - d$. Denote $\alpha > 0$ as the learning rate controlling the update step size; we perform gradient ascent on $\lambda$ with respect to $\mathcal{L}$:

$$\dot{\lambda} = \alpha g(\pi_\theta), \tag{5}$$

Thus, optimizing $\lambda$ reduces to a standard gradient ascent problem:

$$\dot{\lambda} = \alpha \frac{\partial \mathcal{L}(\theta, \lambda)}{\partial \lambda} = \alpha g(\pi_\theta), \tag{6}$$

Discretizing over time, the Lagrangian multiplier is updated iteratively by:

$$\lambda_t = \lambda_{t-1} + \alpha g(\pi_{\theta_{t-1}}), \tag{7}$$

or equivalently, by summing constraint violations over time:

$$\lambda_t = \lambda_0 + \alpha \sum_{\tau=0}^{t-1} g(\pi_{\theta_\tau}) \approx \alpha \int_0^t g(\pi_{\theta_\tau}) d\tau. \tag{8}$$

This shows that classical Lagrangian methods implement a pure Integral (I) controller on the constraint violation signal $g(\pi_\theta)$. With this view, to reduce oscillations during training, PID Lagrangian methods (Stooke et al., 2020) generalize the integral control by adding proportional (P) and derivative (D) terms into the dynamics of $\lambda$:

$$\dot{\lambda} = \alpha g(\pi_\theta) + \beta \dot{g}(\pi_\theta) + \gamma \ddot{g}(\pi_\theta), \tag{9}$$

where $\alpha$, $\beta$, and $\gamma$ are positive coefficients controlling the strength of the I, P, and D terms respectively.

Similarly, integrating this equation over time, the resulting PID update law for the multiplier is:

$$\lambda_t = \left( K_p g(\pi_{\theta_t}) + K_i \int_0^t g(\pi_{\theta_\tau}) d\tau + K_d \dot{g}(\pi_{\theta_t}) \right)_+ , \tag{10}$$

where $K_p$, $K_i$, $K_d$ are proportional, integral, and derivative gains that need to be tuned carefully.

**Active Disturbance Rejection Control** Compared with PID controller, Active Disturbance Rejection Control (ADRC) (Han, 1998; Zhong et al., 2020a) provides a more adaptive and resilient alternative. Unlike traditional PID control, ADRC explicitly estimates and compensates for unknown disturbances through an observer-based framework, reducing reliance on precise model knowledge and hyperparameter sensitivity. The core component of ADRC is the Extended State Observer (ESO), which is designed to simultaneously estimate both the internal system states and the total disturbance affecting the system dynamics. By accurately reconstructing the disturbance in real time, the control input can proactively reject its influence, significantly enhancing system stability and performance. In practical Safe RL scenarios, where exact system dynamics are unknown and only observable quantities like costs are available, a reduced-order ESO design is commonly employed. In addition to disturbance estimation, to achieve smoother system behavior and better transient performance, the control strategy can also incorporate a designed reference trajectory that guides the evolution of the system states toward the desired setpoints.

## 4 METHOD

### 4.1 CLOSED-LOOP SYSTEM REPRESENTATION OF SAFE RL

Lagrangian-based Safe RL can be viewed as a feedback system: the policy affects the cumulative cost, which drives the Lagrange multiplier that in turn influences the policy update. We capture this interaction in a simple closed-loop form,

$$\begin{cases} x_1 = J_c, \\ \dot{x}_1 = x_2, \\ \dot{x}_2 = f(x_1, x_2, t) + u(t), \\ u(t) = \lambda_t, \end{cases} \tag{11}$$

where $x_1$ is the cumulative cost, $x_2$ its derivative, $f(\cdot)$ aggregates all unknown and time-varying effects, and $u(t)$ is the control input given by the multiplier $\lambda_t$. This formulation highlights the root cause of oscillations in existing methods: the dynamics drift as the policy changes, while the multiplier behaves like an integral controller that lags behind disturbances.

Our goal is to replace this fragile mechanism with an observer-augmented update inspired by ADRC. By explicitly estimating the total disturbance from cost signals and compensating it in real time, while guiding constraint satisfaction through a smooth reference trajectory, our method achieves faster and more stable regulation than classical or PID-based approaches. The designs of the observer and reference signal are presented next.

### 4.2 ARRANGING A TRANSIENT PROCESS

In Safe RL, the dual update implicitly drives the cumulative cost $x_1$ toward the safety threshold d. To formalize this objective, we introduce a reference signal $y^*(t)$, which represents the target trajectory that $x_1$ should ideally follow. Since the ultimate goal is constraint satisfaction, the natural choice of reference is a constant at the threshold:

$$y^*(t) = d. \tag{12}$$

However, tracking this signal directly can be problematic in practice. At the beginning of training, policies are usually far from safe, so the gap $x_1(0) - d$ is large. Forcing the multiplier to eliminate this gap immediately leads to abrupt updates, which amplify estimator noise and policy nonstationarity into overshoot and repeated violations. Empirically, this appears as sharp cost spikes in early training and oscillatory swings of $\lambda_t$, even when the constraint is ultimately feasible.

To prevent these instabilities, we need to arrange a transient process that gradually shrinks the effective budget from the current cost level toward $d$ with critically damped dynamics. In the Safe RL context, this corresponds to a smooth budget schedule: early training permits a controlled violation margin that decays over time, enabling exploration while guiding the system toward feasibility.

Concretely, we filter $y^*(t)$ through a second-order system:

$$\ddot{r} = -2c_r \dot{r} - c_r^2(r - d), \qquad r(0) = x_1(0), \ \dot{r}(0) = x_2(0), \tag{13}$$

where $c_r > 0$ controls the tightening speed. The resulting reference trajectory is

$$r(t) = d + \big(x_1(0) - d\big)e^{-c_r t} + \big(x_2(0) + c_r(x_1(0) - d)\big)te^{-c_r t}, \tag{14}$$

which starts from the current cost level and slope, then converges smoothly and non-oscillatorily to $d$. This shaped reference avoids abrupt enforcement in early training, stabilizes the multiplier dynamics, and reduces phase lag in constraint regulation.

A detailed derivation is provided in Appendix A.

### 4.3 EXTENDED STATE OBSERVATION FOR MULTIPLIER UPDATES

Training in Safe RL is inherently noisy and nonstationary: the measured cost fluctuates due to stochastic transitions, estimation error, and abrupt policy changes. If the multiplier reacts to these raw signals directly, it amplifies noise and tends to oscillate. What is missing is an online estimate of the *unmodeled dynamics*—the effective disturbance $f(x_1, x_2, t)$ in Eqn. 11—so that the update can distinguish genuine constraint trends from transient fluctuations.

To this end, we borrow the idea of an *extended state observer* (ESO) from adaptive control, but use it in the simplest reduced-order form Zhong et al. (2020a) suitable for RL. The ESO maintains an auxiliary state $\xi$ that is updated alongside observed costs:

$$\begin{cases} \dot{\xi} = -\omega_o \xi - \omega_o^2 x_2 - \omega_o u, \\ \hat{f} = \xi + \omega_o x_2, \end{cases} \tag{15}$$

where $\hat{f}$ serves as a running estimate of the disturbance and $\omega_o > 0$ is a gain controlling how aggressively it adapts. Intuitively, $\hat{f}$ behaves like a bias-correction term that smooths the effect of noise and policy shifts before they reach the multiplier.

With this estimate, the control input $u(t)$ (i.e., the multiplier update) is designed to track the transient reference $r(t)$ using proportional, derivative, and disturbance feedback:

$$u(t) = k_{ap}(x_1 - r) + k_{ad}(x_2 - \dot{r}) + \hat{f} - \ddot{r}. \tag{16}$$

Substituting (15) into (16), and identifying $u(t)$ with $\lambda_t$, we obtain the ADRC-based update law:

$$\lambda_t = (k_{ap} + \omega_o k_{ad})(x_1 - r) + (k_{ad} + \omega_o)(x_2 - \dot{r}) + \omega_o k_{ap} \int_0^t (x_1(\tau) - r(\tau))d\tau - \ddot{r}. \tag{17}$$

**Proposition 4.1.** *Classical PID Lagrangian methods are a special case of* (17). *Under a specific mapping between $(K_p, K_d, K_i)$ and $(k_{ap}, k_{ad}, \omega_o)$, the ADRC update reduces exactly to the PID rule in Eqn. 10.*

Thus ADRC can be seen as a strict generalization of PID Lagrangian: in addition to $P$, $I$, and $D$ terms, it continuously estimates the disturbance and compensates for it in real time. This additional degree of adaptivity reduces phase lag and improves robustness to noise and nonstationarity, as analyzed in Sec. 4.4.

### 4.4 THE LOWER BOUND OF OPTIMAL PARAMETERS

A central challenge in Safe RL is parameter sensitivity: the same multiplier update rule can behave well in one environment but oscillate or diverge in another. This is particularly acute for PID-based Lagrangian methods, whose stability depends heavily on hand-tuned gains. To make ADRC practical in Safe RL, we aim for a principled condition that guarantees stability and bounded estimation error across environments, thereby removing the need for brittle manual tuning.

We characterize the uncertainty in the closed-loop dynamics of Eqn. 11 by bounding how disturbances can depend on the current state and vary over time. Following Zhong et al. (2020a), we consider

$$f(x_1, x_2, t) = h(x_1, x_2) + w(t), \qquad \left|\frac{\partial h}{\partial x_1}\right| \le L_1, \left|\frac{\partial h}{\partial x_2}\right| \le L_2, |w(t)|, |\dot{w}(t)| \le L_3, \qquad (18)$$

where $L_1$ and $L_2$ bound the sensitivity of disturbances to the cost $x_1$ and its rate $x_2$, while $L_3$ bounds the magnitude and variation of exogenous fluctuations such as noise or nonstationarity. This setting captures both state-dependent effects and purely time-dependent variations.

Within this class, the admissible observer gains $\omega_o$ are constrained by a characteristic polynomial manifold

$$\Omega = \left\{\omega \in \mathbb{R} \,\middle|\, n_0\omega^4 + n_1\omega^3 + n_2\omega^2 + n_3\omega + n_4 = 0\right\}, \qquad (19)$$

where the coefficients $n_i$ depend on $(k_{ap}, k_{ad})$ and the constants $(L_1, L_2, L_3)$. We define

$$\bar{\omega}_o = \begin{cases} \max\{\omega \mid \omega \in \Omega\}, & \text{if } \Omega \ne \emptyset, \\ 0, & \text{otherwise.} \end{cases}$$

The final lower bound ensuring stability and bounded estimation error is therefore

$$\omega_o^* = \max\left\{\bar{\omega}_o,\ 0,\ \frac{L_1 - k_{ap}}{k_{ad}},\ L_2 - k_{ad}\right\}. \qquad (20)$$

Intuitively, this bound ensures that the ESO adapts quickly enough to track disturbances while remaining stable. In practice, this means practitioners only need to set $\omega_o > \omega_o^*$, removing the trial-and-error search that plagues PID tuning.

Beyond stability, we also analyze how fast and accurately ADRC reacts compared to classical integral updates. Let $e(t) = x_1(t) - r(t)$ be the tracking error, $\hat{f}_I = k_i \int_0^t e(\tau)\, d\tau$ the disturbance estimate from integral control, and $\hat{f}$ the ADRC estimate. Their estimation errors are

$$e_{f_I}(t) = \hat{f}_I - f(x_1, x_2, t), \qquad e_f(t) = \hat{f} - f(x_1, x_2, t).$$

Denote the Laplace transforms of $e$, $e_f$, $e_{f_I}$, $\hat{f}_I$, and $\hat{f}$ by $E(s)$, $E_f(s)$, $E_{f_I}(s)$, $\hat{F}_I(s)$, and $\hat{F}(s)$ respectively, and let $F(s)$ be the transform of the disturbance $f$. Then $G_{e_f}(s)$ and $G_{e_{f_I}}(s)$ are the transfer functions from $F(s)$ to $E_f(s)$ and $E_{f_I}(s)$.

We establish the following result:

**Theorem 4.2.** *Suppose $\omega_o > \omega_o^*$. Then, for any frequency $\omega$, ADRC Lagrangian achieves uniformly lower disturbance estimation error than integral control:*

$$\frac{|G_{e_f}(i\omega)|}{|G_{e_{f_I}}(i\omega)|} < 1.$$

*Moreover, if $\omega_o > \max\left\{\frac{k_{ap} - \omega^2}{k_{ad}},\ \omega_o^*\right\}$, ADRC Lagrangian also exhibits smaller phase lag:*

$$\arg(G_{e_f}(i\omega)) < \arg(G_{e_{f_I}}(i\omega)).$$

In words, the frequency-domain guarantees imply concrete benefits in Safe RL training. A smaller disturbance estimation error means the multiplier update is less sensitive to transient noise and policy-induced fluctuations, avoiding spurious reactions to single-batch variability. A smaller phase lag means constraint violations are corrected earlier, rather than several updates later, reducing the amplitude and duration of overshoot. Together, these properties yield training dynamics with fewer repeated safety violations, smoother multiplier trajectories, and faster convergence to the feasible region. Detailed derivations are provided in Appendix C.1.

**Theoretical Guarantees in Safe RL.** While the frequency-domain analysis above explains the reduction in phase lag, we provide a comprehensive time-domain analysis in Appendix C.3. By explicitly linking the ESO bandwidth $\omega_o$ to Safe RL hyperparameters (e.g., trust-region radius $\delta$ and batch size $N$), we establish three fundamental guarantees: (i) **Robustness:** An ISS-type bound proving that the cost tracking error is confined to a tube scaling with $\mathcal{O}(1/\omega_o)$, despite policy non-stationarity and sampling noise; (ii) **Reward Improvement:** A high-probability guarantee of non-decreasing reward under trust-region updates using an ESO-based advantage; (iii) **Safety:** A bounded average constraint violation guarantee, ensuring that safety debt does not accumulate indefinitely.

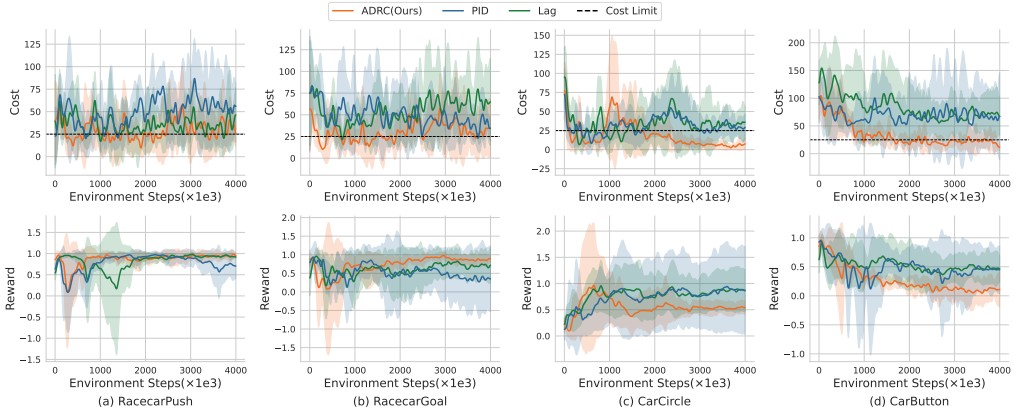

Figure 1: The training curves of PPO with various Lagrangian methods (denoted as CPPOLag, CPPOPID, CPPOADRC) across different tasks, showing episodic returns and costs over five random seeds. Solid lines represent mean values, while shaded areas denote variance. CPPOADRC demonstrates a shorter lag phase and lower costs compared to baselines, while achieving competitive rewards. Additional results are provided in Appendix F.1.

## 4.5 ADRC Lagrangian Methods in Safe RL

We now describe how ADRC Lagrangian methods are applied in Safe RL training. The central idea is to replace the hand-tuned, noise-sensitive dual update with an observer-augmented rule that adapts automatically to the evolving learning dynamics.

In practice, the Lagrangian multiplier $\lambda$ serves as a penalty knob: when the observed cumulative cost approaches the safety threshold, $\lambda$ should increase to discourage unsafe behavior; when costs fall well below the threshold, $\lambda$ can relax to allow more exploration. ADRC realizes this adaptivity by updating $\lambda$ according to Eqn. 17, where the observer $\hat{f}$ continuously estimates the effect of unmodeled disturbances and compensates for them in real time. This makes multiplier updates less myopic and more responsive than classical dual ascent or PID rules.

A key question is how to choose the observer gain $\omega_o$. As shown in Sec. 4.4, $\omega_o$ must exceed a lower bound $\omega_o^*$ that depends on environment sensitivities $(L_1, L_2)$. Since these quantities are unknown beforehand, we approximate them online using finite differences of observed costs:

$$
\begin{aligned}
L_1 &\approx \max_t \left| \frac{\ddot{x}_1(t+1) - \ddot{x}_1(t)}{x_1(t+1) - x_1(t)} \right|, \\
L_2 &\approx \max_t \left| \frac{\ddot{x}_1(t+1) - \ddot{x}_1(t)}{x_2(t+1) - x_2(t)} \right|,
\end{aligned}
\tag{21}
$$

where $x_1$ is the cumulative cost and $x_2$ its derivative. These estimates allow us to adaptively compute $\omega_o$ via Eqns. 19 and 20, ensuring stability without manual tuning even as the training dynamics evolve, overcoming the parameter sensitivity and environment-specific retuning that plague PID-based and classical Lagrangian methods.

Finally, large values of $\lambda$ may destabilize policy learning by forcing aggressive gradient steps. Following the previous method (Stooke et al., 2020), we adopt a rescaled optimization objective:

$$
\theta^*(\lambda) = \arg\max_\theta \frac{1}{1+\lambda} \big( J(\pi_\theta) - \lambda J_c(\pi_\theta) \big),
$$

which tempers the effect of $\lambda$ while preserving constraint enforcement. This adjustment yields smoother policy updates and makes ADRC Lagrangian straightforward to integrate into standard Safe RL algorithms. The full training procedure is summarized in Algorithm 1.

## 5 Experiments

In this section, we conduct a series of experiments to evaluate the performance of our ADRC Lagrangian method. Specifically, we aim to answer the following questions: (1) Does it reduce training

oscillations, having smaller phase-lag of the response, thereby minimizing constraint violations compared to baseline methods? (2) How robust is the method when facing different parameters that we set? (3) Can the ADRC Lagrangian method be applied universally to any Lagrangian-based safe RL algorithm? (4) How does the ADRC-based Lagrangian method perform upon convergence? (5) How does our method compare against existing state-of-the-art Safe RL approaches?

We will address Questions 1 and 3 in Section 5.2, Question 2 in Section 5.3, Question 4 in Appendix F.9 and Question 5 in Section 5.3.3 and Appendix F.4.

## 5.1 EXPERIMENTAL SETUPS

**Environments** We use the OmniSafe (Ji et al., 2024) to do the experiments, which provides a comprehensive and reliable benchmark for safe RL algorithms. We conduct our experiments using four safe RL algorithms with various combinations of agents and tasks. For more detail about the environment, please refer to Appendix E.2.

**Algorithms and Baseline** We utilize two categories of algorithms that have been implemented in Omnisafe: on-policy methods (PPO, TRPO) and off-policy methods (DDPG, TD3). We use the classical Lagrangian method and the PID Lagrangian method as baselines.

## 5.2 PERFORMANCE EVALUATION

Figure 1 illustrates the learning curves of PPO algorithms using various Lagrangian methods across different tasks. Our ADRC methods demonstrate superior constraint satisfaction and a shorter response lag while maintain competitive reward compared to the baseline. To better compare performance with the baseline, Table 6 presents the violation rates, violation magnitude, and average cost during the training phase.

To illustrate the universal applicability of our ADRC Lagrangian methods to any Lagrangian-based safe RL algorithm, we conducted additional experiments on various algorithms in the RacecarGoal and CarPush environments. The learning curves for the RacecarGoal tasks are presented in Figure 4, and the performance metrics are summarized in Table 1. Further experimental results, including detailed analyses, are provided in Appendix F.2.

Table 1: The proportion of constraint violations during training, the average violation magnitude, and the average cost for various algorithms. Our ADRC method consistently outperforms others.

| Algorithm | CarPush | | | RacecarGoal | | |
|---|---|---|---|---|---|---|
| | Vio. Rate (%) | Magnitude | Avg. Cost | Vio. Rate (%) | Magnitude | Avg. Cost |
| CPPOLag | 68.45 ± 18.98 | 21.48 ± 16.81 | 43.38 ± 18.71 | 80.87 ± 19.17 | 31.18 ± 18.99 | 54.24 ± 21.01 |
| CPPOPID | 62.36 ± 10.66 | 12.40 ± 3.08 | 33.74 ± 3.88 | 72.30 ± 24.96 | 27.11 ± 15.58 | 49.02 ± 18.75 |
| CPPOADRC | **42.23 ± 15.34** | **11.68 ± 7.99** | **29.16 ± 10.19** | **47.08 ± 21.58** | **12.31 ± 9.34** | **30.12 ± 12.74** |
| DDPGLag | 65.52 ± 4.60 | 12.53 ± 0.72 | 35.03 ± 0.53 | 71.44 ± 9.47 | 18.47 ± 4.13 | 40.76 ± 5.00 |
| DDPGPID | 52.43 ± 0.12 | 7.25 ± 0.93 | 28.35 ± 0.44 | 72.05 ± 4.09 | 17.93 ± 2.81 | 40.29 ± 2.99 |
| DDPGADRC | **47.36 ± 1.90** | **2.88 ± 0.57** | **21.55 ± 0.29** | **68.41 ± 5.77** | **17.81 ± 1.34** | 39.41 ± 2.14 |
| TD3Lag | 80.94 ± 5.87 | 20.68 ± 3.94 | 43.43 ± 4.23 | 75.26 ± 2.13 | 19.93 ± 1.40 | 42.57 ± 1.59 |
| TD3PID | 70.47 ± 10.44 | 17.00 ± 1.29 | 38.90 ± 2.83 | 73.31 ± 6.06 | 19.19 ± 2.72 | 41.57 ± 3.22 |
| TD3ADRC | **40.62 ± 8.51** | **2.85 ± 0.31** | **20.65 ± 2.46** | **71.24 ± 3.00** | **17.55 ± 3.57** | 39.71 ± 3.94 |
| TRPOLag | 54.86 ± 6.74 | 10.46 ± 4.56 | 30.97 ± 4.69 | 64.31 ± 23.37 | 22.89 ± 18.69 | 43.67 ± 21.56 |
| TRPOPID | 44.79 ± 2.84 | 7.34 ± 1.31 | 25.84 ± 0.88 | 49.33 ± 15.00 | 11.70 ± 7.07 | 30.94 ± 8.81 |
| TRPOADRC | **29.11 ± 3.70** | **3.44 ± 1.21** | **20.48 ± 0.99** | **34.03 ± 8.06** | **6.16 ± 2.37** | **22.02 ± 3.61** |

## 5.3 PARAMETER SENSITIVITY ANALYSIS

To demonstrate whether our ADRC Lagrangian methods are robust to different value of parameters, we test the parameter $c_r$ in Section 5.3.1, we test the control gain $k_{ap}$ and $k_{ad}$ in Section 5.3.2.

### 5.3.1 Tuning Parameter $c_r$

We selected five different values for $c_r$ and conducted the experiments using TRPO under the RacecarGoal benchmarks. As shown in Table 2, the results show that all selected values of the parameter $c_r$ outperform the baseline, demonstrating robustness to parameter variations. More experimental results can be found at Appendix F.5.3.

Table 2: Performance comparison under RacecarGoal for TRPO and PPO with different $c_r$ values.

| Method | TRPO (RacecarGoal) | | | PPO (RacecarGoal) | | |
|---|---|---|---|---|---|---|
| | Vio. Rate (%) | Magnitude | Avg. Cost | Vio. Rate (%) | Magnitude | Avg. Cost |
| Lag | 87.33 | 37.36 | 61.53 | 84.35 | 30.16 | 53.38 |
| PID | 44.60 | 7.04 | 26.15 | 79.25 | 23.88 | 46.44 |
| $c_r = 0.05$ | **33.98** | **5.25** | **20.83** | **34.38** | **3.90** | **22.45** |
| $c_r = 0.1$ | **29.05** | **3.44** | **18.95** | **33.08** | **5.78** | **21.22** |
| $c_r = 0.15$ | **31.25** | **5.34** | **21.16** | **52.88** | **10.69** | **31.37** |
| $c_r = 0.2$ | **40.65** | **6.10** | **23.67** | **48.95** | **8.77** | **26.91** |
| $c_r = 0.25$ | **38.50** | **6.71** | **23.40** | **62.83** | **13.37** | **33.99** |

### 5.3.2 Tuning Parameters $k_{ap}$ and $k_{ad}$

We investigate the impact of the tuning parameters $k_{ap}$ and $k_{ad}$ under the RacecarGoal environment. For $k_{ap}$, experiments were conducted using PPO; for $k_{ad}$, we used TRPO. In both cases, we evaluated three orders of magnitude: 1, 0.1, and 0.01. As shown in Table 3, the results demonstrate that all tested values of $k_{ap}$ and $k_{ad}$ significantly outperform existing methods, including the PID method, the Lagrangian method, and the Classical Lagrangian method. For additional experimental details, please refer to Appendices F.5.1 and F.5.2.

Table 3: Sensitivity analysis of tuning parameters $k_{ap}$ and $k_{ad}$ under RacecarGoal environment.

| $k_{ap}$ Setting | Vio. Rate (%) | Mag. | Avg. Cost | $k_{ad}$ Setting | Vio. Rate (%) | Mag. | Avg. Cost |
|---|---|---|---|---|---|---|---|
| $k_{ap} = 1$ | **69.98** | **18.62** | **39.87** | $k_{ad} = 1$ | **28.55** | **6.17** | **20.23** |
| $k_{ap} = 0.1$ | **33.08** | **5.78** | **21.22** | $k_{ad} = 0.1$ | **38.25** | 9.12 | **23.92** |
| $k_{ap} = 0.01$ | **20.43** | **2.50** | **15.42** | $k_{ad} = 0.01$ | **39.08** | **5.75** | **23.33** |
| PID | 79.25 | 23.88 | 46.44 | PID | 44.60 | 7.04 | 26.15 |
| Lag | 84.35 | 30.16 | 53.38 | Lag | 87.33 | 37.36 | 61.53 |

### 5.3.3 Comparison with State-of-the-Art Safe RL Algorithms

To demonstrate the broader applicability and effectiveness of our ADRC-Lagrangian framework beyond traditional Lagrangian methods, we conduct comprehensive comparisons with state-of-the-art safe RL algorithms. Our evaluation includes both Lagrangian-based methods (RCPO and PDO) Tessler et al. (2018); Chow et al. (2018a) and non-Lagrangian approaches such as CUP Yang et al. (2022) and IPO Liu et al. (2019). This comparison validates our method's superiority across different safe RL paradigms and confirms that the benefits stem from ADRC's adaptive control principles rather than merely being artifacts of the Lagrangian framework.

As detailed in Appendix F.4, ADRC variants consistently improve training stability by reducing violation rates, violation magnitudes, and average costs, while maintaining or even enhancing final task performance.

### 5.4 Ablation Study

To evaluate the effectiveness of our proposed dynamic parameter adjustment and transient process, we conducted ablation studies, with results summarized in Table 4. In this table, "Delete $r(t)$" refers to the removal of the dynamic adjustment component $r(t)$, while "Delete $w_0$" refers to the exclusion of the transient weight $w_0$ from the algorithm. The results show that removing either component results in a clear performance degradation in terms of violation rate, violation magnitude, and average cost. However, even with these removals, the performance of our approach remains superior to the

baseline PID method, demonstrating the robustness of our framework. Additionally, the complete ADRC method achieves the best results across all metrics, further highlighting the significance of combining both $r(t)$ and $w_0$ in achieving optimal performance. For further details and results, please refer to Appendix F.7.

Table 4: Ablation study of CPPO algorithm under RacecarGoal.

| Method | Vio. Rate(%) | Magnitude | Avg. Cost |
|---|---|---|---|
| Delete $r(t)$ | 65.40 | 13.99 | 36.38 |
| Delete $w_0$ | 54.08 | 15.23 | 34.66 |
| ADRC (Ours) | **33.08** | **5.78** | **21.22** |
| PID | 79.25 | 23.88 | 46.44 |
| Lag | 84.35 | 30.16 | 53.38 |

## 6 CONCLUSION

In this paper, we introduce an effective method to optimize the Lagrangian multiplier update process in safe RL, reducing oscillation during training. First, we define the safe RL learning process as a closed-loop system. Next, we introduce ADRC, a robust and innovative controller that estimates and compensates for overall disturbances. We consider the current cost as the control objective and design a second-order closed-loop system to regulate this cost, ensuring compliance with the safety constraint. Additionally, we employed a reduced-order ESO (Zhong et al., 2020a) to estimate the unknown nonlinear function affecting agent costs, revealing that prior approaches, including PID Lagrangian and classic Lagrangian methods, in form, are special cases of our approach. Theoretical proofs and experimental results demonstrate the effectiveness and superiority of our method over existing approaches. While our method is validated extensively in simulated environments, applying it to real-world robotics or safety-critical systems remains an important direction for future work.

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

## A  PROCESS OF SOLVING ODE

We consider the ODE:

$$\ddot{r} = -2c_r\dot{r} - c_r^2(r - d), \quad r(0) = x_1(0), \quad \dot{r}(0) = x_2(0), \tag{22}$$

which can be rewritten in the standard form:

$$\ddot{r} + 2c_r\dot{r} + c_r^2 r = c_r^2 d. \tag{23}$$

First, we solve the associated homogeneous equation:

$$\ddot{r} + 2c_r\dot{r} + c_r^2 r = 0. \tag{24}$$

Assuming a solution of the form $r_h(t) = e^{\lambda t}$ and substituting into Eqn. 24, we obtain the characteristic equation:

$$\lambda^2 + 2c_r\lambda + c_r^2 = 0. \tag{25}$$

Solving for $\lambda$ yields:

$$\lambda = -c_r. \tag{26}$$

Thus, the general solution for the homogeneous equation is:

$$r_h(t) = (A + Bt)e^{-c_r t}, \tag{27}$$

where $A$ and $B$ are constants that determined by the initial value.

For the nonhomogeneous equation Eqn. 23, we assume a particular solution $r_p(t) = C$. Substituting into Eqn. 23 gives:

$$C = d. \tag{28}$$

The general solution to Eqn. 23 is:

$$r(t) = (A + Bt)e^{-c_r t} + d. \tag{29}$$

To determine $A$ and $B$, we use the initial conditions. From $r(0) = x_1(0)$:

$$A + d = x_1(0) \quad \Rightarrow \quad A = x_1(0) - d. \tag{30}$$

The derivative $\dot{r}(t)$ is:

$$\dot{r}(t) = (B - c_r(A + Bt)) e^{-c_r t}. \tag{31}$$

Substitute $t = 0$ into Eqn. 31 and use $\dot{r}(0) = x_2(0)$:

$$B - c_r A = x_2(0). \rightarrow B = x_2(0) + c_r(x_1(0) - d). \tag{32}$$

Substitute $A$ and $B$ into Eqn. 29 to obtain the final solution:

$$r(t) = d + (x_1(0) - d)e^{-c_r t} + (x_2(0) + c_r(x_1(0) - d)) te^{-c_r t}. \tag{33}$$

## B  SIMPLIFY THE ESO

We consider the control law:

$$u = k_{ap}(x_1 - r) + k_{ad}(x_2 - \dot{r}) + \hat{f} - \ddot{r}, \quad k_{ap} > 0, \quad k_{ad} > 0, \tag{34}$$

where $k_{ap}$ and $k_{ad}$ are tuning parameters, and the term $\hat{f}$ compensates for disturbances.

Substitute Eqn. 34 into the Eqn. 15:

$$\dot{\xi} = -\omega_o\xi - \omega_o^2 x_2 - \omega_o \left( k_{ap}(x_1 - r) + k_{ad}(x_2 - \dot{r}) + \hat{f} - \ddot{r} \right). \tag{35}$$

Simplify Eqn. 35:

$$\dot{\xi} = -\omega_o\xi - \omega_o^2 x_2 + \omega_o k_{ap}(x_1 - r) + \omega_o k_{ad}(x_2 - \dot{r}) + \omega_o\hat{f} - \omega_o\ddot{r}. \tag{36}$$

Given $\hat{f} = \xi + \omega_o x_2$, we have $\xi = \hat{f} - \omega_o x_2$ and $\dot{\xi} = \dot{\hat{f}} - \omega_o\dot{x}_2$. Substitute this into Eqn. 36:

$$\dot{\hat{f}} - \omega_o\dot{x}_2 = -\omega_o(\hat{f} - \omega_o x_2) - \omega_o^2 x_2 + \omega_o k_{ap}(x_1 - r) + \omega_o k_{ad}(x_2 - \dot{r}) + \omega_o\hat{f} - \omega_o\ddot{r}. \tag{37}$$

Simplify further:

$$\dot{\hat{f}} = \omega_o k_{ap}(x_1 - r) + \omega_o k_{ad}(x_2 - \dot{r}) - \omega_o \ddot{r} + \omega_o \dot{x}_2. \tag{38}$$

Integrating Eqn. 38, we have:

$$\hat{f} = \omega_o k_{ad}(x_1 - r) + \omega_o(x_2 - \dot{r}) + \omega_o k_{ap} \int_0^t (x_1(\tau) - r(\tau))d\tau. \tag{39}$$

Substitute Eqn. 39 back into Eqn. 34:

$$u = (k_{ap} + \omega_o k_{ad})(x_1 - r) + (k_{ad} + \omega_o)(x_2 - \dot{r}) + \omega_o k_{ap} \int_0^t (x_1(\tau) - r(\tau))d\tau - \ddot{r}. \tag{40}$$

## C    THEORETICAL DETAILS

### C.1    CONVERGENCE AND ERROR BOUNDS

For completeness, we provide the detailed stability conditions and error analysis that were summarized in Sec. 4.4. Recall the disturbance class

$$\mathcal{F} = \Big\{ f \mid f(x_1, x_2, t) = h(x_1, x_2) + w(t),$$
$$\Big| \frac{\partial h}{\partial x_1} \Big| \le L_1, \quad \Big| \frac{\partial h}{\partial x_2} \Big| \le L_2, \tag{41}$$
$$|w(t)| \le L_3, \quad |\dot{w}(t)| \le L_3, \quad \lim_{t \to \infty} w(t) = k \Big\},$$

where $L_1, L_2$ bound state-dependent sensitivity, and $L_3$ bounds the magnitude and rate of purely time-dependent fluctuations.

**Stability manifold and lower bound.**    To guarantee convergence, the observer gain $\omega_o$ must lie in a feasible region determined by the characteristic polynomial

$$\Omega = \Big\{ \omega \in \mathbb{R} \mid n_0 \omega^4 + n_1 \omega^3 + n_2 \omega^2 + n_3 \omega + n_4 = 0 \Big\}, \tag{42}$$

where the coefficients $n_i$ depend on $(k_{ap}, k_{ad})$ and the constants $L_1, L_2, L_3$. Let

$$\bar{\omega}_o = \begin{cases} \max\{\omega \mid \omega \in \Omega\}, & \text{if } \Omega \neq \emptyset, \\ 0, & \text{otherwise.} \end{cases}$$

The admissible observer gains are then those satisfying

$$\omega_o > \omega_o^* = \max \Big\{ \bar{\omega}_o, \ 0, \ \tfrac{L_1 - k_{ap}}{k_{ad}}, \ L_2 - k_{ad} \Big\}. \tag{43}$$

Suppose $f \in \mathcal{F}$ and $\omega_o > \omega_o^*$. Then:

- (*Convergence*) For any initial condition and any cost limit $d \in \mathbb{R}$, the system converges:

$$\lim_{t \to \infty} x_1(t) = d, \qquad \lim_{t \to \infty} x_2(t) = 0.$$

- (*Bounded estimation error*) Let $e(t) = r(t) - x_1(t)$ be the tracking error and $e_f(t) = \hat{f} - f(x_1, x_2, t)$ the disturbance estimation error. Then there exist constants $\eta_1, \eta_2$ such that

$$|\ddot{e}(t) + k_{ad}\dot{e}(t) + k_{ap}e(t)| = |e_f(t)| \ \le \ \eta_1 e^{-\omega_o t} + \tfrac{\eta_2}{\omega_o}, \quad t \ge 0.$$

The first result shows that as long as the observer gain exceeds the lower bound $\omega_o^*$, the cumulative cost $x_1$ converges to the constraint threshold $d$ without oscillation. The second result shows that the estimation error is always bounded, decays over time, and can be reduced by choosing larger $\omega_o$. Together, these properties justify our claim in the main text that ADRC Lagrangian guarantees convergence and robustness without fragile manual tuning. The detailed proofs follow directly from Zhong et al. (2020a) and related ADRC analyses.

## C.2 PROOF OF THEOREM 4.2

*Proof.* From Theorems demonstrated by Zhong et al. (2020a), we know that both $f$ and $\dot{f}$ are bounded. The error dynamics are given by:

$$\begin{cases} \dot{e} = e_d, \\ \dot{e}_d = -k_{ap}e - k_{ad}e_d + e_f. \end{cases} \tag{44}$$

Taking the second derivative of $e$, we have:

$$\ddot{e} = -k_{ap}e - k_{ad}e_d + e_f, \tag{45}$$

or equivalently:

$$e_f = \ddot{e} + k_{ad}\dot{e} + k_{ap}e. \tag{46}$$

Applying the Laplace transform to Eqn. 46, we obtain:

$$E(s) = \frac{1}{s^2 + k_{ad}s + k_{ap}} E_f(s). \tag{47}$$

And we know that, the dynamics of $e_f$ are given by:

$$\dot{e}_f = -\omega_o e_f - \dot{f}. \tag{48}$$

Taking the Laplace transform of Eqn. 48, we have:

$$E_f(s) = G_{e_f}(s)F(s), \quad G_{e_f}(s) = \frac{s}{s + \omega_o}. \tag{49}$$

Similarly, applying the Laplace transform to the integral form of $e_f$, we obtain:

$$E_{f_I}(s) = \frac{s^2 + k_d s + k_p}{s^2 + k_{ad}s + k_{ap}} E_f(s), \tag{50}$$

and the transfer function for $E_{f_I}(s)$ can be expressed as:

$$E_{f_I}(s) = G_{e_{f_I}}(s)F(s), \quad G_{e_{f_I}}(s) = \frac{s^3 + k_d s^2 + k_p s}{(s + \omega_o)(s^2 + k_{ad}s + k_{ap})}. \tag{51}$$

The ratio of the squared magnitudes of $G_{e_f}(i\omega)$ and $G_{e_{f_I}}(i\omega)$ is given by:

$$\frac{|G_{e_f}(i\omega)|^2}{|G_{e_{f_I}}(i\omega)|^2} = \frac{(k_{ap} - \omega^2)^2 + k_{ad}^2\omega^2}{(k_p - \omega^2)^2 + k_d^2\omega^2} < 1. \tag{52}$$

As $t \to \infty$, we have:

$$\lim_{t \to \infty} \frac{e_f(t)}{e_{f_I}(t)} = \lim_{s \to 0} \frac{sE_f(s)}{sE_{f_I}(s)} = \frac{k_{ap}}{k_{ap} + \omega_o k_{ad}}. \tag{53}$$

This completes the first part of this theorem.

Now, consider the phase angle of a transfer function $G(i\omega)$, defined as:

$$\arg(G(i\omega)) = \tan^{-1}\left(\frac{\text{Im}(G(i\omega))}{\text{Re}(G(i\omega))}\right). \tag{54}$$

For $G_{e_f}(i\omega)$ and $G_{e_{f_I}}(i\omega)$, we have:

$$\arg(G_{e_f}(i\omega)) = \tan^{-1}\left(\frac{\omega}{\omega_o}\right), \tag{55}$$

and

$$\arg(G_{e_{f_I}}(i\omega)) = \tan^{-1}\left(\frac{k_{ad}\omega}{k_{ap} - \omega^2}\right). \tag{56}$$

For any $\omega$, if we choose $\omega_o > \max\left\{\frac{k_{ap}-\omega^2}{k_{ad}}, \omega_o^*\right\}$, it follows that:

$$\frac{\omega}{\omega_o} < \frac{k_{ad}\omega}{k_{ap}-\omega^2}. \tag{57}$$

Thus, we conclude:

$$\tan^{-1}\left(\frac{\omega}{\omega_o}\right) < \tan^{-1}\left(\frac{k_{ad}\omega}{k_{ap}-\omega^2}\right), \tag{58}$$

or equivalently:

$$\arg(G_{e_f}(i\omega)) < \arg(G_{e_{f_I}}(i\omega)). \tag{59}$$

This completes the second part of this theorem.

$\square$

### C.3 Theoretical Guarantees

We provide three guarantees for ADRC-Lagrangian in Safe RL: (i) robustness to nonstationarity and noise through an ISS-type tracking bound on the cost return, (ii) a high-probability *non-decreasing reward* guarantee for one trust-region step using an ESO-based lower-confidence-bound (LCB) advantage, and (iii) a *bounded average* constraint violation guarantee in terms of Safe-RL hyperparameters. We also give an optional corollary showing finite cumulative violation if one tracks a safety margin below the threshold.

**Notation.** Let $J_r(\pi)$ and $J_c(\pi)$ be the discounted reward and cost returns of a policy $\pi$, and let $\{\pi_{\theta_t}\}_{t\geq 0}$ be the (continuous-time interpolation of the) policy iterates produced by a Safe RL algorithm. We define

$$x_1(t) := J_c(\pi_{\theta_t})$$

as the cost signal that ADRC tracks, and let $r(t)$ be a second-order critically damped reference approaching the safety threshold $d$ (Sec. 4.2). The tracking error is

$$e(t) := x_1(t) - r(t), \qquad e_d(t) := \dot{e}(t).$$

The ADRC error channel (Sec. 4.3) is modeled as a second-order LTI system driven by an unknown "lumped disturbance" $f$:

$$\ddot{e}(t) + k_{ad}\dot{e}(t) + k_{ap}e(t) = e_f(t), \qquad k_{ap} > 0, \ k_{ad} > 0, \tag{60}$$

where

$$e_f(t) := \hat{f}(t) - f(x_1(t), x_2(t), t)$$

is the ESO disturbance-estimation error. The reduced-order ESO used in Eqns. (15)–(17) yields the *exact* identity

$$\dot{e}_f(t) = -\omega_o e_f(t) - \dot{f}(x_1(t), x_2(t), t), \qquad \omega_o > 0. \tag{61}$$

For (60), define the natural frequency $\omega_n = \sqrt{k_{ap}}$ and the damping ratio $\zeta = \frac{k_{ad}}{2\sqrt{k_{ap}}}$.

### C.3.1 Safe-RL induced disturbance dynamics

We now tie the disturbance $f$ and its temporal variation directly to Safe RL quantities: trust-region radius, advantage bounds, batch size, and sampling noise.

**Assumption C.1** (Bounded dynamics of policy-induced cost). *Consider an iterative Safe RL algorithm updating at $t_k = k\Delta t$ with $\mathrm{KL}(\pi_{k+1}\|\pi_k) \leq \delta$. Let the cost advantage be bounded by $|A_c^{\pi_k}| \leq \varepsilon_c$. The standard Trust Region bound implies the cost shift is limited by $|\Delta J_c(\pi_k)| \leq C_{\mathrm{TR}}^{(c)}\sqrt{\delta}$, where $C_{\mathrm{TR}}^{(c)} = \frac{2\gamma\varepsilon_c\sqrt{2}}{(1-\gamma)^2}$.*

*Simultaneously, the cost $J_c$ is estimated from $N$ trajectories with single-sample range $B_c$ and empirical variance $\hat{v}_c$. For a confidence level $1-\alpha$, we define the empirical Bernstein half-width as:*

$$r_{N,\alpha}^{(c)} := \sqrt{\frac{2\hat{v}_c\log(3/\alpha)}{N}} + \frac{3B_c\log(3/\alpha)}{N}. \tag{62}$$

*By the empirical Bernstein inequality, the estimation error is bounded by $r_{N,\alpha}^{(c)}$ with probability at least $1 - \alpha$.*

*We assume the continuous-time cost $x_1(t)$ evolves smoothly, and the lumped disturbance $f$ aggregates policy-induced non-stationarity and sampling noise. Specifically, its rate of change is bounded by:*

$$\sup_{t \geq 0} |\dot{f}(x_1, x_2, t)| \leq L_f(\delta, N, \alpha) := \frac{\kappa_c}{\Delta t} \left( C_{\mathrm{TR}}^{(c)} \sqrt{\delta} + r_{N,\alpha}^{(c)} \right), \tag{63}$$

*where $\kappa_c$ is a Lipschitz constant relating discrete updates to continuous derivatives.*

**Remark C.2.** *Assumption C.1 makes the Safe RL dependence explicit: faster policy changes (larger $\delta$ or smaller $\Delta t$), higher-cost advantage magnitude $\varepsilon_c$, and smaller batch size $N$ all increase $L_f$ and thus enlarge the disturbance seen by ADRC. Conversely, larger $N$ or smaller $\delta$ shrink $L_f$.*

For notational simplicity, we write $L_f := L_f(\delta, N, \alpha)$ in the sequel.

### C.3.2 ROBUST TRACKING: ISS-TYPE BOUNDS WITH DAMPING-AWARE $L_1$ CONSTANTS

We first bound the ESO error explicitly, then propagate it to the tracking error using a damping-aware bound on the impulse $L_1$ norm of (60).

**Lemma C.3** (ESO error bound). *Under Assumption C.1, the solution of* (61) *satisfies, for all $t \geq 0$,*

$$|e_f(t)| \leq e^{-\omega_o t} |e_f(0)| + \frac{L_f}{\omega_o}. \tag{64}$$

*Proof.* By variation of constants,

$$e_f(t) = e^{-\omega_o t} e_f(0) - \int_0^t e^{-\omega_o(t-\tau)} \dot{f}(\tau) \, d\tau.$$

Taking absolute values and using $|\dot{f}(\tau)| \leq L_f$ from Assumption C.1 yields

$$|e_f(t)| \leq e^{-\omega_o t}|e_f(0)| + \int_0^t e^{-\omega_o(t-\tau)} L_f \, d\tau = e^{-\omega_o t}|e_f(0)| + \frac{L_f}{\omega_o}(1 - e^{-\omega_o t}) \leq e^{-\omega_o t}|e_f(0)| + \frac{L_f}{\omega_o}.$$

$\square$

**Lemma C.4** (Impulse $L_1$ norm of the second-order tracker). *Consider the stable LTI system $H(s) = \frac{1}{s^2 + k_{ad}s + k_{ap}}$ with impulse response $h(t)$. Let $\omega_n = \sqrt{k_{ap}}$ and $\zeta = \frac{k_{ad}}{2\sqrt{k_{ap}}}$. Then*

$$\|h\|_{L_1} := \int_0^\infty |h(t)| \, dt \leq \frac{C_h(\zeta)}{k_{ap}}, \qquad C_h(\zeta) := \begin{cases} 1, & \zeta \geq 1 \quad \text{(critical/overdamped)}, \\ \frac{1}{\zeta\sqrt{1-\zeta^2}}, & 0 < \zeta < 1 \quad \text{(underdamped)}. \end{cases} \tag{65}$$

*Moreover, $\int_0^\infty h(t) \, dt = H(0) = \frac{1}{k_{ap}}$.*

*Proof.* The DC gain identity $\int_0^\infty h(t) \, dt = H(0)$ is standard. If $\zeta \geq 1$ the impulse response is nonnegative and thus $\|h\|_{L_1} = \int_0^\infty h(t) \, dt = 1/k_{ap}$. If $0 < \zeta < 1$, the impulse is $h(t) = \frac{1}{\omega_d} e^{-\zeta\omega_n t} \sin(\omega_d t)$ with $\omega_d = \omega_n \sqrt{1-\zeta^2}$. Using $|\sin(\cdot)| \leq 1$ we obtain

$$\|h\|_{L_1} \leq \int_0^\infty \frac{1}{\omega_d} e^{-\zeta\omega_n t} dt = \frac{1}{\zeta\omega_n\omega_d} = \frac{1}{\zeta\sqrt{1-\zeta^2}\,\omega_n^2} = \frac{1}{\zeta\sqrt{1-\zeta^2}\,k_{ap}}.$$

$\square$

**Theorem C.5** (ISS-type robust tracking bound for Safe RL). *Under Assumption C.1, for the error dynamics* (60) *and ESO gain $\omega_o > 0$, there exist $C_0, \alpha > 0$ (depending on $k_{ap}, k_{ad}$ and initial $(e(0), \dot{e}(0))$) such that, for all $t \geq 0$,*

$$|e(t)| \leq C_0 e^{-\alpha t} + \frac{C_h(\zeta)}{k_{ap}} \left( |e_f(0)| + \frac{L_f(\delta, N, \alpha)}{\omega_o} \right), \tag{66}$$

*where $C_h(\zeta)$ is given in* (65). *In particular,*

$$\limsup_{t \to \infty} |e(t)| \leq \frac{C_h(\zeta)}{k_{ap}} \frac{L_f(\delta, N, \alpha)}{\omega_o} = \mathcal{O}\left(\frac{\sqrt{\delta}}{\omega_o \Delta t}\right) + \mathcal{O}\left(\frac{1}{\omega_o \Delta t} \sqrt{\frac{\log(1/\alpha)}{N}}\right). \tag{67}$$

*Proof.* The solution of (60) can be written as $e(t) = e_{\text{hom}}(t) + (h * e_f)(t)$, where $h$ is the impulse response of $H(s)$ and $*$ denotes convolution. The homogeneous solution decays exponentially because the characteristic polynomial has positive coefficients: there exist $C_0, \alpha > 0$ such that $|e_{\text{hom}}(t)| \leq C_0 e^{-\alpha t}$.

For the forced part, Young's inequality and Lemmas C.3–C.4 give

$$|(h * e_f)(t)| \leq \|h\|_{L_1} \sup_{0 \leq \tau \leq t} |e_f(\tau)| \leq \frac{C_h(\zeta)}{k_{ap}} \Big( |e_f(0)| + \frac{L_f}{\omega_o} \Big),$$

where we used (64). Combining the homogeneous and forced contributions yields (66). The $\lim \sup$ statement (67) follows by letting $t \to \infty$ and substituting the explicit Safe-RL expression (63) for $L_f$. $\qquad\square$

**Remark C.6** (Interpretation for Safe RL training). *Theorem C.5 states that in Safe RL, the tightness of constraint enforcement (tube radius of $J_c(\pi_{\theta_t})$ around $r(t)$) is controlled by: (i) the ESO bandwidth $\omega_o$, (ii) the trust-region radius $\delta$ and update interval $\Delta t$, and (iii) the batch size $N$ and target confidence $\alpha$. Larger $\omega_o$ and $N$ and smaller $\delta$ and $\alpha$ yield a tighter tube.*

### C.3.3 HIGH-PROBABILITY NON-DECREASING REWARD UNDER A TRUST REGION

We now show a *conditional* monotonic-improvement guarantee on $J_r$: if the LCB objective under a trust region exceeds a computable threshold, then reward is non-decreasing with high probability.

**Assumption C.7** (Trust region, ESO-based LCB advantage, and confidence level). *At update $t$, let $\pi_t$ be the behavior policy. We consider the trust-region problem*

$$\pi_{t+1} \in \arg \max_{\pi:\ \text{KL}(\pi \| \pi_t) \leq \delta} \mathcal{L}_t(\pi), \qquad \mathcal{L}_t(\pi) := \mathbb{E}_{s \sim d_{\pi_t},\, a \sim \pi(\cdot|s)}[\tilde{A}_t(s, a)],$$

*where $\tilde{A}_t(s, a) = \widehat{A}_t^{\text{ESO}}(s, a) - b_t$ is a lower-confidence bound (LCB) for the true advantage $A^{\pi_t}(s, a)$.*

*We assume $\widehat{A}_t^{\text{ESO}}$ is formed from $N$ trajectories using the ESO-based critic, and that single-sample (per-trajectory) advantages are bounded by $B_A$ with empirical variance $\hat{v}_A$. For a user-specified confidence level $\alpha \in (0, 1)$, we choose*

$$b_t := \underbrace{\sqrt{\frac{2\hat{v}_A \log(3/\alpha)}{N}} + \frac{3B_A \log(3/\alpha)}{N}}_{\text{empirical Bernstein radius}} + c_e\, \varepsilon_o, \tag{68}$$

*where $c_e$ is a Lipschitz constant that transports ESO bias on the cost channel to bias on the reward advantage, and $\varepsilon_o$ is the steady-state ESO bias radius from Theorem C.5. Then, by the empirical Bernstein inequality,*

$$\Pr\big[\tilde{A}_t(s, a) \leq A^{\pi_t}(s, a) \text{ for all } (s, a) \text{ visited at update } t\big] \geq 1 - \alpha.$$

**Lemma C.8** (TRPO performance bound for reward; Schulman et al. (2015)). *Let $\varepsilon_A := \max_{s,a} |A^{\pi_t}(s, a)|$. Then, for any $\pi$,*

$$J_r(\pi) - J_r(\pi_t) \geq \mathbb{E}_{s \sim d_{\pi_t},\, a \sim \pi}[A^{\pi_t}(s, a)] - \frac{2\gamma}{(1 - \gamma)^2} \varepsilon_A \sqrt{2\,\text{KL}(\pi \| \pi_t)}.$$

**Theorem C.9** (High-probability non-decreasing reward). *Under Assumption C.7 and Lemma C.8, define $C_{\text{TR}} := \frac{2\gamma}{(1-\gamma)^2} \varepsilon_A \sqrt{2}$. If the trust-region solution satisfies*

$$\mathcal{L}_t(\pi_{t+1}) \geq C_{\text{TR}} \sqrt{\delta},$$

*then, with probability at least $1 - \alpha$, we have $J_r(\pi_{t+1}) \geq J_r(\pi_t)$.*

*Proof.* With probability at least $1 - \alpha$, the choice of $b_t$ in (68) guarantees $\tilde{A}_t(s, a) \leq A^{\pi_t}(s, a)$ for all $(s, a)$ in the support of $d_{\pi_t}\pi_{t+1}$. Thus,

$$\mathbb{E}_{s \sim d_{\pi_t},\, a \sim \pi_{t+1}}[\tilde{A}_t(s, a)] \leq \mathbb{E}_{s \sim d_{\pi_t},\, a \sim \pi_{t+1}}[A^{\pi_t}(s, a)].$$

Applying Lemma C.8 with $\mathrm{KL}(\pi_{t+1}\|\pi_t) \leq \delta$ yields

$$J_r(\pi_{t+1}) - J_r(\pi_t) \;\geq\; \mathbb{E}_{d_{\pi_t},\pi_{t+1}}[A^{\pi_t}] \;-\; C_{\mathrm{TR}}\sqrt{\delta}.$$

Combining the two displays gives

$$J_r(\pi_{t+1}) - J_r(\pi_t) \;\geq\; \mathbb{E}_{d_{\pi_t},\pi_{t+1}}[\tilde{A}_t] \;-\; C_{\mathrm{TR}}\sqrt{\delta} \;=\; \mathcal{L}_t(\pi_{t+1}) - C_{\mathrm{TR}}\sqrt{\delta}.$$

By assumption, $\mathcal{L}_t(\pi_{t+1}) \geq C_{\mathrm{TR}}\sqrt{\delta}$, so the right-hand side is nonnegative. Thus, conditioned on the high-probability event in Assumption C.7, we have $J_r(\pi_{t+1}) \geq J_r(\pi_t)$, and the overall probability of this event is at least $1 - \alpha$ by construction of $b_t$. $\qquad\square$

**Remark C.10** (How high is "high probability"?). *For fixed batch size $N$, range $B_A$, variance $\hat{v}_A$, and ESO bias radius $\varepsilon_o$, one can also treat $b_t$ as given and invert* (68) *to obtain an* achieved *confidence level*

$$\alpha \;\leq\; 3\exp\Big(-\frac{N\,(b_t - c_e\varepsilon_o)^2}{2\hat{v}_A + \frac{2}{3}B_A\,(b_t - c_e\varepsilon_o)}\Big),$$

*making the "$1-\alpha$" in Theorem C.9 fully quantitative.*

### C.3.4    BOUNDED AVERAGE CONSTRAINT VIOLATION (AND AN OPTIONAL STRONG VARIANT)

Because the steady tracking tube in Theorem C.5 has radius $\mathcal{O}(L_f/\omega_o)$ when $L_f > 0$, the *cumulative* violation $\int_0^\infty (x_1(t) - d)_+ \, dt$ generally diverges if one tracks $r(t) \to d$ exactly. The correct statement is that the *time-average* violation is bounded by the tube radius. A stronger, finite-cumulative-violation statement holds if the reference tracks a safety margin below $d$.

**Theorem C.11** (Bounded average constraint violation). *Under Assumption C.1, let $r(t)$ be the critically damped reference converging to $d$. Then*

$$\limsup_{T\to\infty} \frac{1}{T}\int_0^T \big(x_1(t) - d\big)_+ \, dt \;\leq\; \frac{C_h(\zeta)}{k_{ap}}\,\frac{L_f(\delta, N, \alpha)}{\omega_o}. \tag{69}$$

*Proof.* Using $x_1 - d = e + (r - d)$ and $(\cdot)_+ \leq |\cdot|$, we have $(x_1(t) - d)_+ \leq |e(t)| + |r(t) - d|$. The reference $r(t)$ is critically damped and converges exponentially to $d$, so there exist constants $C_r, c_r > 0$ such that

$$|r(t) - d| \leq C_r(1+t)e^{-c_r t}.$$

Thus

$$\frac{1}{T}\int_0^T |r(t) - d| \, dt \;\leq\; \frac{1}{T}\int_0^T C_r(1+t)e^{-c_r t} \, dt \;\xrightarrow[T\to\infty]{}\; 0.$$

For $|e(t)|$, Theorem C.5 gives $|e(t)| \leq C_0 e^{-\alpha t} + C_{ss}$ with

$$C_{ss} \;:=\; \frac{C_h(\zeta)}{k_{ap}}\Big(|e_f(0)| + \frac{L_f}{\omega_o}\Big).$$

Hence

$$\frac{1}{T}\int_0^T |e(t)| \, dt \;\leq\; \frac{1}{T}\int_0^T C_0 e^{-\alpha t} \, dt \;+\; C_{ss}.$$

The first term converges to 0 as $T \to \infty$, while the second term is constant. Since $e_f(0)$ is fixed independently of $T$, the $\limsup$ of the average violation is bounded by the contribution from $L_f/\omega_o$, yielding (69). $\qquad\square$

**Corollary C.12** (Finite cumulative violation with a safety margin). *Suppose the reference tracks a margin $\varepsilon > 0$ below the threshold, i.e., $r(t) \to d - \varepsilon$, and choose $\varepsilon$ such that*

$$\varepsilon \;>\; \frac{C_h(\zeta)}{k_{ap}}\,\frac{L_f(\delta, N, \alpha)}{\omega_o}.$$

*Then there exists $T_\varepsilon < \infty$ such that $(x_1(t) - d)_+ = 0$ for all $t \geq T_\varepsilon$, and consequently $\int_0^\infty (x_1(t) - d)_+ \, dt < \infty$.*

*Proof.* From Theorem C.5 with the margin reference, for any $\xi \in \left(0, \varepsilon - \frac{C_h(\zeta)}{k_{ap}} \frac{L_f}{\omega_o}\right)$ there exists $T$ such that for all $t \geq T$,

$$|e(t)| \leq \frac{C_h(\zeta)}{k_{ap}} \frac{L_f}{\omega_o} + \xi, \qquad |r(t) - (d - \varepsilon)| \leq \xi.$$

Then, for $t \geq T$,

$$x_1(t) - d = e(t) + r(t) - d \leq \left(\frac{C_h(\zeta)}{k_{ap}} \frac{L_f}{\omega_o} + \xi\right) - \varepsilon + \xi < 0$$

by the choice of $\xi$. Hence $(x_1(t) - d)_+ = 0$ for all $t \geq T_\varepsilon := T$, and the cumulative violation $\int_0^\infty (x_1(t) - d)_+ \, dt$ is finite. $\qquad \square$

**Takeaways.** (i) With $r(t) \to d$, the *average* constraint violation is $\mathcal{O}(L_f/\omega_o)$, where $L_f$ is explicitly controlled by the Safe-RL design choices $(\delta, N, \alpha, \Delta t)$. (ii) If one slightly tightens the reference to $d - \varepsilon$ with $\varepsilon$ larger than the tube radius from Theorem C.5, a finite time $T_\varepsilon$ exists beyond which there is *no* constraint violation, yielding finite cumulative violation.

## D IMPLEMENTATION DETAIL

This section outlines the details of the proposed method through the pseudo-code presented in Algorithm 1. The algorithm describes the procedure for adjusting the Lagrange multipliers using ADRC during training, ensuring robust performance and adaptability to varying conditions.

---

**Algorithm 1** ADRC-Controlled Lagrange Multiplier

---
**Require:** Choosed parameters $k_{ap}, k_{ad} \geq 0$
1: Integral: $I \leftarrow 0$
2: Previous Cost: $J_{C,\text{prev}} \leftarrow 0$
3: **repeat** at each iteration $t$
4:     Receive current cost $J_C$, reference cost $r$, its time derivative $\dot{r}, \ddot{r}$ and the optimal gain $\omega_o$.
5:     $\Delta \leftarrow J_C - r$
6:     $\partial \leftarrow (J_C - J_{C,\text{prev}} - \dot{r})_+$
7:     $I \leftarrow (I + \Delta)_+$
8:     $K_P \leftarrow k_{ap} + \omega_o k_{ad}$
9:     $K_I \leftarrow \omega_o k_{ap}$
10:     $K_D \leftarrow \omega_o + k_{ap}$
11:     $\lambda \leftarrow (K_P \Delta + K_I I + K_D \partial - \ddot{r})_+$
12:     $J_{C,\text{prev}} \leftarrow J_C$
13:     **return** $\lambda$

---

### D.1 HYPER-PARAMETERS

For the on-policy algorithms TRPO and PPO, we adopt the default parameters provided by Omnisafe (Ji et al., 2024), as detailed in Table 5. These parameters are consistently applied across all tasks.

## E EXPERIMENTAL DETAILS

### E.1 BASELINE

To comprehensively evaluate the effectiveness of our proposed ADRC method, we compare it against four well-established reinforcement learning algorithms. These include two off-policy algorithms, TD3 and DDPG, as well as two on-policy algorithms, PPO and TRPO. These algorithms were chosen due to their widespread adoption and proven performance across various RL tasks, providing a robust foundation for benchmarking.

Table 5: Parameter Comparison: ADRC, PID, and Lag Methods

| Parameter | ADRC | PID | Lag |
|---|---|---|---|
| $k_p(k_{ap})$ | 0.1 | 0.1 | - |
| $k_i$ | - | 0.01 | 0.035 |
| $k_d(k_{ad})$ | 0.01 | 0.01 | - |
| Delay | 10 | 10 | - |
| EMA $\alpha$ (Proportional Term) | 0.95 | 0.95 | - |
| EMA $\alpha$ (Derivative Term) | 0.95 | 0.95 | - |
| Sum Normalization | True | True | - |
| Derivative Normalization | False | False | - |
| Cost Limit | 25.0 | 25.0 | 25.0 |
| Max Penalty Coefficient | 100.0 | 100.0 | - |
| Initial Lagrangian Multiplier | 0.001 | 0.001 | 0.001 |
| Hidden Layer Sizes (Actor) | [64, 64] | [64, 64] | [64, 64] |
| Activation Function (Actor) | tanh | tanh | tanh |
| Hidden Layer Sizes (Critic) | [64, 64] | [64, 64] | [64, 64] |
| Activation Function (Critic) | tanh | tanh | tanh |
| Critic Learning Rate | 0.0003 | 0.0003 | 0.0003 |
| Linear Learning Rate Decay | True | True | True |
| Clip Ratio | 0.2 | 0.2 | 0.2 |
| Target KL | 0.02 | 0.02 | 0.02 |
| Use Max Gradient Norm | True | True | True |
| Max Gradient Norm | 40.0 | 40.0 | 40.0 |

### E.2 TASKS SPECIFICATION

To demonstrate the effectiveness and generalizability of our proposed methods, we conduct comprehensive experiments across diverse environments. We select three distinct agents, namely Car, Racecar, and Ant, each governed by different physical dynamics.

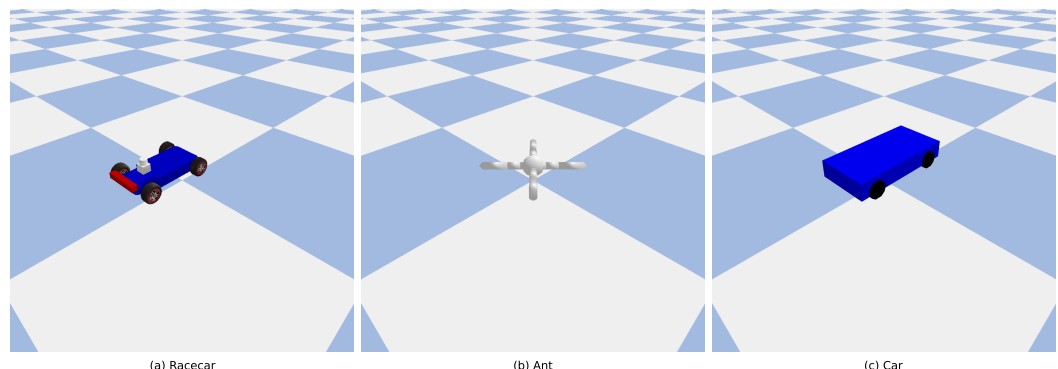

(a) Racecar          (b) Ant          (c) Car

Figure 2: Illustration of the three distinct agents used in our experiments. Car: A simple wheeled agent with low degrees of freedom. Racecar: A dynamic and agile wheeled agent with higher motion complexity. Ant: A multi-legged bionic agent with high degrees of freedom and non-linear dynamics. These agents represent diverse physical characteristics, allowing us to comprehensively evaluate the performance of our method under various physical dynamics.

As illustrated in Figure 2, the three agents represent diverse physical characteristics, enabling us to evaluate the performance of our method comprehensively across varying physical dynamics.

We consider four tasks in our experiments, as shown in Figure 3:

- **Goal Task** The robot must navigate to a specified goal region while avoiding hazards.
- **Button Task** The robot must press the correct button while avoiding hazards and gremlins, and must not press any wrong buttons.
- **Push Task** The robot must push a box to the goal region while avoiding hazards. A pillar is present but does not penalize collisions.
- **Circle Task** The robot moves around a circular track, without additional objects or hazards. This is mainly for testing circular navigation behavior.

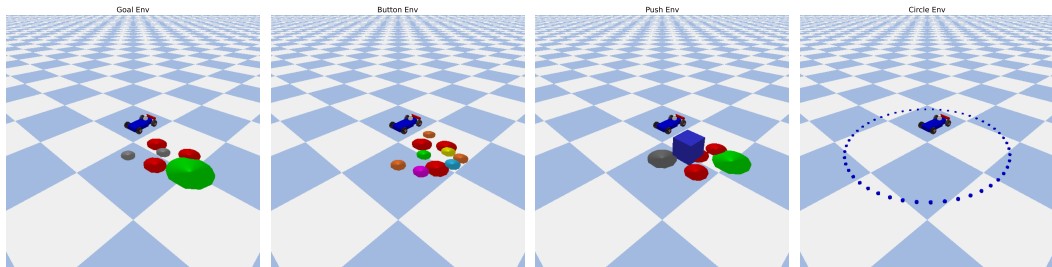

Figure 3: Four different tasks used in our experiments. (a) *Goal Task*: The agent must reach the goal area (blue sphere) without entering dangerous zones (red circles). (b) *Button Task*: The agent must press the correct button (green) and avoid pressing wrong ones (yellow, purple, etc.) or colliding with gremlins. (c) *Push Task*: The agent must push the box to the goal location (green circle) while avoiding hazards (red). (d) *Circle Env*: The agent moves around a simple circular track.

### E.3 EVALUATION METRICS

To comprehensively assess the performance of the proposed reinforcement learning algorithms, we employ several evaluation metrics. These metrics evaluate both the agent's ability to minimize costs and its adherence to safety constraints.

**Reward and Cost** The primary performance metrics are the **reward** and **cost**, which respectively measure the benefits and penalties accumulated by the agent over the course of an episode. For an episode consisting of $T$ time steps:

- The **return reward**, $R$, is defined as:

$$R = \sum_{t=1}^{T} r_t,$$

  where $r_t$ is the reward received at time step $t$. This metric reflects the agent's ability to achieve its objective efficiently.

- The **return cost**, $C$, is calculated as:

$$C = \sum_{t=1}^{T} c_t,$$

  where $c_t$ is the cost incurred at time step $t$. This metric assesses the penalties associated with the agent's actions, capturing its safety and resource efficiency.

**Violation Rate (Vio. Rate)** The Violation Rate quantifies the proportion of episodes during training in which the agent breaches predefined safety constraints. It is expressed as:

$$\text{Vio Rate} = \frac{N_{\text{violations}}}{N_{\text{total\_episodes}}},$$

where $N_{\text{violations}}$ is the number of episodes in which the agent's cumulative cost $C$ exceeds the allowable threshold, and $N_{\text{total\_episodes}}$ is the total number of training episodes. A lower violation rate indicates better safety performance.

**Constraint Violation Magnitude(Magnitude)**   The Violation Magnitude measures the severity of constraint violations in episodes where breaches occur. It is calculated as the average amount by which the return cost exceeds the allowable threshold across all violating episodes:

$$\text{Violation Magnitude} = \frac{1}{N_{\text{violations}}} \sum_{i=1}^{N_{\text{violations}}} \max(0, C_i - d),$$

where $C_i$ is the return cost of the $i$-th violating episode and $d$ is the cost threshold that we set. Smaller magnitudes indicate less severe constraint violations.

**Average Cost(Avg. Cost)**   To evaluate the overall performance during training, we calculate the Average Cost across all episodes:

$$\text{Average Cost} = \frac{1}{N_{\text{total\_episodes}}} \sum_{i=1}^{N_{\text{total\_episodes}}} C_i,$$

where $C_i$ is the return cost of the $i$-th episode.

By analyzing these metrics, we can comprehensively assess the effectiveness of each algorithm in achieving a balance between reward maximization and safety constraint adherence.

# F   MORE EXPERIMENTAL RESULTS

## F.1   TABLES AND FIGURES REFERENCED IN THE MAIN TEXT

Table 6: Constraint violation rate (Vio.), violation magnitude (Mag.), and average cost (Cost) during PPO training with various Lagrangian methods.

| Task | Method | Vio. (%) | Mag. | Cost |
|------|--------|----------|------|------|
| CarButton | Lag | $89.77 \pm 19.38$ | $59.77 \pm 39.05$ | $84.05 \pm 40.18$ |
| | PID | $85.09 \pm 16.67$ | $45.27 \pm 46.80$ | $68.95 \pm 47.94$ |
| | ADRC | $\mathbf{50.16 \pm 17.08}$ | $\mathbf{14.80 \pm 3.96}$ | $\mathbf{34.20 \pm 5.75}$ |
| CarCircle | Lag | $46.74 \pm 20.16$ | $15.85 \pm 17.40$ | $33.54 \pm 20.09$ |
| | PID | $52.78 \pm 17.62$ | $12.29 \pm 8.14$ | $31.24 \pm 10.31$ |
| | ADRC | $\mathbf{21.35 \pm 13.09}$ | $\mathbf{7.74 \pm 6.46}$ | $\mathbf{18.85 \pm 9.30}$ |
| RacecarGoal | Lag | $80.87 \pm 19.17$ | $31.18 \pm 18.99$ | $54.24 \pm 21.01$ |
| | PID | $72.30 \pm 24.96$ | $27.11 \pm 15.58$ | $49.02 \pm 18.75$ |
| | ADRC | $\mathbf{47.08 \pm 21.58}$ | $\mathbf{12.31 \pm 9.34}$ | $\mathbf{30.12 \pm 12.74}$ |
| RacecarPush | Lag | $57.91 \pm 22.12$ | $15.45 \pm 12.92$ | $35.54 \pm 15.56$ |
| | PID | $70.84 \pm 23.97$ | $28.16 \pm 16.84$ | $49.67 \pm 19.96$ |
| | ADRC | $\mathbf{47.28 \pm 17.05}$ | $\mathbf{12.50 \pm 7.05}$ | $\mathbf{29.35 \pm 11.03}$ |

As shown in Table 6, the ADRC methods significantly reduce violation rate, a smaller violation magnitude, indicating reduced oscillations and a shorter phase-lag in response. The calculation of metrics are detailed in Appendix E.3.

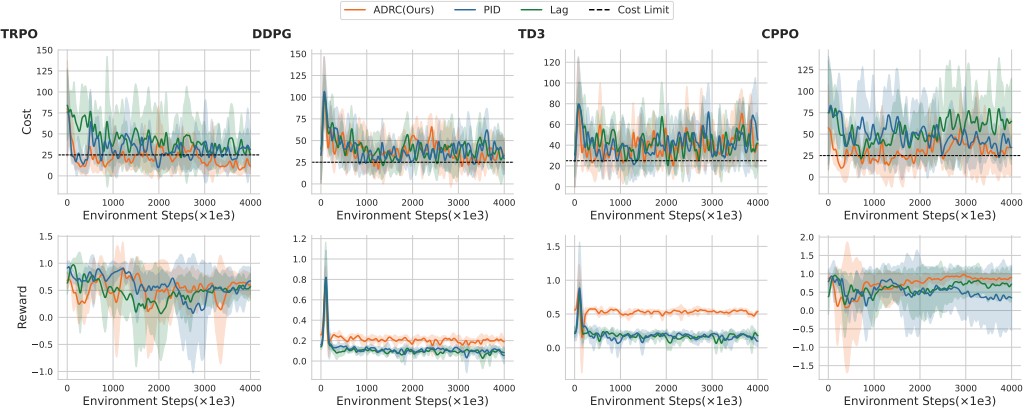

Figure 4: Training curves of Racecargoal task.

Figure 4 shows training curves for different constraint-handling methods (ADRC, PID, and Classical Lagrangian) across four RL algorithms (TRPO, DDPG, TD3, and PPO) in the RacecarGoal environment. Results show that ADRC consistently maintains cost below the limit while achieving competitive or superior rewards compared to other methods across different RL backbones.

## F.2 MAIN RESULTS

To ensure clarity and readability, we present the training curves for each environment separately, avoiding the complexity of overlaying multiple curves on a single plot. This approach allows for a more intuitive comparison of performance across different settings. For a comprehensive evaluation of our method's effectiveness, we conducted experiments across three agents—Ant, Racecar, and Car—and four reinforcement learning tasks: Goal, Circle, Button, and Push. This setup resulted in a total of 12 experimental groups. For each group, we ran experiments with 5 different random seeds to account for variability and ensure statistical robustness. Furthermore, we benchmarked our method against four widely used reinforcement learning algorithms: TRPO, PPO, DDPG, and TD3, covering both on-policy and off-policy approaches. This rigorous experimental design provides a thorough validation of our method's adaptability and performance across diverse scenarios.

### F.2.1 ANT ENVIRONMENTS

Figures 5 to Figure 8 present the training curves for the Ant environment across four tasks: Button, Circle, Goal, and Push. Each plot illustrates the episodic returns and costs averaged over five random seeds, with solid lines representing the mean and shaded areas denoting the variance.

To provide a more thorough and quantitative evaluation of our method, we report the results of experiments conducted on four challenging environments, AntButton, AntCircle, AntPush and AntGoal in Table 7 and Table 8. The metrics compared include violation rate (%), magnitude of violations, and average cost. Across all experiments, our ADRC method consistently outperforms baseline approaches (PID and Lagrange) in achieving lower violation rates and magnitudes, while maintaining competitive or reduced average costs. These results, validated across four RL algorithms (TRPO, PPO, DDPG, TD3), demonstrate the effectiveness and robustness of ADRC in handling constraint-aware reinforcement learning tasks.

### F.2.2 CAR ENVIRONMENTS

Figures 9 to Figure 12 present the training curves for the Car environment across four tasks: Button, Circle, Goal, and Push. Each plot illustrates the episodic returns and costs averaged over five random seeds, with solid lines representing the mean and shaded areas denoting the variance.

To provide a more thorough and quantitative evaluation of our method, we report the results of experiments conducted on four challenging environments, AntButton, AntCircle, AntPush and AntGoal in Table 9 and Table 10. The metrics compared include violation rate (%), magnitude of

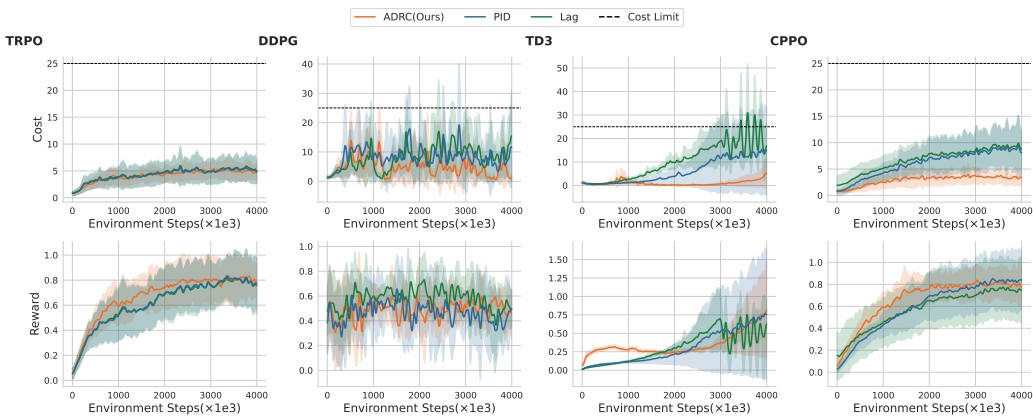

Figure 5: The training curves of AntButton with various Lagrangian methods across different algorithms.

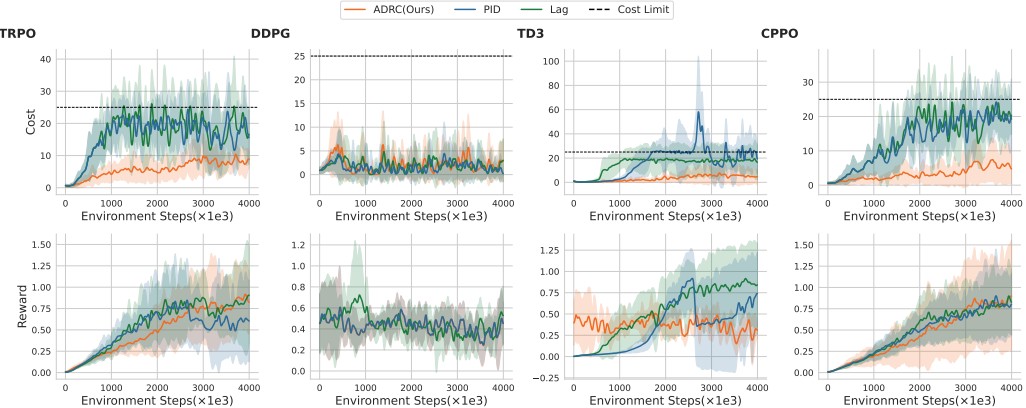

Figure 6: The training curves of AntCircle with various Lagrangian methods across different algorithms.

Table 7: Comparison of violation rate, magnitude, and average cost on AntButton and AntCircle.

| Algorithm | AntButton | | | AntCircle | | |
|---|---|---|---|---|---|---|
| | Vio. Rate (%) | Magnitude | Avg. Cost | Vio. Rate (%) | Magnitude | Avg. Cost |
| CPPOLag | 0.22 ± 0.03 | ± 0.83 | 6.09 ± 3.94 | 13.98 ± 3.11 | 6.59 ± 1.89 | 17.30 ± 3.00 |
| CPPOPID | 0.30 ± 0.28 | 0.01 ± 0.01 | 6.95 ± 2.11 | 10.82 ± 3.77 | 0.62 ± 0.36 | 12.95 ± 0.92 |
| CPPOADRC | **0.01 ± 0.01** | **0.00 ± 0.00** | **2.69 ± 0.72** | **0.00 ± 0.00** | **0.00 ± 0.00** | **3.23 ± 2.24** |
| DDPGLag | 5.72 ± 4.67 | 0.30 ± 0.24 | 7.35 ± 2.97 | 0.07 ± 0.15 | 0.00 ± 0.00 | 1.93 ± 0.68 |
| DDPGPID | 6.22 ± 7.10 | 0.47 ± 0.65 | 7.92 ± 3.86 | 0.04 ± 0.07 | 0.00 ± 0.00 | 1.59 ± 0.50 |
| DDPGADRC | **1.93 ± 1.15** | **0.12 ± 0.10** | **4.87 ± 0.64** | 0.15 ± 0.27 | 0.00 ± 0.00 | 2.15 ± 0.41 |
| TD3Lag | 3.31 ± 5.67 | 0.32 ± 0.64 | 6.22 ± 4.96 | 31.26 ± 2.86 | 1.76 ± 0.22 | 15.02 ± 0.99 |
| TD3PID | 2.24 ± 5.01 | 0.11 ± 0.25 | 3.14 ± 4.22 | 21.32 ± 11.16 | 3.61 ± 2.06 | 13.74 ± 4.75 |
| TD3ADRC | **0.00 ± 0.00** | **0.00 ± 0.00** | **0.86 ± 0.41** | **0.02 ± 0.02** | **0.00 ± 0.00** | **2.49 ± 2.61** |
| TRPOLag | 0.01 ± 0.02 | 0.00 ± 0.00 | 4.40 ± 1.48 | 20.80 ± 3.52 | 1.64 ± 0.37 | 16.58 ± 2.01 |
| TRPOPID | 0.01 ± 0.02 | 0.00 ± 0.00 | 4.29 ± 1.35 | 17.73 ± 2.21 | 1.06 ± 0.20 | 16.14 ± 1.31 |
| TRPOADRC | **0.00 ± 0.00** | **0.00 ± 0.00** | **4.15 ± 0.58** | **0.14 ± 0.28** | **0.00 ± 0.01** | **5.74 ± 1.93** |

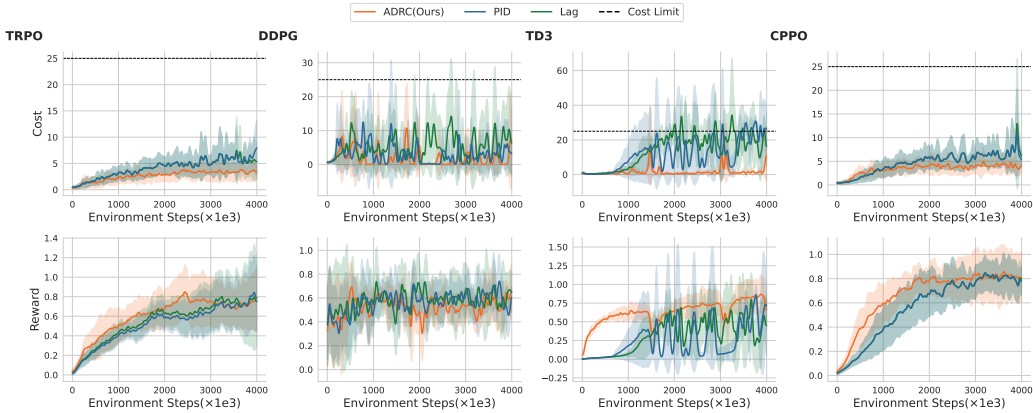

Figure 7: The training curves of AntGoal with various Lagrangian methods across different algorithms.

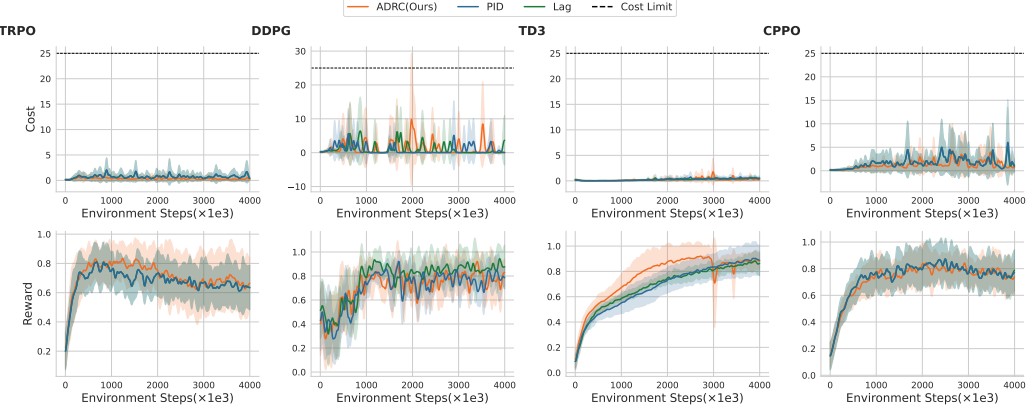

Figure 8: The training curves of AntPush with various Lagrangian methods across different algorithms.

Table 8: Comparison of violation rate, magnitude, and average cost on AntGoal and AntPush.

| Algorithm | AntGoal | | | AntPush | | |
|---|---|---|---|---|---|---|
| | Vio. Rate (%) | Magnitude | Avg. Cost | Vio. Rate (%) | Magnitude | Avg. Cost |
| CPPOLag | $0.41 \pm 0.66$ | $0.05 \pm 0.01$ | $4.68 \pm 1.05$ | $0.48 \pm 0.48$ | $3.65 \pm 3.93$ | $0.61 \pm 0.55$ |
| CPPOPID | $0.31 \pm 0.46$ | $0.03 \pm 0.06$ | $4.63 \pm 0.96$ | $0.36 \pm 0.54$ | $0.02 \pm 0.03$ | $1.70 \pm 0.96$ |
| CPPOADRC | $\mathbf{0.00 \pm 0.00}$ | $\mathbf{0.00 \pm 0.00}$ | $\mathbf{3.43 \pm 0.47}$ | $\mathbf{0.09 \pm 0.13}$ | $\mathbf{0.00 \pm 0.01}$ | $\mathbf{1.25 \pm 0.37}$ |
| DDPGLag | $3.09 \pm 1.37$ | $0.32 \pm 0.20$ | $5.34 \pm 1.18$ | $0.03 \pm 0.04$ | $0.00 \pm 0.00$ | $1.05 \pm 0.47$ |
| DDPGPID | $1.42 \pm 0.58$ | $0.19 \pm 0.07$ | $3.03 \pm 1.01$ | $0.17 \pm 0.24$ | $0.02 \pm 0.02$ | $\mathbf{0.82 \pm 0.51}$ |
| DDPGADRC | $\mathbf{1.19 \pm 1.27}$ | $\mathbf{0.15 \pm 0.15}$ | $\mathbf{1.88 \pm 1.19}$ | $0.67 \pm 1.19$ | $0.11 \pm 0.20$ | $1.50 \pm 0.80$ |
| TD3Lag | $20.21 \pm 4.27$ | $1.94 \pm 0.64$ | $13.46 \pm 1.39$ | $0.00 \pm 0.00$ | $0.00 \pm 0.00$ | $\mathbf{0.12 \pm 0.07}$ |
| TD3PID | $18.74 \pm 12.17$ | $2.00 \pm 1.75$ | $11.81 \pm 5.51$ | $0.00 \pm 0.00$ | $0.00 \pm 0.00$ | $0.16 \pm 0.14$ |
| TD3ADRC | $\mathbf{1.54 \pm 1.50}$ | $\mathbf{0.20 \pm 0.22}$ | $\mathbf{2.04 \pm 1.34}$ | $0.00 \pm 0.00$ | $0.00 \pm 0.00$ | $0.18 \pm 0.16$ |
| TRPOLag | $0.25 \pm 0.50$ | $0.02 \pm 0.04$ | $4.10 \pm 0.89$ | $0.00 \pm 0.00$ | $0.00 \pm 0.00$ | $0.73 \pm 0.26$ |
| TRPOPID | $0.15 \pm 0.30$ | $0.01 \pm 0.02$ | $4.13 \pm 0.94$ | $0.00 \pm 0.00$ | $0.00 \pm 0.00$ | $0.73 \pm 0.26$ |
| TRPOADRC | $\mathbf{0.00 \pm 0.00}$ | $\mathbf{0.00 \pm 0.00}$ | $\mathbf{2.70 \pm 0.69}$ | $0.00 \pm 0.00$ | $0.00 \pm 0.00$ | $\mathbf{0.34 \pm 0.15}$ |

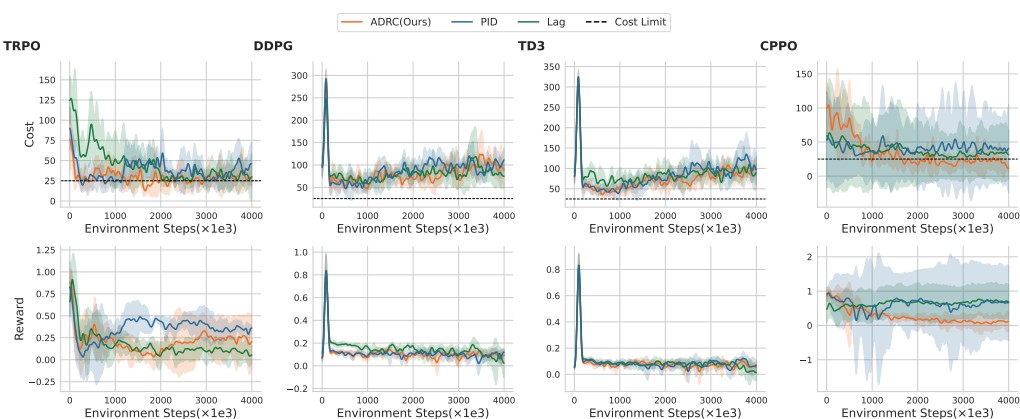

Figure 9: The training curves of CarButton with various Lagrangian methods across different algorithms.

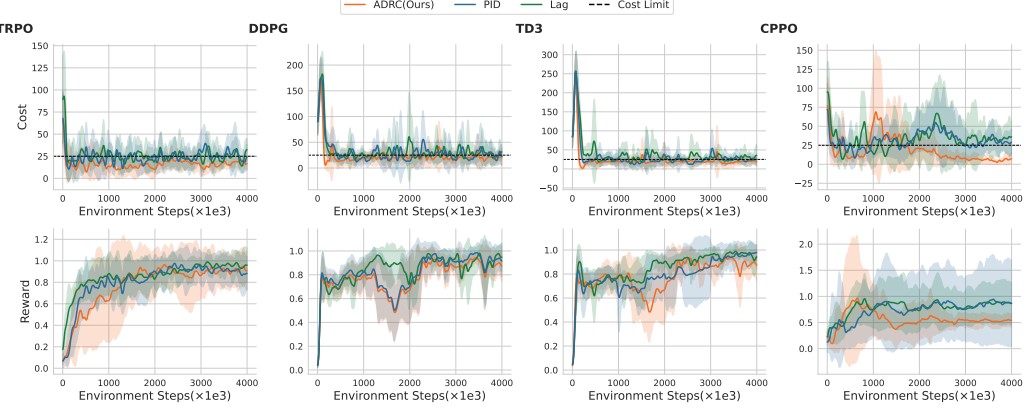

Figure 10: The training curves of CarCircle with various Lagrangian methods across different algorithms.

violations, and average cost. Across all experiments, our ADRC method consistently outperforms baseline approaches (PID and Lagrange) in achieving lower average cost and violation magnitudes, while maintaining competitive.

Table 9: Comparison of violation rate, magnitude, and average cost on CarButton and CarCircle.

| Algorithm | CarButton | | | CarCircle | | |
|---|---|---|---|---|---|---|
| | Vio. Rate (%) | Magnitude | Avg. Cost | Vio. Rate (%) | Magnitude | Avg. Cost |
| CPPOLag | 89.77 ± 19.38 | 59.77 ± 39.05 | 84.05 ± 40.18 | 46.73 ± 20.16 | 15.85 ± 17.40 | 33.54 ± 20.09 |
| CPPOPID | 85.09 ± 16.67 | 45.27 ± 46.80 | 68.95 ± 47.94 | 52.77 ± 17.62 | 12.29 ± 8.14 | 31.24 ± 10.31 |
| CPPOADRC | **50.16 ± 17.08** | **14.80 ± 3.96** | **34.20 ± 5.75** | **21.35 ± 13.09** | **7.74 ± 6.46** | **18.85 ± 9.30** |
| DDPGLag | 99.93 ± 0.11 | 58.18 ± 12.27 | 83.17 ± 12.27 | 51.85 ± 0.74 | 13.55 ± 2.86 | 33.49 ± 2.45 |
| DDPGPID | 98.87 ± 2.05 | 64.10 ± 6.43 | 89.05 ± 6.51 | 39.77 ± 4.31 | 13.56 ± 1.49 | 30.89 ± 1.44 |
| DDPGADRC | 99.49 ± 0.08 | **58.62 ± 7.25** | **83.60 ± 7.25** | 51.50 ± 5.10 | **7.73 ± 1.79** | **23.82 ± 1.33** |
| TD3Lag | 99.97 ± 0.04 | 62.20 ± 10.10 | 87.20 ± 10.11 | 53.00 ± 1.06 | 17.18 ± 1.25 | 38.06 ± 1.57 |
| TD3PID | 99.03 ± 1.00 | 59.44 ± 11.92 | 84.40 ± 11.95 | 39.20 ± 1.56 | 12.52 ± 1.64 | 30.87 ± 1.18 |
| TD3ADRC | **99.04 ± 0.73** | **50.84 ± 9.18** | **75.80 ± 9.21** | 48.41 ± 6.50 | **7.49 ± 1.54** | **24.26 ± 1.28** |
| TRPOLag | 74.24 ± 11.05 | 21.90 ± 7.38 | 44.84 ± 8.09 | 40.04 ± 1.30 | 6.45 ± 1.28 | 25.52 ± 0.51 |
| TRPOPID | 69.04 ± 17.53 | 14.60 ± 4.66 | 37.11 ± 6.10 | 40.67 ± 3.51 | 6.58 ± 1.15 | 24.24 ± 1.44 |
| TRPOADRC | **53.28 ± 15.44** | **8.53 ± 4.33** | **29.57 ± 6.18** | **17.71 ± 2.92** | **1.92 ± 0.64** | **16.22 ± 1.75** |

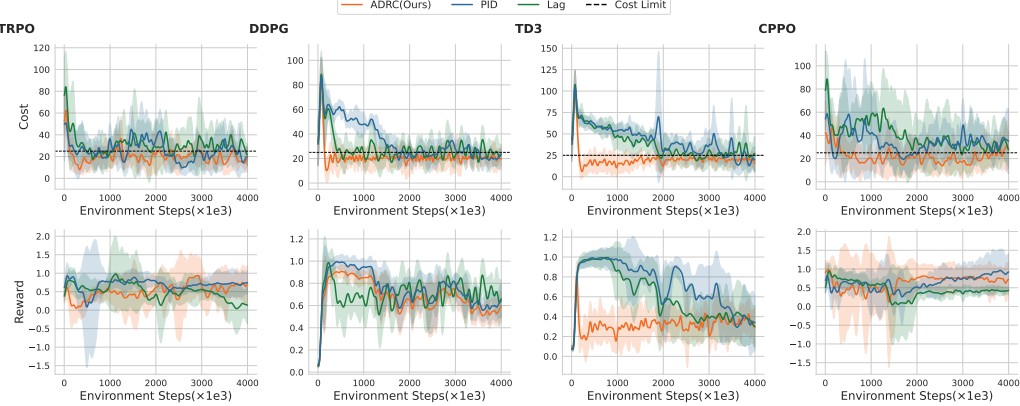

Figure 11: The training curves of CarGoal with various Lagrangian methods across different algorithms.

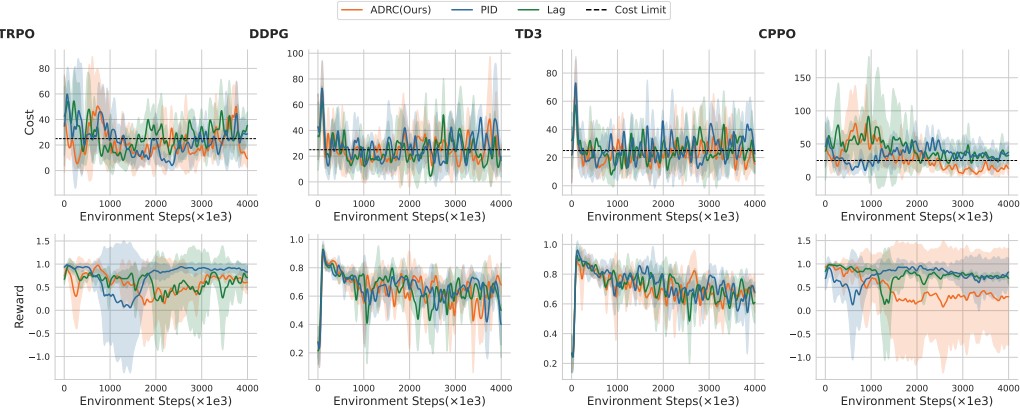

Figure 12: The training curves of CarPush with various Lagrangian methods across different algorithms

Table 10: Comparison of violation rate, magnitude, and average cost on CarGoal and CarPush.

| Algorithm | CarGoal | | | CarPush | | |
|---|---|---|---|---|---|---|
| | Vio. Rate (%) | Magnitude | Avg. Cost | Vio. Rate (%) | Magnitude | Avg. Cost |
| CPPOLag | 65.20 ± 22.57 | 18.54 ± 11.41 | 40.05 ± 14.00 | 68.45 ± 18.98 | 21.48 ± 16.81 | 43.38 ± 18.71 |
| CPPOPID | 62.04 ± 18.18 | 12.95 ± 7.45 | 34.41 ± 9.04 | 62.36 ± 10.66 | 12.40 ± 3.08 | 33.74 ± 3.88 |
| CPPOADRC | **34.97 ± 13.88** | **4.72 ± 2.14** | **21.99 ± 5.32** | **42.23 ± 15.34** | **11.68 ± 7.99** | **29.16 ± 10.19** |
| DDPGLag | 52.43 ± 0.12 | 7.25 ± 0.93 | 28.35 ± 0.44 | 48.81 ± 1.62 | 8.20 ± 1.38 | 27.37 ± 1.19 |
| DDPGPID | 65.52 ± 4.60 | 12.53 ± 0.72 | 35.03 ± 0.53 | 48.81 ± 1.62 | 8.20 ± 1.38 | 27.37 ± 1.19 |
| DDPGADRC | **47.36 ± 1.90** | **2.88 ± 0.57** | **21.55 ± 0.29** | **42.43 ± 9.36** | **7.07 ± 1.69** | **25.70 ± 2.85** |
| TD3Lag | 70.47 ± 10.44 | 17.00 ± 1.29 | 38.90 ± 2.83 | 46.97 ± 2.20 | 6.27 ± 1.53 | 25.51 ± 1.36 |
| TD3PID | 80.94 ± 5.87 | 20.68 ± 3.94 | 43.43 ± 4.23 | 49.83 ± 0.77 | 7.65 ± 1.15 | 26.66 ± 1.14 |
| TD3ADRC | **40.62 ± 8.51** | **2.85 ± 0.31** | **20.65 ± 2.46** | **39.92 ± 1.66** | **5.15 ± 0.82** | **23.64 ± 0.93** |
| TRPOLag | 54.86 ± 6.74 | 10.46 ± 4.56 | 30.97 ± 4.69 | 48.24 ± 6.24 | 8.51 ± 2.35 | 27.58 ± 3.16 |
| TRPOPID | 44.79 ± 2.84 | 7.34 ± 1.31 | 25.84 ± 0.88 | 40.05 ± 1.69 | 6.66 ± 1.82 | 23.92 ± 1.74 |
| TRPOADRC | **29.12 ± 3.70** | **3.44 ± 1.21** | **20.48 ± 0.99** | **34.75 ± 8.43** | **6.32 ± 2.13** | **22.40 ± 2.34** |

### F.2.3 RACECAR ENVIRONMENTS

Figures 13 to Figure 16 present the training curves for the Ant environment across four tasks: Button, Circle, Goal, and Push. Each plot illustrates the episodic returns and costs averaged over five random seeds, with solid lines representing the mean and shaded areas denoting the variance.

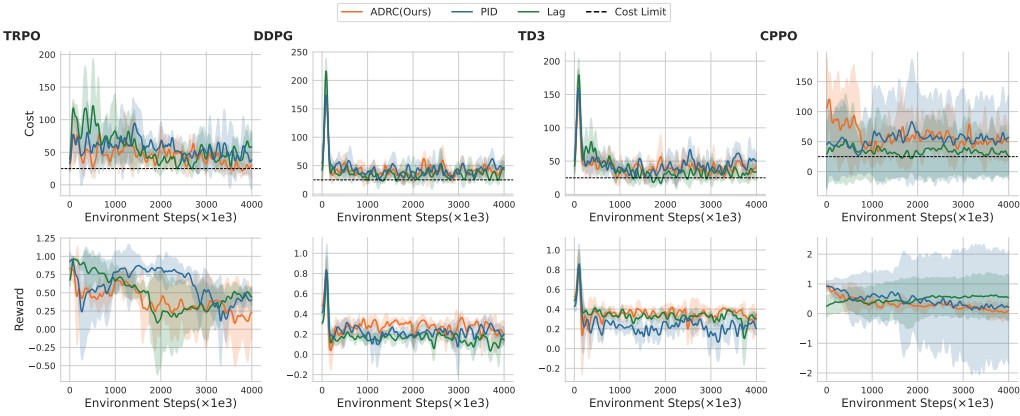

Figure 13: The training curves of RacecarButton with various Lagrangian methods across different algorithms.

To provide a more thorough and quantitative evaluation of our method, we report the results of experiments conducted on four challenging environments, AntButton, AntCircle, AntPush and AntGoal in Table 11 and Table 12. The metrics compared include violation rate (%), magnitude of violations, and average cost. Across all experiments, our ADRC method consistently outperforms baseline approaches (PID and Lagrange) in achieving lower violation rates and magnitudes, while maintaining competitive or reduced average costs.

### F.3 VELOCITY CONTROL RESULTS

To further evaluate our method's performance in dynamic and velocity-sensitive environments, we conducted experiments on the Safety Velocity Control tasks, including SafetySwimmer and SafetyHopper. These tasks pose additional challenges by requiring agents to manage both positional constraints and velocity profiles.

The following tables present the violation rates, violation magnitudes, average costs, and average rewards achieved by different methods. Across all settings, our ADRC Lagrangian method consistently

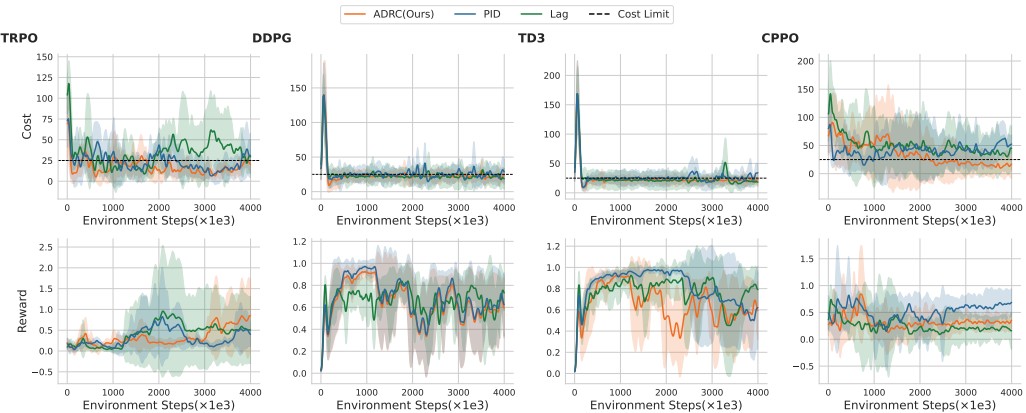

Figure 14: The training curves of RacecarCircle with various Lagrangian methods across different algorithms.

Table 11: Comparison of violation rate, magnitude, and average cost on RacecarButton and Racecar-Circle.

| Algorithm | RacecarButton | | | RacecarCircle | | |
|---|---|---|---|---|---|---|
| | Vio. Rate (%) | Magnitude | Avg. Cost | Vio. Rate (%) | Magnitude | Avg. Cost |
| CPPOLag | 97.38 ± 2.85 | 65.75 ± 16.98 | 90.53 ± 17.10 | 58.32 ± 31.47 | 30.72 ± 37.77 | 50.41 ± 41.68 |
| CPPOPID | 97.37 ± 4.24 | 78.44 ± 43.36 | 103.21 ± 43.64 | 56.82 ± 31.13 | 21.77 ± 25.28 | 40.95 ± 29.63 |
| CPPOADRC | **81.18 ± 20.97** | **35.76 ± 29.48** | **59.11 ± 31.27** | **39.94 ± 38.28** | **21.78 ± 31.79** | **35.72 ± 39.00** |
| DDPGLag | 75.88 ± 4.41 | 15.77 ± 3.10 | 39.47 ± 3.34 | 50.52 ± 0.25 | 7.87 ± 0.74 | 25.03 ± 0.67 |
| DDPGPID | 88.26 ± 2.48 | 20.12 ± 2.35 | 44.62 ± 2.47 | 46.17 ± 1.40 | 7.94 ± 1.09 | 27.09 ± 0.56 |
| DDPGADRC | 85.18 ± 6.77 | **18.49 ± 2.43** | **42.87 ± 2.74** | 50.92 ± 2.27 | **5.00 ± 0.79** | **24.51 ± 0.33** |
| TD3Lag | 72.49 ± 2.60 | 16.31 ± 2.92 | 39.65 ± 3.12 | 49.42 ± 0.19 | 8.22 ± 0.52 | 24.95 ± 0.52 |
| TD3PID | 85.99 ± 7.60 | 21.46 ± 4.96 | 45.75 ± 5.43 | 48.72 ± 1.24 | 8.57 ± 1.07 | 27.76 ± 1.30 |
| TD3ADRC | 84.48 ± 4.33 | **18.37 ± 2.74** | **42.65 ± 2.93** | **42.08 ± 1.39** | **4.56 ± 0.79** | **24.21 ± 0.82** |
| TRPOLag | 87.31 ± 4.65 | 33.07 ± 7.84 | 57.34 ± 7.59 | 51.39 ± 10.47 | 16.10 ± 10.65 | 34.48 ± 11.14 |
| TRPOPID | 87.04 ± 3.69 | 32.94 ± 3.94 | 56.86 ± 3.65 | 37.99 ± 3.47 | 7.92 ± 2.78 | 23.45 ± 1.84 |
| TRPOADRC | **80.06 ± 4.19** | **21.38 ± 5.00** | **45.00 ± 5.21** | **23.64 ± 3.83** | **3.54 ± 0.43** | **17.30 ± 1.32** |

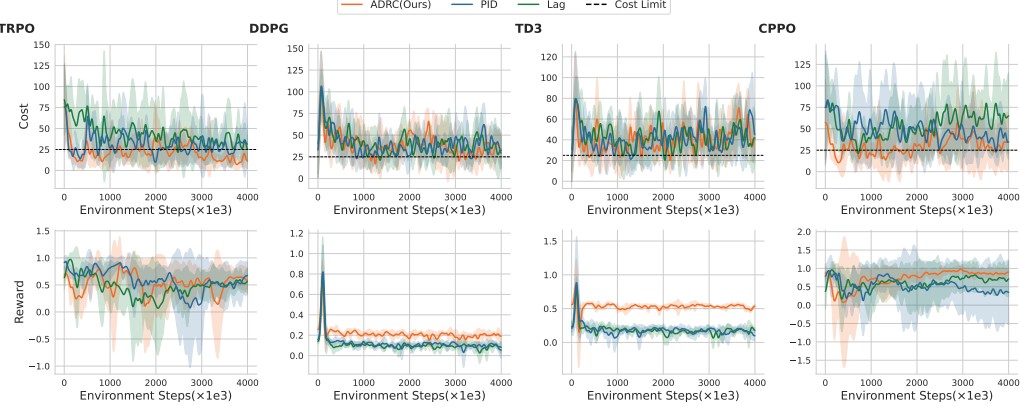

Figure 15: The training curves of RacecarGoal with various Lagrangian methods across different algorithms.

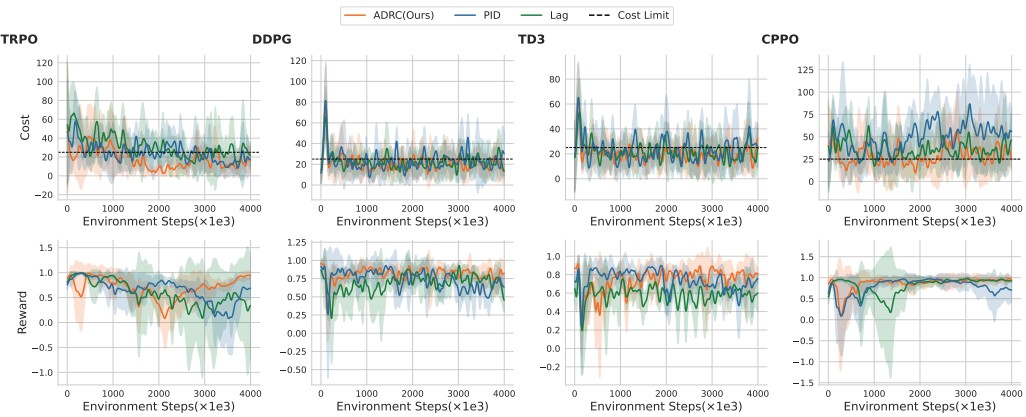

Figure 16: The training curves of RacecarPush with various Lagrangian methods across different algorithms.

Table 12: Comparison of violation rate, magnitude, and average cost on RacecarGoal and Racecar-Push.

| Algorithm | RacecarGoal | | | RacecarPush | | |
|---|---|---|---|---|---|---|
| | Vio. Rate (%) | Magnitude | Avg. Cost | Vio. Rate (%) | Magnitude | Avg. Cost |
| CPPOLag | 80.87 ± 19.17 | 31.18 ± 18.99 | 54.24 ± 21.01 | 57.91 ± 22.12 | 15.45 ± 12.92 | 35.54 ± 15.56 |
| CPPOPID | 72.30 ± 24.96 | 27.11 ± 15.58 | 49.02 ± 18.75 | 70.84 ± 23.97 | 28.16 ± 16.84 | 49.67 ± 19.96 |
| CPPOADRC | **47.08 ± 21.59** | **12.31 ± 9.34** | **30.12 ± 12.74** | **47.28 ± 17.05** | **12.50 ± 7.05** | **29.35 ± 11.03** |
| DDPGLag | 71.44 ± 9.47 | 18.47 ± 4.13 | 40.76 ± 5.00 | 39.58 ± 2.98 | 5.00 ± 0.49 | 22.52 ± 0.58 |
| DDPGPID | 72.05 ± 4.09 | 17.93 ± 2.81 | 40.29 ± 2.99 | 39.31 ± 1.84 | 6.89 ± 0.81 | 23.46 ± 0.98 |
| DDPGADRC | **68.41 ± 5.77** | **17.81 ± 1.34** | **39.41 ± 2.14** | **38.08 ± 3.96** | **5.10 ± 0.98** | **21.99 ± 1.13** |
| TD3Lag | 75.26 ± 2.13 | 19.93 ± 1.40 | 42.57 ± 1.59 | 36.76 ± 1.53 | 5.25 ± 0.92 | 21.64 ± 0.97 |
| TD3PID | 73.31 ± 6.06 | 19.19 ± 2.72 | 41.57 ± 3.22 | 38.87 ± 1.99 | 6.76 ± 1.65 | 23.32 ± 2.45 |
| TD3ADRC | **71.24 ± 3.00** | **17.55 ± 3.57** | **39.71 ± 3.94** | **36.21 ± 3.11** | **4.25 ± 1.50** | **20.70 ± 1.49** |
| TRPOLag | 64.31 ± 23.37 | 22.89 ± 18.69 | 43.67 ± 21.56 | 51.18 ± 11.61 | 12.82 ± 3.76 | 31.19 ± 5.63 |
| TRPOPID | 49.33 ± 15.00 | 11.70 ± 7.07 | 30.94 ± 8.81 | 40.06 ± 1.74 | 7.98 ± 1.92 | 24.46 ± 1.55 |
| TRPOADRC | **34.03 ± 8.06** | **6.16 ± 2.37** | **22.02 ± 3.61** | **26.06 ± 11.09** | **5.41 ± 2.66** | **18.28 ± 5.41** |

demonstrates superior safety performance with competitive or improved final rewards compared to baseline PID Lagrangian methods.

Table 13: Performance comparison on SafetySwimmer environment.

| Algorithm | Violate Rate (%) | Magnitude | Avg Cost | Avg Reward |
|---|---|---|---|---|
| CPPOPID | 28.33 | 1.84 | 23.64 | 32.54 |
| CPPOADRC | **6.95** | **1.56** | **18.34** | 29.07 |
| TRPOPID | 35.43 | 1.78 | 22.48 | 27.73 |
| TRPOADRC | **11.30** | 2.44 | 20.82 | **35.66** |

Table 14: Performance comparison on SafetyHopper environment.

| Algorithm | Violate Rate (%) | Magnitude | Avg Cost | Avg Reward |
|---|---|---|---|---|
| CPPOPID | 40.33 | 5.32 | 23.92 | **1365.60** |
| CPPOADRC | **17.40** | **1.84** | **17.35** | 1155.59 |
| TRPOPID | 39.33 | 7.52 | 24.41 | **1448.46** |
| TRPOADRC | **0.93** | **0.06** | **12.02** | 1080.80 |

These results validate that the ADRC-based methods significantly improve safety metrics (lower violation rate and cost) while maintaining comparable or strong reward performance in dynamic velocity control tasks. This further demonstrates the effectiveness and robustness of ADRC Lagrangian formulations under more complex and realistic settings.

### F.4 COMPARISON WITH STATE-OF-THE-ART SAFE RL ALGORITHMS

To demonstrate the broader applicability and effectiveness of our ADRC-Lagrangian framework beyond traditional Lagrangian methods, we conduct comprehensive comparisons with state-of-the-art safe RL algorithms. Our evaluation includes both Lagrangian-based methods (RCPO and PDO) Tessler et al. (2018); Chow et al. (2018a) and non-Lagrangian approaches such as CUP Yang et al. (2022) and IPO Liu et al. (2019). This comparison validates our method's superiority across different safe RL paradigms and confirms that the benefits stem from ADRC's adaptive control principles rather than merely being artifacts of the Lagrangian framework.

We evaluate all methods on two challenging continuous control environments: HalfCheetah-Velocity and Hopper-Velocity from the Safety-Gymnasium benchmark. Each algorithm is trained with identical hyperparameters and evaluated using three random seeds. We report both training metrics (averaged over the entire training process) and final policy evaluation results to provide comprehensive performance assessment.

Table 15: Training performance on HalfCheetah-Velocity. Best results in **bold**, runner-up in underline.

| Algorithm | Vio. Rate (%) | Magnitude | Avg. Cost | Avg. Reward |
|---|---|---|---|---|
| CUP | 22.63±7.21 | 4.48±4.58 | 16.25±5.08 | 1532.23±255.05 |
| IPO | 29.17±1.63 | 0.81±0.02 | 19.60±0.08 | 1460.56±210.91 |
| PDO | 31.95±5.09 | 9.68±2.84 | 22.16±1.47 | 1690.62±421.72 |
| RCPO | 18.26±9.69 | 4.75±3.71 | 15.77±7.03 | 1497.99±410.94 |
| RCPO-ADRC | **0.00±0.00** | **0.00±0.00** | **8.40±2.40** | 1329.62±293.58 |
| TRPO-ADRC | 1.19±0.25 | 0.09±0.10 | 10.89±0.34 | **1743.09±295.33** |
| CPPO-ADRC | 8.53±12.06 | 0.55±0.78 | 15.36±2.05 | 1504.17±198.53 |

Tables 15 and 16 show that our ADRC variants markedly enhance *training-time stability*: compared with existing safe RL methods, ADRC achieves consistently lower violation rates, smaller violation magnitudes, and reduced average costs. For example, on HalfCheetah, RCPO-ADRC eliminates violations entirely (0.00±0.00% vs. 18.26±9.69% for RCPO) and attains the lowest training cost (8.40±2.40); on Hopper, RCPO-ADRC sharply suppresses violations (2.57±2.19%) with the smallest

Table 16: Training performance on Hopper-Velocity. Best results in **bold**, runner-up in underline.

| Algorithm | Vio. Rate (%) | Magnitude | Avg. Cost | Avg. Reward |
|---|---|---|---|---|
| CUP | 37.11±3.59 | 5.47±0.59 | 21.73±1.03 | 1085.14±204.86 |
| IPO | 51.99±8.89 | 1.68±0.18 | 24.70±0.84 | 1082.95±103.43 |
| PDO | 27.56±9.25 | 8.42±3.48 | 20.00±7.01 | 1098.22±178.78 |
| RCPO | 37.57±7.00 | 5.45±0.93 | 23.24±2.72 | **1247.72±292.68** |
| RCPO-ADRC | **2.57±2.19** | **0.06±0.05** | **14.23±2.74** | 1186.87±70.94 |
| TRPO-ADRC | 7.76±9.59 | 0.33±0.40 | 15.13±3.04 | 1167.62±90.74 |
| CPPO-ADRC | 11.01±5.65 | 0.92±0.67 | 15.43±1.40 | 1083.94±68.49 |

Table 17: Evaluation performance on HalfCheetah-Velocity. Best results in **bold**, runner-up in underline.

| Algorithm | Reward | Cost | Length |
|---|---|---|---|
| CUP | 2175.61±491.09 | 28.57±21.15 | 1000.00±0.00 |
| IPO | 1819.75±292.45 | 16.10±12.99 | 1000.00±0.00 |
| PDO | **2468.78±581.21** | **5.77±7.32** | 1000.00±0.00 |
| RCPO | 2296.82±665.64 | 15.00±8.81 | 1000.00±0.00 |
| RCPO-ADRC | 1642.22±211.39 | 10.23±3.51 | 1000.00±0.00 |
| TRPO-ADRC | 2394.09±419.84 | 14.63±15.65 | 1000.00±0.00 |
| CPPO-ADRC | 2098.71±464.92 | 13.87±11.43 | 1000.00±0.00 |

Table 18: Evaluation performance on Hopper-Velocity. Best results in **bold**, runner-up in underline.

| Algorithm | Reward | Cost | Length |
|---|---|---|---|
| CUP | 1326.84±386.22 | 26.90±10.59 | 854.40±165.37 |
| IPO | 1216.01±129.12 | 27.57±4.95 | 797.33±118.41 |
| PDO | 1177.19±135.02 | 18.03±25.50 | 797.57±167.85 |
| RCPO | **1554.56±223.49** | 37.53±24.35 | 979.30±29.27 |
| RCPO-ADRC | 1248.94±220.47 | **9.27±5.18** | 818.10±128.67 |
| TRPO-ADRC | 1470.61±152.19 | 10.67±3.76 | **1000.00±0.00** |
| CPPO-ADRC | 1322.83±157.06 | 10.43±7.12 | 910.20±127.00 |

magnitudes (0.06±0.05) and cost (14.23±2.74). Crucially, this improved safety does *not* come at the expense of learning quality: ADRC maintains competitive training rewards—and can be better—e.g., TRPO-ADRC attains the highest training reward on HalfCheetah (1743.09±295.33) with only 1.19±0.25% violations, indicating stable and efficient optimization.

Tables 17 and 18 further examine *convergence-time* performance (evaluation). Even without explicitly measuring constraints at evaluation, ADRC remains competitive—or superior—on task metrics: on HalfCheetah, TRPO-ADRC reaches runner-up reward (2394.09±419.84), close to the best; on Hopper, RCPO-ADRC achieves the lowest evaluation cost (9.27±5.18) and TRPO-ADRC sustains the maximum horizon (1000.00±0.00) with strong reward (1470.61±152.19). Together, these results confirm that ADRC improves training stability and safety while preserving (and in cases improving) final task performance and convergence behavior, offering a plug-and-play safety enhancement over existing safe RL baselines.

### F.5    PARAMETER SENSITIVITY ANALYSIS

#### F.5.1    TUNING PARAMETER $k_{ap}$

To assess the effect of the control gain $k_{ap}$ on the performance of ADRC-based Lagrangian methods, we conducted a series of ablation experiments. Specifically, we evaluated three distinct values of $k_{ad}$ $(0.01, 0.1, 1)$ and compared them with existing approaches, including PID-based and classical Lagrangian methods. These experiments were carried out in two challenging environments, *CarPush* and *RacecarGoal*, using two reinforcement learning algorithms, *PPO* and *TRPO*. The results highlight ADRC's ability to dynamically adjust the control gain, demonstrating superior adaptability and improved performance with carefully selected parameter settings.

Table 19: The proportion of constraint violations during training (Vio. Rate), the average magnitude of violations (Magnitude), and the average cost (Avg. Cost) for TRPO and PPO algorithms across CarPush and RacecarGoal environments with various $k_{ap}$ values, PID, and Lag methods. Bold values indicate better performance compared to PID.

| Algorithm | Method | CarPush | | | RacecarGoal | | |
|---|---|---|---|---|---|---|---|
| | | Vio. Rate (%) | Magnitude | Avg. Cost | Vio. Rate (%) | Magnitude | Avg. Cost |
| TRPO | $k_{ap} = 1$ | **36.38** | **4.51** | **21.99** | **32.73** | **5.78** | **22.27** |
| | $k_{ap} = 0.1$ | **33.08** | 5.78 | **21.22** | **29.05** | **3.44** | **18.95** |
| | $k_{ap} = 0.01$ | **20.43** | **2.50** | **15.42** | **23.83** | **4.29** | **19.36** |
| | PID | 38.40 | 4.84 | 21.96 | 44.60 | 7.04 | 26.15 |
| | Lag | 39.88 | 5.38 | 23.31 | 87.33 | 37.36 | 61.53 |
| CPPO | $k_{ap} = 1$ | 86.08 | 24.38 | 48.31 | **69.98** | **18.62** | **39.87** |
| | $k_{ap} = 0.1$ | **16.25** | **4.05** | **13.46** | **29.05** | **3.44** | **18.95** |
| | $k_{ap} = 0.01$ | **42.83** | **5.75** | **23.56** | **23.83** | **4.29** | **19.36** |
| | PID | 67.28 | 12.80 | 34.67 | 79.25 | 23.88 | 46.44 |
| | Lag | 46.43 | 6.67 | 25.90 | 84.35 | 30.16 | 53.38 |

As shown in Table 19, we report the **Violation Rate (Vio. Rate)**, the **Magnitude** of constraint violations, and the **Average Cost (Avg. Cost)** for both the *CarPush* and *RacecarGoal* environments using the *TRPO* and *PPO* algorithms. The results demonstrate the superior performance of our ADRC approach with varying $k_{ap}$ values compared to baseline methods (PID and Lag). Specifically:

- For the **CarPush** environment:
  - Under **TRPO**, the configuration $k_{ap} = 0.01$ achieves the lowest violation rate (20.43%) and magnitude (2.50), alongside a significantly reduced average cost (15.42), outperforming both PID and Lag.
  - For **CPPO**, $k_{ap} = 0.1$ shows remarkable results, with a violation rate of 16.25%, the smallest magnitude (4.05), and the lowest average cost (13.46). This highlights the adaptability of ADRC at this gain level.
- For the **RacecarGoal** environment:
  - With **TRPO**, $k_{ap} = 0.01$ again demonstrates the best performance, achieving a violation rate of 23.83%, a moderate magnitude (4.29), and a reduced average cost (19.36). This represents a clear improvement over both PID and Lag methods.

- Similarly, under **CPPO**, $k_{ap} = 0.1$ achieves the best performance with a violation rate of 29.05%, the smallest magnitude (3.44), and the lowest average cost (18.95). These results further emphasize ADRC's effectiveness.

- For both environments, the baseline PID and Lag methods generally exhibit higher violation rates, magnitudes, and costs. Lag in particular performs poorly, especially in the *RacecarGoal* environment, where it yields the highest violation rates and costs.

These results confirm that our ADRC method, with its dynamic control gain adjustments, consistently outperforms traditional methods, particularly when $k_{ap} = 0.01$ or $k_{ap} = 0.1$, demonstrating its robustness and adaptability across diverse environments and algorithms.

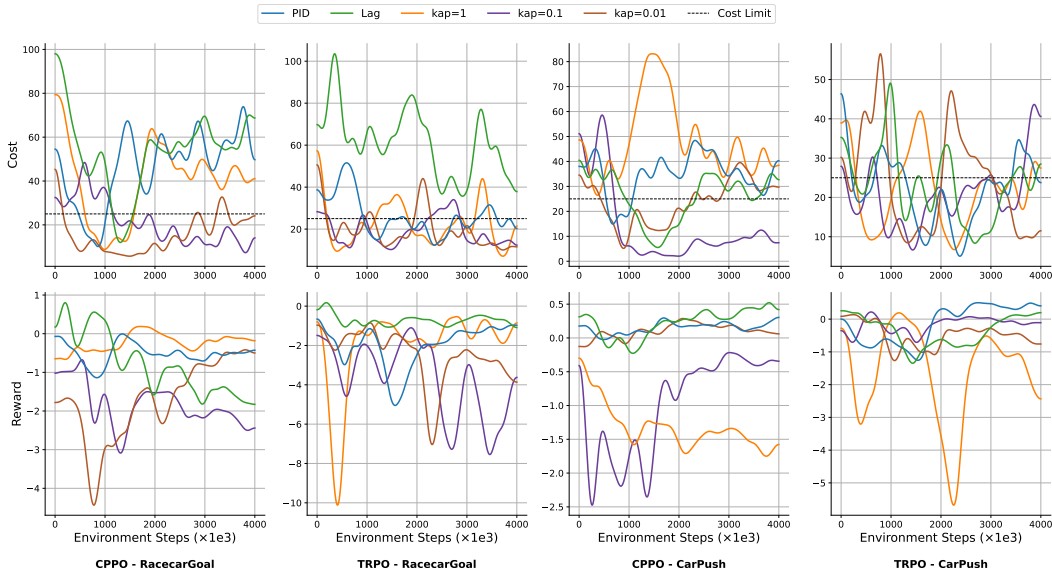

Figure 17: The training curve for TRPO and CPPO algorithms across CarPush and RacecarGoal environments with various $k_{ap}$ values, PID, and Lag methods.

Figure 19 provides the training curves for reward and cost across the evaluated $k_{ap}$ values, PID, and Lag methods. These curves illustrate the consistent performance improvements of our method throughout the training process. Our approach not only converges more effectively but also demonstrates a more favorable trade-off between reward maximization and cost minimization.

### F.5.2 TUNING PARAMETER $k_{ad}$

To assess the effect of the control gain $k_{ad}$ on the performance of ADRC-based Lagrangian methods, we conducted a series of ablation experiments. Specifically, we evaluated three distinct values of $k_{ad}$ (0.01, 0.1, 1) and compared them with existing approaches, including PID-based and classical Lagrangian methods. These experiments were carried out in two challenging environments, *CarPush* and *RacecarGoal*, using two reinforcement learning algorithms, *CPPO* and *TRPO*. The results highlight ADRC's ability to dynamically adjust the control gain, demonstrating superior adaptability and improved performance with carefully selected parameter settings. To evaluate the impact of the tuning parameter $k_{ad}$ on ADRC Lagrangian methods' performance, we conducted ablation experiments by selecting three different values of $k_{ad} = 0.01, 0.1, 1$ and comparing them against existing methods, including PID Lagrangian methods and classical Lagrangian methods. The experiments were performed across two environments which are CarPush and RacecarGoal and adopt two algorithms which are CPPO and TRPO.

As shown in Table 3, we report the **Violation Rate (Vio. Rate)**, the **Magnitude** of constraint violations, and the **Average Cost (Avg. Cost)** for both the *CarPush* and *RacecarGoal* environments using the *TRPO* and *CPPO* algorithms. The results highlight the superior performance of our ADRC approach with varying $k_{ad}$ values compared to the baseline methods (PID and Lag). Specifically:

Table 20: The proportion of constraint violations during training (Vio. Rate), the average magnitude of violations (Magnitude), and the average cost (Avg. Cost) for TRPO and CPPO algorithms across CarPush and RacecarGoal environments with various $k_{ad}$ values, PID, and Lag methods. Bold values indicate better performance compared to PID.

| Algorithm | Method | CarPush | | | RacecarGoal | | |
|---|---|---|---|---|---|---|---|
| | | Vio. Rate (%) | Magnitude | Avg. Cost | Vio. Rate (%) | Magnitude | Avg. Cost |
| TRPO | $k_{ad} = 1$ | **38.23** | 6.16 | 23.11 | **28.55** | **6.17** | **20.23** |
| | $k_{ad} = 0.1$ | **36.68** | 6.29 | 21.99 | **38.25** | 9.12 | **23.92** |
| | $k_{ad} = 0.01$ | **30.20** | **3.86** | **20.36** | **39.08** | **5.75** | **23.33** |
| | PID | 38.40 | 4.84 | 21.96 | 44.60 | 7.04 | 26.15 |
| | Lag | 39.88 | 5.38 | 23.31 | 87.33 | 37.36 | 61.53 |
| CPPO | $k_{ad} = 1$ | **16.20** | **1.78** | **16.60** | **48.68** | 9.12 | **27.80** |
| | $k_{ad} = 0.1$ | **15.08** | **5.06** | **15.45** | **48.55** | 8.59 | **27.80** |
| | $k_{ad} = 0.01$ | **16.25** | **4.05** | **13.46** | **33.08** | **5.75** | **23.33** |
| | PID | 67.28 | 12.80 | 34.67 | 79.25 | 23.88 | 46.44 |
| | Lag | 46.43 | 6.67 | 25.90 | 84.35 | 30.16 | 53.38 |

- For the **CarPush** environment:
    - Under **TRPO**, $k_{ad} = 0.01$ achieves the lowest violation rate (30.20%) and the smallest magnitude (3.86), alongside a reduced average cost (20.36). This indicates better constraint satisfaction and efficiency compared to PID and Lag.
    - For **CPPO**, $k_{ad} = 0.1$ yields the best performance with the lowest violation rate (15.08%), a moderate magnitude (5.06), and the smallest average cost (15.45). These results highlight ADRC's adaptability at this parameter setting.
- For the **RacecarGoal** environment:
    - With **TRPO**, $k_{ad} = 1$ shows excellent performance, achieving the lowest violation rate (28.55%) and average cost (20.23), alongside a relatively small magnitude (6.17).
    - Under **CPPO**, $k_{ad} = 0.01$ demonstrates the best results, with a low violation rate (33.08%), a reduced magnitude (5.75), and the smallest average cost (23.33). This showcases ADRC's ability to manage constraints effectively in this challenging environment.
- Across both environments, the baseline methods (PID and Lag) consistently exhibit higher violation rates, larger magnitudes, and higher costs. Lag performs particularly poorly, with significantly higher metrics, especially in the *RacecarGoal* environment, where it records the highest violation rate (87.33%) and average cost (61.53).

### F.5.3 TUNING PARAMETER $c_r$

To evaluate the impact of the tuning parameter $c_r$ on ADRC Lagrangian methods' performance, we conducted ablation experiments by selecting five different values of $c_r = 0.05, 0.1, 0.15, 0.2, 0.25$ and comparing them against existing methods, including PID Lagrangian methods and classical Lagrangian methods. The experiments were performed across two environments which are CarPush and RacecarGoal and adopt two algorithms which are CPPO and TRPO.

As shown in Table 21, we report the **Violation Rate (Vio. Rate)**, the **Magnitude** of constraint violations, and the **Average Cost (Avg. Cost)**. The results demonstrate that our method consistently outperforms the baseline methods (PID Lagrangian methods and classical Lagrangian methods) across a majority of the $c_r$ values. Specifically:

- For the CarPush environment, our method achieves lower violation rates and magnitudes in most cases, while maintaining competitive average costs.
- Similarly, in the RacecarGoal environment, our approach demonstrates significant improvements, particularly with $c_r = 0.1$ and $c_r = 0.2$, where it achieves the lowest violation rates and magnitudes.

Figure 19 provides the training curves for reward and cost across the evaluated $c_r$ values, PID, and Lag methods. These curves illustrate the consistent performance improvements of our method throughout

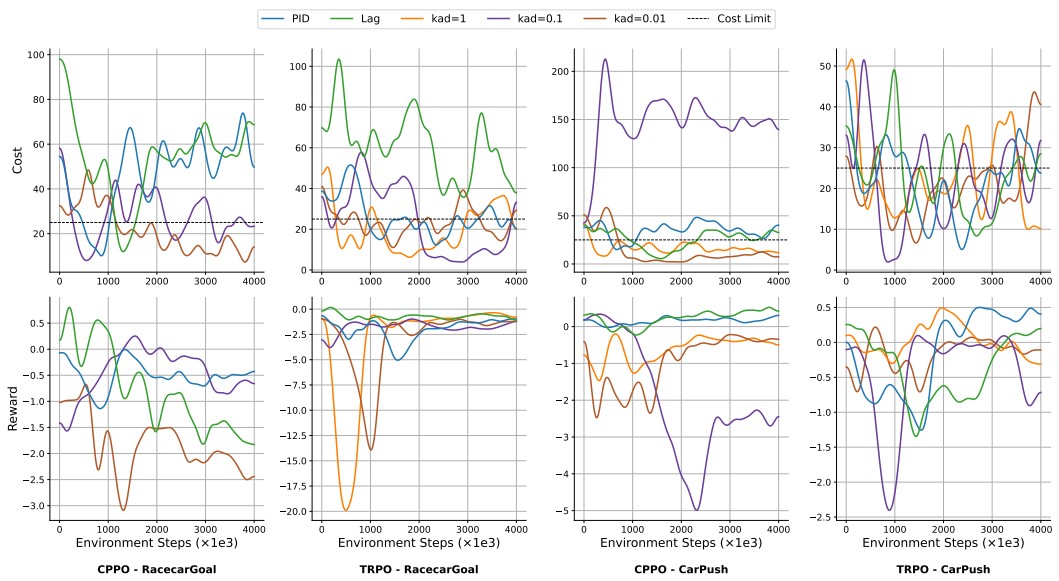

Figure 18: The training curve for TRPO and CPPO algorithms across CarPush and RacecarGoal environments with various $k_{ad}$ values, PID, and Lag methods.

Table 21: The proportion of constraint violations during training (Vio. Rate), the average magnitude of violations (Magnitude), and the average cost (Avg. Cost) for TRPO and CPPO algorithms across CarPush and RacecarGoal environments with various $c_r$ values, PID, and Lag methods. Bold values indicate the better performance compared to PID.

| Algorithm | Method | CarPush | | | RacecarGoal | | |
|---|---|---|---|---|---|---|---|
| | | Vio. Rate (%) | Magnitude | Avg. Cost | Vio. Rate (%) | Magnitude | Avg. Cost |
| TRPO | Lag | 43.83 | 7.99 | 26.19 | 87.33 | 37.36 | 61.53 |
| | PID | 38.40 | 4.84 | 21.96 | 44.60 | 7.04 | 26.15 |
| | $c_r = 0.05$ | 46.25 | 6.35 | 24.79 | **33.98** | **5.25** | **20.83** |
| | $c_r = 0.1$ | **30.20** | **3.86** | **20.36** | **29.05** | **3.44** | **18.95** |
| | $c_r = 0.15$ | **34.83** | 10.92 | 27.00 | **31.25** | **5.34** | **21.16** |
| | $c_r = 0.2$ | **28.60** | 7.32 | **20.72** | **40.65** | **6.10** | **23.67** |
| | $c_r = 0.25$ | **34.13** | 6.26 | 22.46 | **38.50** | **6.71** | **23.40** |
| CPPO | Lag | 46.43 | 6.67 | 25.90 | 84.35 | 30.16 | 53.38 |
| | PID | 67.28 | 12.80 | 34.67 | 79.25 | 23.88 | 46.44 |
| | $c_r = 0.05$ | **44.73** | **5.95** | **24.80** | **34.38** | **3.90** | **22.45** |
| | $c_r = 0.1$ | **16.25** | **4.05** | **13.46** | **33.08** | **5.78** | **21.22** |
| | $c_r = 0.15$ | **56.70** | **10.22** | **30.40** | **52.88** | **10.69** | **31.37** |
| | $c_r = 0.2$ | **12.30** | **2.31** | **13.45** | **48.95** | **8.77** | **26.91** |
| | $c_r = 0.25$ | **34.00** | **6.12** | **22.09** | **62.83** | **13.37** | **33.99** |

the training process. Our approach not only converges more effectively but also demonstrates a more favorable trade-off between reward maximization and cost minimization.

Overall, the results highlight the robustness of our method, as it achieves superior performance in the majority of scenarios. This indicates that the choice of $c_r$ significantly influences the balance between reward and cost, and our approach consistently outperforms existing methods under comparable conditions.

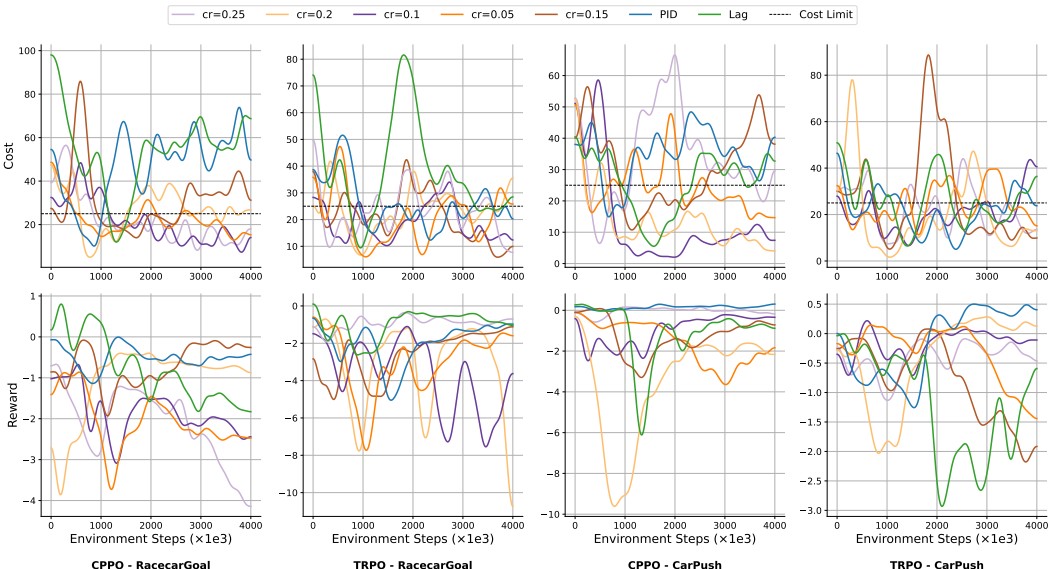

Figure 19: The training curve for TRPO and CPPO algorithms across CarPush and RacecarGoal environments with various $c_r$ values, PID, and Lag methods.

## F.6 NOISE SENSITIVITY ANALYSIS

Table 22: Training noise sensitivity on Swimmer-Velocity.

| Method / $\sigma$ | Vio. Rate (%) | Magnitude | Avg. Cost | Avg. Reward |
|---|---|---|---|---|
| PID | 35.43 | 1.77 | 22.47 | 27.73 |
| $\sigma = 0$ | 11.30 | 2.44 | 20.82 | 35.66 |
| $\sigma = 2$ | 10.93 | 2.55 | 21.93 | 22.41 |
| $\sigma = 5$ | **5.93** | 1.62 | **8.04** | 16.18 |
| $\sigma = 10$ | 23.95 | 3.55 | 19.93 | 12.41 |

Table 23: Evaluation noise sensitivity on Swimmer-Velocity.

| Method / $\sigma$ | Avg. Cost | Vio. Rate (%) | Magnitude | Avg. Reward |
|---|---|---|---|---|
| PID | 23.05 | 39.79 | 6.23 | 32.19 |
| $\sigma = 0$ | 18.40 | 12.07 | 11.35 | 39.55 |
| $\sigma = 2$ | 18.00 | 27.60 | 14.52 | 19.72 |
| $\sigma = 5$ | **4.16** | 4.57 | 9.18 | 17.23 |
| $\sigma = 10$ | 18.20 | 28.72 | 15.06 | 20.51 |

Tables 22–23 report the sensitivity of our TRPO-based ADRC approach to injected noise in the SWIMMER-Velocity environment. The results demonstrate that TRPO-ADRC is robust to disturbances: compared with PID-type controllers, TRPO-ADRC achieves consistently lower violation rates, smaller violation magnitudes, and reduced average costs, while maintaining stable reward learning. For example, during training, TRPO-ADRC reduces the violation rate from 35.43% (PID)

to 11.30% under $\sigma=0$, and further to only 5.93% under $\sigma=5$, with the average cost dropping from 22.47 to 8.04.

Importantly, these safety and stability benefits do not compromise convergence performance. At evaluation, TRPO-ADRC achieves competitive or even higher rewards while retaining robustness. Under $\sigma=0$, TRPO-ADRC attains higher reward than TRPO-PID (39.55 vs. 32.19) while simultaneously lowering both cost and violation rate. Even when the disturbance level increases ($\sigma=5, 10$), TRPO-ADRC sustains reasonable rewards with substantially reduced safety violations compared to PID. These findings confirm that TRPO-ADRC improves training stability and safety without hindering convergence, demonstrating strong robustness to noise in the SWIMMER-Velocity environment.

### F.7 ABLATION STUDY

To evaluate the contribution of key components in the ADRC-Lagrangian framework, we conducted an ablation study by systematically modifying specific features of the proposed method. Specifically, we examined the impact of replacing the transient process $r(t)$ with a static reference signal and fixing the dynamically adjusted compensation gain $\omega_o$ to a constant value. These modifications simplify the framework to a configuration resembling a PID-based Lagrangian method, enabling a fair comparison of their relative contributions.

In the first ablation, the transient process $r(t)$ is replaced with a fixed reference signal $r(t) = d$, as used in traditional PID Lagrangian methods. While $r(t)$ is designed in the full ADRC framework to provide a smooth transition toward the cost threshold denoted as $d$, setting $r(t) = d$ eliminates this smoothing effect. This simplification forces the system to directly track the constant reference signal, potentially causing abrupt updates to the policy parameters and destabilizing training. The updating law is transformed into:

$$\lambda_t = k_{ap}(x_1 - d) + k_{ad}x_2 + \omega_o k_{ap} \int_0^t (x_1(\tau) - d)d\tau. \tag{70}$$

For the second ablation, we fixed the compensation gain $\omega_o$ to ensure its parameters match those of the PID baseline. Specifically, we solved the following equations to determine fixed values for $\omega_o$, $k_{ap}$, and $k_{ad}$:

$$\begin{aligned} k_{ap} + \omega_o k_{ad} &= k_p, \\ \omega_o + k_{ad} &= k_d, \\ \omega_o k_{ap} &= k_i. \end{aligned} \tag{71}$$

The solutions to these equations yield parameter values that maintain equivalence with the PID updating law while removing the adaptivity of $\omega_o$. With these parameters, the updating law for the Lagrangian multiplier $\lambda_t$ reduces to:

$$\lambda_t = k_p(x_1 - r) + k_d(x_2 - \dot{r}) + k_i \int_0^t (x_1(\tau) - r(\tau))d\tau - \ddot{r}. \tag{72}$$

The ablation experiments evaluate the importance of the dynamic transient process $r(t)$ and the adaptive compensation gain $\omega_o$ in the ADRC-Lagrangian framework. Specifically, we tested two simplified configurations: "delete_r(t)," where the transient process $r(t)$ is replaced with a static reference signal $r(t) = d$, and "delete_$\omega_o$," where the dynamic adjustment of $\omega_o$ is replaced with fixed parameter values derived from PID-based methods. These modifications isolate the contributions of each component while retaining equivalent control parameters. The experiments were conducted in the CarButton and RacecarGoal environments using TRPO and CPPO as base algorithms, with the results summarized in Table 24.

Replacing $r(t)$ with a static reference signal ("delete_r(t)") resulted in higher violation rates and magnitudes compared to the full ADRC framework but still outperformed the PID baseline. For instance, in the RacecarGoal environment with CPPO, the violation rate decreased from 79.25% (PID) to 54.08% ("delete_r(t)"), but ADRC achieved a further reduction to 33.08%. Similarly, fixing $\omega_o$ ("delete_w0") impaired the system's adaptability to environmental changes, leading to increased average costs. In the same environment, the average cost dropped from 46.44 (PID) to 36.38 ("delete_w0") but was substantially lower with ADRC at 21.22. These results demonstrate that

Table 24: The proportion of constraint violations during training (Vio. Rate), the average magnitude of violations (Magnitude), and the average cost (Avg. Cost) for TRPO and CPPO algorithms across CarButton and RacecarGoal environments with various methods. Bold values indicate better performance compared to PID.

| Algorithm | Method | CarButton | | | RacecarGoal | | |
|---|---|---|---|---|---|---|---|
| | | Vio. Rate (%) | Magnitude | Avg. Cost | Vio. Rate (%) | Magnitude | Avg. Cost |
| TRPO | Lag | 55.90 | 17.65 | 39.28 | 87.33 | 37.36 | 61.53 |
| | PID | 85.15 | 18.12 | 42.13 | 44.60 | 7.04 | 26.15 |
| | delete_$\omega_o$ | **71.85** | **9.42** | **32.56** | **40.53** | 7.29 | **26.14** |
| | delete_r(t) | **74.95** | **16.36** | **40.23** | **31.65** | **6.15** | **22.26** |
| | ADRC | **26.35** | **6.14** | **23.42** | **29.05** | **3.44** | **18.95** |
| CPPO | Lag | 99.88 | 98.34 | 123.31 | 84.35 | 30.16 | 53.38 |
| | PID | 99.88 | 57.95 | 82.92 | 79.25 | 23.88 | 46.44 |
| | delete_$\omega_o$ | 99.90 | **51.61** | **76.58** | **65.40** | **13.99** | **36.38** |
| | delete_r(t) | **92.53** | **33.78** | **58.20** | **54.08** | **15.23** | **34.66** |
| | ADRC | **93.68** | **33.97** | **58.74** | **33.08** | **5.78** | **21.22** |

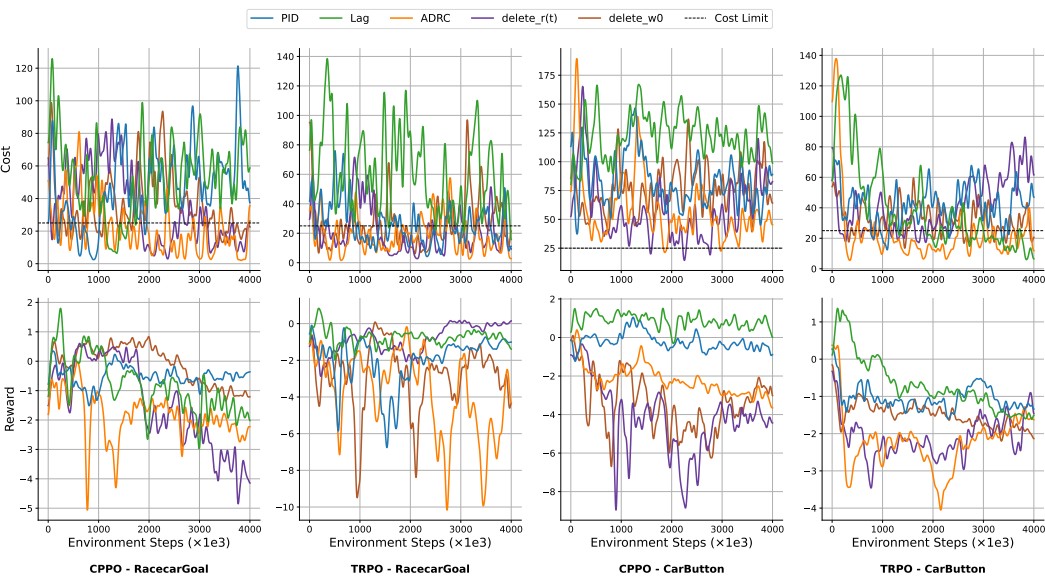

Figure 20: The training curve for TRPO and CPPO algorithms across CarButton and RacecarGoal environments with ablation study.

while both components are essential for optimal performance, even the simplified versions outperform PID, highlighting the robustness of the ADRC-Lagrangian framework.

Figure 20 provides the training curves for reward and cost across the ablation study. These curves illustrate the consistent performance improvements of our method throughout the training process. Our approach not only converges more effectively but also demonstrates a more favorable trade-off between reward maximization and cost minimization.

### F.7.1 ADDITIONAL ABLATION ON CPPO

To evaluate the effectiveness of our proposed dynamic parameter adjustment and transient process, we conducted ablation studies, with results summarized in Table 25. In this table, "Delete $r(t)$" refers to the removal of the dynamic adjustment component $r(t)$, while "Delete $\omega_o$" refers to the exclusion of the transient weight $\omega_o$ from the algorithm. The results show that removing either component results in a clear performance degradation in terms of violation rate, violation magnitude, and average cost. However, even with these removals, the performance of our approach remains superior to the baseline PID method, demonstrating the robustness of our framework. Additionally, the complete ADRC method achieves the best results across all metrics, further highlighting the significance of combining both $r(t)$ and $\omega_o$ in achieving optimal performance. For further details and results, please refer to Appendix F.7.

Table 25: Ablation study of CPPO algorithm under RacecarGoal.

| Method | Vio. Rate(%) | Magnitude | Avg. Cost |
|---|---|---|---|
| Delete $r(t)$ | 65.40 | 13.99 | 36.38 |
| Delete $\omega_o$ | 54.08 | 15.23 | 34.66 |
| ADRC | **33.08** | **5.78** | **21.22** |
| PID | 79.25 | 23.88 | 46.44 |
| Lag | 84.35 | 30.16 | 53.38 |

### F.8 CASE STUDY

To gain a deeper understanding of how our ADRC Lagrangian methods outperform the baseline, we conduct a case study adopting the TRPO algorithm in the CarCircle-1 environment. In this environment, agents are tasked with navigating around a fixed-radius circle. The agents' goal is to maintain a smooth circular trajectory while staying within the designated circular boundary and avoiding collisions with obstacles.

The reward structure in the CarCircle-1 environment is designed to encourage agents to follow the circle boundary as closely as possible while maintaining a smooth motion. High rewards are achieved when the agent's trajectory aligns with the circle's radius, and its velocity vector aligns tangentially to the circular path. The reward for the agent is calculated using the following formula:

$$\text{Reward} = \frac{\frac{-u \cdot y + v \cdot x}{r}}{1 + |r - R|} \cdot \text{reward\_factor}, \tag{73}$$

where $x, y$ represent the agent's position, $u, v$ represent the agent's velocity, $r$ is the distance from the agent to the circle center ($r = \sqrt{x^2 + y^2}$), $R$ is the fixed radius of the circle, and reward\_factor is a scaling constant. This formula incentivizes agents to maintain a smooth and stable trajectory around the circle.

To increase the complexity of the environment, CarCircle-1 introduces two vertical walls positioned symmetrically near the circle's boundary. These walls present an additional challenge, as agents must avoid crossing into the wall regions while navigating the circle. Costs are incurred when the agent violates safety constraints, such as exceeding the circle's radius or crossing the boundaries defined by the walls. The cost is computed using the following conditions:

$$\text{Cost} = \begin{cases} 1 & \text{if } |x| > \text{wall\_threshold or } \sqrt{x^2 + y^2} > R, \\ 0 & \text{otherwise,} \end{cases} \tag{74}$$

where wall_threshold is the horizontal boundary defined by the walls, and $R$ is the circle's radius.

In this study, we adopt the default settings in OmniSafe (Ji et al., 2024), with $R = 1.5$, wall_threshold $= 1.125$, and reward_factor $= 0.1$. The study is conducted using the TRPO algorithm, with hyperparameters detailed in Appendix D.1. The models are trained over 4,000 episodes, with each episode consisting of 1,000 steps. After training, the final checkpoint is used for evaluation. During evaluation, we simulate a single episode of 5,000 steps, where rewards are calculated using Eqn. 73, and costs are determined by Eqn. 74. We collected the position ant the corrsponding reward and the cost of the agents and the results of this case study are presented in Figure 21.

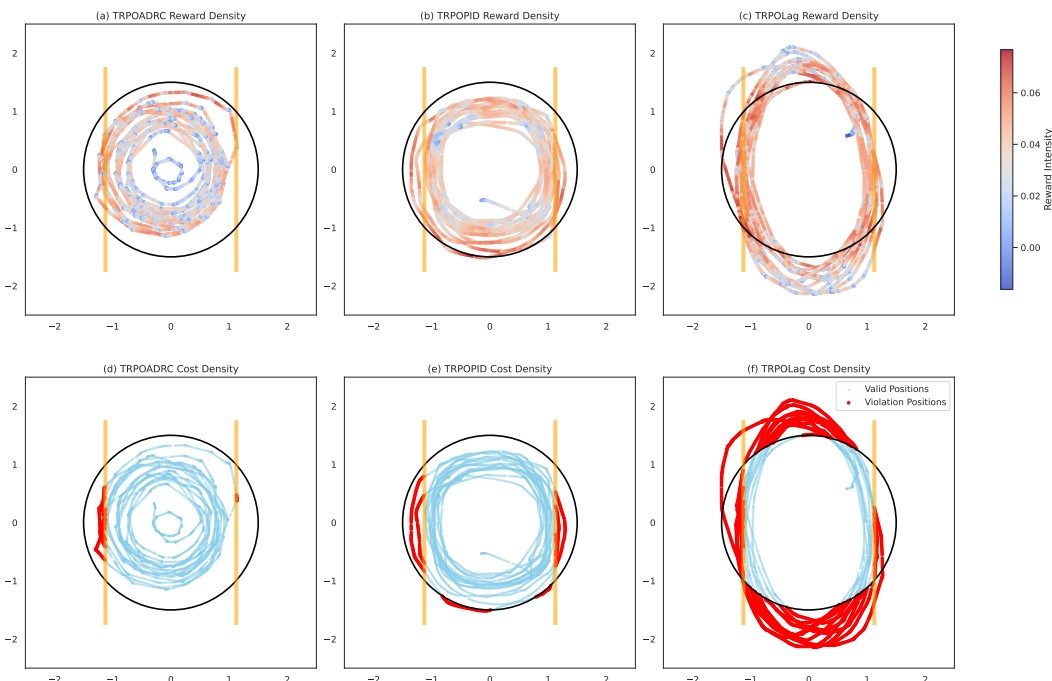

Figure 21: Reward density and cost analysis for TRPO algorithm adapting three Lagrangian methods under CarCircle environment.

In Figure 21, the color of the points represents the reward, with deeper colors indicating higher rewards. The first row visualizes the reward density at each position during the episode, while the second row illustrates the cost associated with each position. Blue points indicate safe positions, whereas red points represent unsafe ones. The results show that the classical Lagrangian method achieves the highest reward, but the trajectory deviates from a perfect circle, forming an ellipse instead. Agents trained with this method learn to avoid the walls but fail to recognize the importance of staying within the circle. To maximize rewards and bypass the walls, these agents move outside the circle, resulting in a total of 496 safety violations. In contrast, agents trained with the PID Lagrangian method demonstrate better safety awareness, recognizing that moving outside the circle is unsafe. However, to achieve higher rewards, they still cross the walls frequently, leading to 320 safety violations. Finally, agents trained with our ADRC method maintain a strict adherence to staying within the circle and exhibit only 132 safety violations, establishing a superior safety performance.

The improved results achieved by the ADRC method can be attributed to its ability to reduce phase lag and minimize oscillations, thereby enhancing the stability of training. These properties enable ADRC to maintain better control over the agent's behavior, ensuring stricter adherence to safety constraints while still optimizing for rewards. The stability and precision offered by ADRC during training allow the agent to effectively balance the trade-off between maximizing rewards and minimizing safety violations, demonstrating the advantages of our proposed method in challenging environments.

## F.9 FINAL POLICY PERFORMANCE

To assess the final performance of the trained policies rather than intermediate training behavior, we conducted experiments on the Swimmer and Hopper environments from the Velocity tasks suite. The results compare ADRC-based and PID-based methods under CPPO and TRPO frameworks.

Table 26: Performance on Swimmer environment.

| Algorithm | Avg Reward | Avg Cost | Violate Rate (%) |
|-----------|-----------|----------|------------------|
| CPPOPID   | 33.10     | 22.44    | 28.02            |
| CPPOADRC  | 29.39     | 16.77    | 14.16            |
| TRPOPID   | 28.72     | 21.34    | 37.85            |
| TRPOADRC  | 36.32     | 19.03    | 12.16            |

Table 27: Performance on Hopper environment.

| Algorithm | Avg Reward | Avg Cost | Violate Rate (%) |
|-----------|-----------|----------|------------------|
| CPPOPID   | 1466.47   | 48.20    | 30.00            |
| CPPOADRC  | 1520.18   | 8.20     | 24.63            |
| TRPOPID   | 1038.47   | 18.70    | 29.76            |
| TRPOADRC  | 1384.11   | 12.90    | 10.98            |

These results demonstrate that the ADRC-based method achieves lower constraint violation rates and costs, while maintaining or improving the overall reward compared to PID-based baselines.

## F.10 SENSITIVITY ANALYSIS OF PID LAGRANGIAN METHODS

We also empirically validated the sensitivity of PID Lagrangian methods to the control gain tuning, particularly for the derivative term $k_d$. Experiments were conducted in the CarPush and CarButton environments using CPPO algorithms with varying $k_d$ values.

Table 28: Sensitivity analysis of PID Lagrangian methods by varying the derivative gain $k_d$ on CarPush and CarButton environments (using CPPO).

| Environment | $k_d$ Value | Violate Rate (%) | Violate Magnitude | Avg Cost | Avg Reward |
|-------------|-------------|------------------|-------------------|----------|------------|
| CarPush (CPPO) | 1 | 50.55 | 19.17 | 35.28 | -1.37 |
|  | 0.1 | 66.00 | 24.87 | 46.81 | -2.26 |
|  | 0.01 | 67.28 | 12.80 | 34.67 | 0.15 |
|  | 0.001 | **99.80** | **96.50** | **121.45** | -0.02 |
| CarButton (CPPO) | 1 | **99.88** | **89.36** | **114.33** | 0.21 |
|  | 0.1 | 99.80 | 43.32 | 68.29 | -1.89 |
|  | 0.01 | 99.88 | 64.74 | 89.72 | -0.69 |

These results clearly illustrate that PID Lagrangian methods are highly sensitive to the choice of the $k_d$ value. Suboptimal tuning can lead to substantial degradation in both safety and overall performance.

## G LARGE LANGUAGE MODELS USAGE

We used GPT-4 (OpenAI) for grammar and style editing of the paper and for debugging auxiliary code (e.g., resolving error messages and minor refactoring). All technical ideas, method designs, experiments, and conclusions were created and verified by the authors. No confidential or reviewer-only information was shared with the model.

## H    Computational Cost Analysis

In this section, we evaluate the computational cost of our ADRC Lagrangian method compared to the PID and classical Lagrangian methods (Lag). The evaluation includes normalized computation time during the rollout phase (interaction with the environment) and the update phase (policy updates). The analysis is conducted using the DDPG and CPPO algorithms on the RacecarButton task, which features a multi-dimensional action space.

Table 29: Normalized Computation Time for DDPG and CPPO under RacecarButton task.

| Metric | DDPG | | | CPPO | | |
|---|---|---|---|---|---|---|
| | PID | Lag | ADRC | PID | Lag | ADRC |
| Rollout Time | 0.95 | 1.00 | 0.95 | 0.95 | 1.00 | 0.95 |
| Update Time | 0.94 | 1.00 | 1.00 | 0.94 | 1.00 | 1.00 |
| Total Time | 0.95 | 1.00 | 0.97 | 0.95 | 1.00 | 0.97 |

Compared with PID, at each episode, our ADRC method introduces minimal additional computation. Specifically, ADRC calculates the reference signal $r(t)$, solves the equation defined by Eqn.19, and determines the optimal parameters $\omega_o$ based on Eqn.20. These operations involve fixed and lightweight calculations that do not scale with the problem size, ensuring no additional time complexity is introduced.

The results in Table 29 confirm that ADRC achieves comparable computation times to the baselines. The rollout time of ADRC matches that of PID and slightly outperforms Lag, demonstrating efficiency in policy adjustments. During the update phase, ADRC incurs no additional cost compared to PID and Lag, aligning with the theoretical analysis that these computations are efficiently integrated into the training process.

All experiments were conducted on a machine equipped with an NVIDIA RTX 3090 GPU with 24GB of memory. Each task was trained for 4 million steps. For TRPO and CPPO, each training run takes approximately 10 GPU-hours, while DDPG and TD3 require about 18 GPU-hours per run.

