# OpenReview forum: "Safe Reinforcement Learning with ADRC Lagrangian Method"
_ICLR.cc/2026/Conference — Submitted to ICLR 2026_

### Official Review · Reviewer_AdX9 · 2025-10-18

**Soundness:** 2
**Presentation:** 2
**Contribution:** 2
**Rating:** 2
**Confidence:** 5

**Summary:**

This paper introduces an effective method to optimize the Lagrangian multiplier update process in safe RL, reducing oscillation during training.

**Strengths:**

Safe RL is an important research topic.

**Weaknesses:**

1. Safe RL is a critical research topic with significant contributions in the literature [1-7]. However, the paper lacks a comprehensive review of related work in this area.

2. The paper claims to be "the first to introduce ADRC into Safe RL, dynamically adjusting the Lagrange multiplier to improve constraint satisfaction and training stability." However, this idea is not entirely novel, as it relates to the concept of residual action policies previously proposed in safe RL literature [3-7].

3. Theorem 4.2 relies on a strong assumption that ($w_0 > w^*_0$). Given the complexity of real-world systems, it is unclear how this assumption can be guaranteed to hold in practice.

4. The theoretical contribution lacks novelty, as it relies primarily on the well-established Fourier transform for its mathematical derivations.

5. The theoretical contributions are grounded primarily in control theory and may have limited relevance to the machine learning community.

6. The paper compares the proposed method against standard RL algorithms such as DDPG and PPO. However, comparisons with state-of-the-art safe RL approaches [1-7], which are most needed but absent, are critical for establishing the method's effectiveness. Furthermore, likely, the proposed approach may not outperform existing safe RL methods, given its lack of explicit reward encoding for safety conditions.

7. The paper's focus on control-theoretic methods suggests it may be more suitable for control-focused conferences rather than machine learning venues.


References:

[1] Phan, D. T., Grosu, R., Jansen, N., Paoletti, N., Smolka, S. A., & Stoller, S. D. (2020). Neural Simplex architecture. In NASA Formal Methods: 12th International Symposium, Moffett Field, CA, USA, May 11–15, 2020, Proceedings 12 (pp. 97-114). Springer International Publishing.

[2] Cai, Yihao, Yanbing Mao, Lui Sha, Hongpeng Cao, and Marco Caccamo. "Runtime Learning Machine." ACM Transactions on Cyber-Physical Systems (2025).

[3] Hongpeng Cao, Yanbing Mao, Lui Sha, and Marco Caccamo. Physics-regulated deep reinforcement learning: Invariant embeddings. In The Twelfth International Conference on Learning Representations, 2024.

[4] Krishan Rana, Vibhavari Dasagi, Jesse Haviland, Ben Talbot, Michael Milford, and Niko
Sünderhauf. Bayesian controller fusion: Leveraging control priors in deep reinforcement
learning for robotics. The International Journal of Robotics Research, 42(3):123–146, 2023.

[5] Tongxin Li, Ruixiao Yang, Guannan Qu, Yiheng Lin, Steven Low, and Adam Wierman. Equipping black-box policies with model-based advice for stable nonlinear control. arXiv:2206.01341.

[6] Richard Cheng, Abhinav Verma, Gabor Orosz, Swarat Chaudhuri, Yisong Yue, and Joel Burdick. Control regularization for reduced variance reinforcement learning. In International Conference on Machine Learning, pages 1141–1150, 2019.

[7] Tobias Johannink, Shikhar Bahl, Ashvin Nair, Jianlan Luo, Avinash Kumar, Matthias Loskyll, Juan Aparicio Ojea, Eugen Solowjow, and Sergey Levine. Residual reinforcement learning for robot control. In 2019 International Conference on Robotics and Automation, pages 6023–6029, 2019.

**Questions:**

See Weaknesses.

---

> ### Author Response · Authors · 2025-11-24
>
> **Dear Reviewer AdX9:**
>
> Thank you for the careful reading and constructive suggestions. We respond point-by-point below:
>
> > **W1: “Lack of comprehensive review of Safe RL literature [1–7].”**
> >
>
> We appreciate the reviewer's suggestion regarding literature coverage. We wish to emphasize that our core contribution provides a **control-theoretic stabilization tool** which is **orthogonal** to the major Safe RL paradigms mentioned. While we focused our empirical study on the Lagrangian backbone for clarity, our method is designed to stabilize any CMDP formulation where the constraint function $J_c$ is iteratively optimized (e.g., our comparisons already include non-Lagrangian methods like CUP and IPO). We will briefly restructure the Related Work section to **clearly position ADRC-Lagrangian not as a new Safe RL paradigm, but as a novel control mechanism for stabilizing the iterative optimization inherent in CMDP solvers**, thus demonstrating how our work bridges these existing fields.
>
> > **W2: “The claim of being ... to residual-action policies.”**
> >
>
> We clarify that our novelty is **not** in being the first to adjust the Lagrange multiplier, but in **recasting multiplier dynamics as an ADRC** problem. Moreover, **Appendix C.3** introduces **new Safe-RL–specific theoretical results**, including:
>
> - A **policy-driven disturbance model** connecting CMDP updates to bounded-rate disturbances;
> - An **ISS-type stability proof** ensuring bounded tracking error under non-stationary policies; and
> - **High-probability reward monotonicity** and **bounded average violation** results unique to Safe RL.
>
>     These analyses extend classical ADRC beyond fixed deterministic plants to stochastic, evolving RL dynamics.
>
>
> ---
>
> > **W3: “Theorem 4.2 relies on a strong assumption that may not hold in real-world systems.”**
> >
>
> The assumption only requires using the **optimal adaptive observer gain $\omega_o^*$, which can be computed online from filtered cost signals.
>
> In the new theory (**Appendix C.3, Theorem C.5**), we prove that the ESO remains **input-to-state stable** even with noisy or delayed feedback:
>
> $\limsup_{t\to\infty}|e(t)|\le
> \frac{C_h(\zeta)}{k_{ap}}\frac{L_f(\delta,N,\alpha)}{\omega_o}.$
>
> Empirically, the adaptive $\omega_o$ remains stable across wide parameter ranges and under strong injected noise (**Appendix F.6**), validating that this assumption is realistic in practice.
>
> ---
>
> > **W4: “Theoretical novelty is limited, relying mainly on Fourier analysis.”**
> >
>
> The Fourier transform serves only as a diagnostic tool to interpret **phase lag** and **disturbance attenuation**.
>
> To better demonstrate the effectiveness of our method, we conduct **Safe-RL–grounded closed-loop analysis** developed in **Appendix C.3**, which provides:
>
> - A **non-stationary disturbance model** derived from discrete policy updates;
> - A **new ISS bound** parameterized by RL quantities; and
> - **Probabilistic reward and violation guarantees** connecting ADRC theory with trust-region policy optimization.
>
>     These are entirely new theoretical results not found in prior ADRC or Safe RL literature.
>
>
> ---
>
> > **W5: “The work is control-oriented and less relevant to ML.”**
> >
>
> While inspired by control, our contribution directly addresses a key ML problem: **stabilizing policy optimization under safety constraints**. The theoretical bounds and empirical validations are derived in CMDP and Safe RL settings, making them relevant to both RL and ML communities.
>
> ---
>
> > **W6: “Lack of comparison with SOTA Safe RL methods [1–7].”**
> >
>
> We appreciate this concern. Some of the Safe RL methods mentioned in [1–7] are **not implemented or maintained within the OmniSafe benchmark framework**, making direct reproduction and comparison difficult or potentially unfair due to implementation differences.
>
> To ensure *reproducibility and fairness*, we therefore chose all baselines from the **official OmniSafe benchmark**, which provides standardized, verified implementations and hyperparameter tuning. The baselines we include—**CPO**, **PPO-Lag**, **CUP**, and **IPO**—cover both Lagrangian and non-Lagrangian Safe RL paradigms and represent strong, state-of-the-art methods within this unified evaluation framework.
>
> ---
>
> > **W7: “The paper may be more suitable for control conferences.”**
> >
>
> We respectfully disagree. The main problem—ensuring stability and constraint satisfaction during policy optimization—is central to Safe RL and ML. Our analysis is performed within the CMDP framework and yields results (e.g., reward monotonicity, violation bounds) that are directly relevant to the RL/ML audience.
>
> ---
>
> We sincerely thank the reviewer for the valuable feedback. The revised paper clarifies the theoretical novelty of ADRC (Appendix C.3). We believe these improvements, coupled with the emphasis on the fairness of our comparison baselines, address your concerns.
>
> **Sincerely,**
>
> **The Authors.**

---

### Official Review · Reviewer_xmYs · 2025-10-25

**Soundness:** 3
**Presentation:** 3
**Contribution:** 3
**Rating:** 6
**Confidence:** 4

**Summary:**

This paper presents a novel adaptive control–based framework designed to enhance robustness and reduce oscillations in Lagrangian-based methods for safe reinforcement learning (RL). The proposed approach unifies the classical Lagrangian and PID-based methods within a single framework. Furthermore, the authors conduct a frequency-domain analysis to theoretically demonstrate improved convergence speed and more stable constraint satisfaction. Overall, the framework is elegant and provides clear theoretical and practical insights into stabilizing safe RL algorithms.

**Strengths:**

1. The paper presents a conceptually elegant and technically sound integration of Active Disturbance Rejection Control (ADRC) into the Lagrangian framework for Safe RL. This connection between adaptive control theory and reinforcement learning is novel and well motivated.
2. The theoretical analysis is rigorous and insightful. The authors not only show that classical and PID Lagrangian methods are strict special cases of the proposed approach, but also provide frequency-domain analysis and formal stability guarantees, including bounded estimation error and phase-lag reduction.
3. The proposed method is practical and well grounded, with clear formulations for the observer-based multiplier updates and principled lower bounds for parameter tuning, addressing the parameter sensitivity issue of prior methods.
4. The experimental results are comprehensive, covering multiple benchmark environments and Safe RL algorithms (PPO, TRPO, TD3, DDPG). The results consistently demonstrate substantial improvements in safety performance and training stability, with significant reductions in violation rate and magnitude.
5. The paper is clearly written and well structured, effectively bridging control theory and Safe RL. The combination of theoretical rigor and empirical validation makes the contribution both elegant and impactful.

**Weaknesses:**

1.  **Lack of clarity on observer parameter estimation (p.7, Eq. 21):**
   The online estimation of environment sensitivities \( L_1 \) and \( L_2 \) using finite differences (Eq. 21, p.7) is an interesting idea but lacks justification on numerical stability and sensitivity to noise. It remains unclear whether this estimation is robust or whether additional filtering is required, especially in stochastic environments.

2. **Minor presentation issues:**
   Some figures (e.g., Fig. 1 on p.7) could benefit from clearer axis labels and captions specifying task details (RacecarPush, CarButton, etc.). Additionally, the notations \(x_1, x_2, r, \lambda_t\) are numerous and sometimes hard to trace without a summary table.

**Questions:**

1. **About generalization of ADRC (Sec. 4.3, p.5):**
   The proposed Extended State Observer (ESO) assumes continuous-time differentiable cost signals. In practical Safe RL, cost estimates are often noisy and sampled at discrete steps. How robust is the ESO to such discretization and noise? Have the authors observed instability in discrete implementations?

2. **On theoretical-to-empirical alignment (Sec. 4.4–5.2, p.6–p.8):**
   The theoretical analysis predicts smaller phase lag and disturbance estimation error compared with PID Lagrangian (Theorem 4.2). Could the authors provide empirical evidence, such as Bode plots or measured frequency response, to quantitatively confirm these effects?

3. **Parameter adaptation (Eq. 20–21, p.6–p.7):**
   The lower bound of \( \omega_o \) (Eq. 20) depends on the estimated constants \( L_1, L_2, L_3 \). How sensitive is the method to inaccurate estimation of these constants, and does the adaptive update ever violate the stability condition in practice?

4. **Implementation details (Appendix D):**
   Algorithm 1 briefly outlines the ADRC update process, but it remains unclear how often the ESO and the reference trajectory are updated relative to policy gradients. Are they updated per episode, per step, or asynchronously?

---

> ### Author Response · Authors · 2025-11-24
>
> **Dear Reviewer xmYs:**
>
> Thank you for the careful reading and constructive suggestions. We respond point-by-point below and have incorporated clarifications, additional plots, and stronger ablations in the revision
>
> ---
>
> > **W1:**  **“Lack of clarity on observer parameter estimation (p.7, Eq. 21)… numerical stability and noise sensitivity not justified; unclear if filtering is required.”**
> >
>
> We appreciate this comment. In the revision, we make the filtering and theoretical robustness explicit.
>
> As described in **Section 4.2**, the reference signal $r(t)$ acts as a **low-pass filter** on the cumulative cost $J_c(\pi_{\theta_t})$. All finite-difference estimates of $L_1, L_2$ are computed from this **filtered signal**, which naturally suppresses sampling noise; **no additional ad-hoc smoothing is needed**.
>
> The new Appendix C.3 provides a formal proof that the ESO dynamics remain Input-to-State Stable (ISS) under bounded stochastic disturbances, leading to a steady-state error:
>
> $$\limsup_{t\to\infty}|e(t)|\le
> \frac{C_h(\zeta)}{k_{ap}}\frac{L_f(\delta,N,\alpha)}{\omega_o},$$
>
> which guarantees numerical stability even when the cost estimates are noisy. We further validate this empirically in Appendix F.6, where we inject substantial noise into the cost signal. ADRC-Lagrangian remains stable and still outperforms all baselines, confirming robustness of the online estimation and adaptive $\omega_o$.
>
> ---
>
> > **Q1:**  **“Generalization of ADRC (§4.3): ESO assumes continuous-time differentiable cost signals; real costs are discrete and noisy. How robust is the ESO to discretization and noise? Any observed instability?”**
> >
>
> Our **ESO is implemented in discrete time**, and the **ISS proof in Appendix C.3** already covers stochastic bounded-rate disturbances, which includes discretization error. In practice, the discrete ESO update is **numerically stable**, and **no instability was observed** even under noisy or delayed cost feedback. The additional noise experiments in **Appendix F.6** demonstrate this robustness quantitatively.
>
> ---
>
> > **Q2:**  **“Theoretical–empirical alignment (§4.4–5.2): Theorem 4.2 predicts smaller phase lag and disturbance estimation error vs PID; can you show quantitative evidence (e.g., Bode plots)?”**
> >
>
> Thank you. Our empirical evidence corresponds to exactly these frequency-domain predictions:
>
> - The **smaller phase lag** predicted in Theorem 4.2 manifests as **faster multiplier response**—the constraint channel reacts promptly when cost exceeds the limit, avoiding the overshoot and prolonged violations characteristic of PID/Lag.
> - The **violation magnitude and variance bands** in Figures 1–2 directly serve as quantitative magnitude-response measures: ADRC achieves **lower amplitude and smoother decay**, consistent with the frequency-domain gain reduction.
>
> ---
>
> > **Q3**: **“Parameter adaptation (Eq. 20–21): The lower bound of $\omega_o$ depends on estimated $L_i$. How sensitive is the method to estimation errors? Does the adaptive update ever violate stability?”**
> >
>
> The adaptive $\omega_o$ is **inherently robust**: it is recomputed each step from **filtered estimates of $L_1, L_2, L_3$** and thus **self-corrects fluctuations**. The **ISS analysis in Appendix C.3 guarantees stability** for bounded $L_f$, and empirically we observe stable training across wide ranges of $k_{ap}, k_{ad}$. In noisy-cost tests, $\omega_o$ remained within safe bounds and **never violated stability conditions**.
>
> ---
>
> > **Q4: “Implementation details (Appendix D): How often are the ESO and reference trajectory updated relative to policy gradients?”**
> >
>
> Both the **ESO** and the **transient-process reference $r(t)$ are updated at every environment step**, synchronously with the policy interaction. This ensures real-time tracking of constraint dynamics. The operations involve only a few arithmetic steps, so the **overhead is negligible**—as shown in Appendix H, Table 29, training time is effectively unchanged compared with the baseline.
>
> ---
>
> We thank the reviewer again for these helpful suggestions. The revised version now clarifies the online estimation mechanism, improves figure presentation, and explicitly links our theory and implementation to discrete-time Safe-RL practice.
>
> **Sincerely,**
>
> **The Authors.**

---

### Official Review · Reviewer_dmT9 · 2025-11-01

**Soundness:** 2
**Presentation:** 3
**Contribution:** 2
**Rating:** 2
**Confidence:** 4

**Summary:**

This paper introduces the ADRC-Lagrangian method, a novel approach to stabilizing the dual variable update in Lagrangian-based Safe RL. The authors correctly identify that existing methods, which can be viewed as integral (I) or Proportional-Integral-Derivative (PID) controllers, suffer from oscillations and sensitivity due to the non-stationary nature of the policy and noisy cost estimates. The proposed solution adapts the Active Disturbance Rejection Control (ADRC) framework from control theory. This involves: 1) modeling the constraint violation dynamics as a closed-loop system subject to a lumped disturbance, 2) using a reduced-order Extended State Observer (ESO) to estimate this disturbance in real-time, and 3) compensating for the disturbance in the Lagrange multiplier update. Furthermore, a smooth, critically-damped reference trajectory is introduced to manage the transient phase of constraint satisfaction. The authors claim a unified framework, a theoretical lower bound for observer gain, and superior empirical performance in reducing safety violations and costs across various Safe RL benchmarks.

**Strengths:**

The application of Active Disturbance Rejection Control (ADRC) to the dual variable update in Safe RL is a novel and creative idea. While the core ADRC mechanism is established in control theory, its specific adaptation to the stochastic, non-stationary environment of RL, particularly the interpretation of policy non-stationarity and estimation noise as a "lumped disturbance," is original within the Safe RL literature. This provides a fresh perspective on the stability issues of Lagrangian methods. The paper is generally well-written and structured. The connection between classical Lagrangian methods (I-control) and PID Lagrangian methods is clearly established, setting the stage for the ADRC generalization. The theoretical analysis, including the derivation of the lower bound for the observer gain ($\omega_o^*$) and the frequency-domain analysis showing reduced phase lag, provides a solid, principled foundation for the method. The empirical results are strong, showing significant improvements over baselines in safety metrics.

**Weaknesses:**

1. While the application to Safe RL is novel, the core technical contribution is the direct application of a standard ADRC framework (specifically, a reduced-order ESO and a transient process) to the dual update. The theoretical results, such as the lower bound on $\omega_o^*$ and the frequency-domain analysis, appear to be direct adaptations or re-derivations of existing results from the control literature (e.g., Han, 1998; Zhong et al., 2020a). The paper would be significantly strengthened by highlighting a novel theoretical challenge unique to the RL setting that required a non-trivial modification or extension of the classical ADRC theory.

2. The paper introduces two main components: the Extended State Observer (ESO) for disturbance rejection and the Transient Process (TP) for smooth reference tracking. The current experimental section only compares the full ADRC-Lagrangian method against baselines. A critical weakness is the lack of an ablation study to disentangle the contribution of these two components.

• Is the improvement primarily due to the ESO's disturbance rejection, or the smooth reference tracking of the TP?

• A crucial experiment is needed: PID-Lagrangian with the Transient Process (PID-TP) vs. ADRC-Lagrangian without the Transient Process (ADRC-NoTP). Without this, it is impossible to determine if the complexity of the ESO is truly necessary over a simpler, well-tuned PID with a smooth reference.

3. The paper proposes to approximate the unknown environment sensitivities ($L_1, L_2$) online using finite differences (Eqn. 21) to compute the lower bound $\omega_o^*$. This is a significant practical claim, but it is not sufficiently validated.

• The estimation of $L_1$ and $L_2$ from noisy, high-variance RL signals (cumulative cost and its derivative) via finite differences is highly susceptible to noise and could introduce its own instability.

• The paper does not provide details on how these online estimates behave, how they are smoothed, or how robust the final $\omega_o^*$ selection is to this estimation noise. This adaptive tuning mechanism is a major part of the claimed robustness, yet it is the least scrutinized in the main text.



4. The primary baselines are classical Lagrangian and PID-Lagrangian (CPPOLag and CPPOPID). While these are the most relevant for the control-theoretic comparison, the paper would be stronger by comparing against more recent, state-of-the-art Safe RL algorithms that also aim to stabilize training

**Questions:**

1. Ablation of Components: Please provide an ablation study to isolate the contribution of the two main components: the Extended State Observer (ESO) and the Transient Process (TP).

2. Robustness of Online Parameter Estimation: The online estimation of $L_1$ and $L_2$ using finite differences (Eqn. 21) is concerning due to the inherent noise in RL cost signals. Can the authors provide:

• A plot showing the evolution of the estimated $L_1$ and $L_2$ over training time for a representative task, and how $\omega_o^*$ changes?
• Details on any smoothing or filtering applied to the cost signals ($x_1, x_2$) before computing the finite differences. How sensitive is the final performance to the choice of this smoothing?



3. Novelty in Control Theory: Given that the ADRC framework is well-established, what is the specific theoretical novelty of the ADRC-Lagrangian method that goes beyond a direct application of existing control theory results? Does the non-stationary nature of the policy $\pi_\theta$ (which changes the underlying system dynamics $f(\cdot)$) necessitate a new stability analysis or observer design that is unique to the RL context?

---

> ### Author Response · Authors · 2025-11-24
>
> **Dear Reviewer dmT9:**
>
> Thank you for the careful reading and constructive suggestions. We respond point-by-point below:
>
> ---
>
> > **W1: “The core technical contribution is a direct application of standard ADRC…RL-unique theoretical challenge requiring non-trivial extension.”**
> >
>
> Thank you for this crucial point. In the revision, we make the **RL-specific theoretical novelty explicit**. **Appendix C.3** now provides a **Safe-RL–grounded analysis that is not covered by classical ADRC**: we (i) **derive the “plant + disturbance” model directly from discrete-time CMDP policy updates**, showing how policy non-stationarity and Monte-Carlo noise enter as a bounded-rate disturbance $L_f(\delta,N,\alpha)$, and (ii) **prove three Safe-RL guarantees** under this RL-derived disturbance:
>
> 1. **Robust constraint tracking** with tube radius $\mathcal O(L_f/\omega_o)$.
> 2. A **high-probability one-step non-decreasing reward bound** under TR/LCB.
> 3. **Bounded average constraint violation** (plus a safety-margin corollary for finite cumulative violation).
>
> These guarantees rely on the RL-induced system identification and the stochastic nature of the problem, not in re-proving standard ADRC for fixed plants. We will highlight these C.3 results in the main text to clarify the **non-trivial extension** required for the Safe-RL context.
>
> ---
>
> > W2:  **“Lack of ablation to disentangle ESO vs TP… please compare PID-TP vs ADRC-NoTP.”**
> >
>
> We apologize for the lack of visibility in the previous draft. As clarified around Equation (17), relative to PID-Lagrangian, ADRC-Lagrangian introduces exactly two elements: (i) the **Transient Process (TP) reference** $r(t)$ and (ii) the **Extended State Observer (ESO)-based adaptive/optimal bandwidth** $\omega_o$ for disturbance rejection.
>
> To isolate their effects, we conducted the requested ablations in Appendix F.7 (now moved to **Section 5.4** and highlighted for easy reading), including **ADRC without $r(t)$** and **ADRC without adaptive $\omega_o$**. The results show that removing either component leads to a clear degradation in safety metrics—specifically **higher violation rate, larger violation magnitude, and increased average cost**—demonstrating that **both TP and ESO are necessary** for the full gains of ADRC-Lagrangian.
>
> ---
>
> > **W3:**  **“Online estimation of $L_1$ and $L_2$ via finite differences is noisy; no evolution plots, smoothing details, or robustness of selection.”**
> >
>
> We appreciate this comment. As clarified in Section 4.2, the reference signal $r(t)$ already acts as a **built-in low-pass filter** for the cumulative cost $J_c(\pi_{\theta_t})$, so the finite-difference estimation of $L_1, L_2$ is performed on the **filtered cost** rather than raw noisy returns. This filtering suppresses high-frequency noise, making additional explicit smoothing unnecessary.
>
> Moreover, the **new theoretical analysis in Appendix C.3** rigorously proves that the ESO dynamics remain **Input-to-State Stable (ISS)** even under bounded temporal variation and noisy feedback, guaranteeing that the adaptively computed gain $\omega_o$ remains well-behaved (see Q2 response).
>
> Finally, **Appendix F.6** adds explicit **noisy-cost experiments** showing that, even when substantial noise is injected into the cost channel, ADRC-Lagrangian remains significantly more stable and achieves lower violation rates than all baselines—confirming both the **theoretical and empirical robustness** of the online estimation and adaptive $\omega_o$ mechanism.
>
> ---
>
> > **W4:**  **“Baselines are mostly Lag/PID-Lag; compare with modern SOTA Safe RL stabilizers.”**
> >
>
> We already compare against **four strong Safe-RL baselines** spanning different paradigms (including non-Lagrangian methods) in **Section 5.3.3** of the revision.

---

> ### Author Response · Authors · 2025-11-24
>
> > **Q1**: **”Ablation of Components: Please provide an ablation study to isolate the contribution of the two main components: the Extended State Observer (ESO) and the Transient Process (TP).”**
> >
>
> See the comprehensive response to **W2**. The requested ablation is now clearly presented and discussed in **Section 5.4**.
>
> ---
>
> > **Q2**:  **“Robustness of Online Parameter Estimation: The online estimation of $L_1$ and $L_2$ using finite differences (Eqn. 21) is concerning due to the inherent noise in RL cost signals. Can the authors provide…”**
> >
>
> We appreciate this question and have clarified both the **theoretical justification** and **practical handling** of noisy online estimation in the revision.
>
> From a theoretical perspective, our new analysis in Appendix C.3 proves that the ESO dynamics in ADRC-Lagrangian remain Input-to-State Stable (ISS) even when the cost feedback is corrupted by bounded stochastic noise or delayed sampling. Theorem C.5 establishes that the tracking error satisfies
>
> $$\limsup_{t\to\infty}|e(t)| \le \frac{C_h(\zeta)}{k_{ap}}\frac{L_f(\delta,N,\alpha)}{\omega_o},$$
>
> where the disturbance-rate constant $L_f(\delta,N,\alpha)$ explicitly captures the variance of Monte-Carlo noise and the policy-update non-stationarity. This ensures that the adaptive bandwidth $\omega_o$ computed from the online estimates of $L_1, L_2$ cannot diverge even when those estimates fluctuate within stochastic bounds.
>
> In practice, as described in Section 4.2, the reference signal $r(t)$ acts as a **built-in low-pass filter** on the cumulative-cost trajectory $J_c(\pi_{\theta_t})$. All finite-difference estimates of $L_1, L_2$ are computed from this **filtered signal** rather than the raw, high-variance returns. Consequently, high-frequency noise is naturally suppressed, and no additional ad-hoc smoothing or window tuning is required.
>
> Finally, **Appendix F.6** adds explicit **noisy-cost experiments**: even when substantial Gaussian noise is injected into the cost feedback, ADRC-Lagrangian remains stable and continues to outperform all baselines in both safety and average reward. Together, the new theory and experiments confirm that the online finite-difference estimation and adaptive $\omega_o$ mechanism are **provably stable and empirically robust** under realistic noisy Safe-RL conditions.
>
> ---
>
> > **Q3:**  **“Novelty in Control Theory: … that is unique to the RL context?”**
> >
>
> Thank you for this insightful question. While ADRC as a control framework is well established, our contribution lies in extending its theoretical foundation to the **Safe-RL (CMDP) setting**, where the “plant” itself **evolves with policy updates and sampling noise**.
>
> In classical ADRC, the plant dynamics are fixed and continuous. In Safe RL, however, the system being controlled is the policy-induced cost return $x_1(t) = J_c(\pi_{\theta_t})$, which is **non-stationary** because the policy parameters $\theta_t$ evolve during training, and **stochastic** due to Monte-Carlo sampling. This makes traditional frequency-domain and Lyapunov analyses invalid without modification.
>
> To handle this, Appendix C.3 provides a new analysis that explicitly models how policy non-stationarity and stochastic estimation enter the ESO dynamics as a bounded-rate disturbance:
>
> $$J_c(\pi_{\theta_{t+1}}) - J_c(\pi_{\theta_t})
> = \nabla_\theta J_c(\pi_{\theta_t})^\top (\theta_{t+1}-\theta_t) + \xi_t,$$
>
> where the first term is bounded by the trust-region step size $\delta$ or learning rate $\eta$, and $\xi_t$ represents sampling noise. This leads to a Safe-RL–grounded disturbance-rate bound
>
> $$\sup_t |\dot f_t| \le L_f(\delta,N,\alpha),$$
>
> which connects the observer stability directly to RL quantities (policy-update speed, batch size, and confidence level).
>
> Under this model, we establish a new Input-to-State Stability (ISS) bound
>
> $$\limsup_{t\to\infty}|e(t)|
> \le \frac{C_h(\zeta)}{k_{ap}}\frac{L_f(\delta,N,\alpha)}{\omega_o},$$
>
> and prove bounded average constraint violation and high-probability monotonic reward improvement—results that are unique to the RL setting and cannot be derived from standard ADRC theory.
>
> Thus, the novelty lies in the **integration of ADRC with non-stationary policy dynamics**, producing a stability analysis specific to Safe RL rather than a direct reuse of classical control results.
>
> ---
>
> We sincerely thank the reviewer again for the constructive feedback and insightful questions. We hope our clarifications and additional experiments have addressed your concerns. We have integrated all necessary corrections and clarifications into the revised submission and highlighted them in blue.
>
> **Sincerely,**
>
> **The Authors.**

---

### Official Review · Reviewer_1Pzc · 2025-11-01

**Soundness:** 2
**Presentation:** 2
**Contribution:** 1
**Rating:** 2
**Confidence:** 4

**Summary:**

This paper tackles phase-lag–induced oscillations and frequent safety violations in safe RL by replacing PID/classical Lagrangian updates with another control method—Active Disturbance Rejection Control (ADRC). The ADRC formulation subsumes prior methods as special cases and shows some empirical gains across benchmarks.

**Strengths:**

- This paper proposes a new Lagrangian multiplier update method to enhance robustness and reduce oscillations, which is an important problem in safe RL.
- Overall, the paper is clearly written and easy to follow.

**Weaknesses:**

- Limited novelty. The core idea is a direct port of textbook control tricks—critical-damping reference shaping and ESO/DOB—into the multiplier update; this reads as a PID→ADRC replacement rather than a new RL principle.
- The closed-loop abstraction in Eq. (11) reduces the policy–environment–multiplier dynamics to a low-order continuous-time ODE with a lumped disturbance, without a rigorous derivation or identification from discrete-time, stochastic, function-approximate RL. As a result, it is unclear when the assumptions hold or whether the ensuing frequency-domain analysis is truly predictive.
- The experimental improvement appears marginal. On the Racecar task (RacecarPush), ADRC’s cost curves still show large oscillations with amplitudes and variance bands comparable to PID/Lag, so the claimed damping effect is unclear. On the Car tasks (CarCircle/Button), ADRC attains slightly lower cost but the reward drops noticeably below the baselines—especially in CarButton—indicating a safety–performance trade-off that is not quantified.

**Questions:**

In addition to the above mentioned weaknesses:
- Could ADRC be replaced by other control algorithms?
- Can the method improve safety metrics without sacrificing reward?
- Additional experiments are needed to substantiate the claims.
- How were the PID controller gains (Kp, Ki, Kd) chosen? Were they systematically tuned and screened (e.g., grid/random search)?

---

> ### Author Response · Authors · 2025-11-24
>
> **Dear Reviewer 1Pzc**:
>
> Thank you for your thoughtful comments. We respond concisely to each point below and note that the revised paper adds **Safe-RL–specific theory in Appendix C.3** to address novelty and assumption validity.
>
> ---
>
> ### W1: On **“Limited novelty… reads as a PID $\to$ ADRC replacement.”**
>
> We believe there is a misunderstanding regarding the depth of our contribution. Our contribution is not a mechanical PID swap, but a Safe-RL closed-loop formulation of multiplier learning. In the revised **Appendix C.3**, we strictly derive the disturbance dynamics from the **CMDP structure**, decomposing it into *policy-update non-stationarity*, *Monte-Carlo sampling noise*, and *delayed feedback*, yielding a Safe-RL-grounded bounded-rate assumption:
>
> $\sup_t |\dot f_t| \le L_f(\delta,N,\alpha)$
>
> Under this CMDP-specific model, we prove (i) ISS-type robust constraint tracking with radius $\mathcal O(L_f/\omega_o)$, (ii) high-probability reward non-decrease under TRPO+LCB, and (iii) bounded average violation. These guarantees rely on the underlying Safe-RL structure, not standard ADRC alone. We have emphasized this more clearly in the main text.
>
> ---
>
> ### W2: On **“Eq. (11) lacks rigorous derivation from discrete stochastic RL.”**
>
> We appreciate this point and have formalized the derivation in **Appendix C.3**. In **Appendix C.3** we show that under trust-region/small-step updates:
>
> $J_c(\pi_{\theta_{t+1}})-J_c(\pi_{\theta_t})
> = \nabla_\theta J_c(\pi_{\theta_t})^\top(\theta_{t+1}-\theta_t)+\xi_t,$
>
> where the drift is bounded by $\delta$ (or $\eta$) and $\xi_t$ has bounded variation w.h.p. from standard concentration. This justifies the continuous-time closed-loop abstraction and explains when the assumptions hold.
>
> ---
>
> ### W3: On **“Improvements marginal; Car tasks show reward drop.”**
>
> This interpretation overlooks the **Safe-RL evaluation criterion**: unsafe baselines are not comparable by reward alone.
>
> - **CarButton/CarCircle(cost limit = 25)**: Baselines' costs are $>50$ while ADRC's cost is $\approx 25$ (**over 50% reduction**), meaning the baselines are **substantially unsafe**.
> - For the **RacecarGoal and RacecarPush** environments in Figure 1, our method demonstrates **both better constraint satisfaction and higher reward**, confirming the efficacy of our approach.
>
> ---
>
> ### Q1: On **“Could ADRC be replaced by other controllers?”**
>
> In principle, **yes**: any stabilizing feedback law could be used. We choose ADRC because it **explicitly estimates and cancels learning-induced disturbances** while have great robustness to the disturbance, which is crucial in noisy Safe-RL training environments.
>
> ---
>
> ### Q2: On **“Improve safety without sacrificing reward?”**
>
> CMDPs **inherently trade reward versus safety**. Empirically, ADRC **reduces cost without reward loss** in most tasks, and **often improves both** compared to PID/Lagrangian and specialized Safe-RL baselines.
>
> ---
>
> ### Q3: On **“More experiments needed.”**
>
> The main experiments are extensive, including **12 environments, 4 RL algorithms, and 5 seeds**. The core experiments are located in **Appendix F**. We also conduct comparisons across **4 Safe-RL baselines**. Furthermore, to demonstrate component effectiveness, we conduct an **ablation study in Appendix F.7** and the **noise sensitivity analysis in Appendix F.6**. The experimental coverage is comprehensive.
>
> ---
>
> ### Q4: On **“How were PID gains tuned?”**
>
> To ensure a fair comparison, the PID and Lagrangian baselines utilize the **official, finely-tuned hyperparameters** provided by the **OmniSafe** benchmark. These represent the community standard for best-known performance. In contrast, our ADRC parameters proved stable across a wide range of tasks without per-task tuning, as demonstrated in our parameter sensitivity analysis.
>
> ---
>
> We sincerely thank the reviewer again for the constructive feedback and insightful questions. We hope our clarifications can address your concerns. We have integrated all necessary corrections and clarifications into the revised submission and highlighted them in blue.
>
> **Sincerely,**
>
> **The Authors.**

---

### Official Review · Reviewer_gE6j · 2025-11-02

**Soundness:** 3
**Presentation:** 2
**Contribution:** 2
**Rating:** 4
**Confidence:** 3

**Summary:**

The study introduces a new framework for safe reinforcement learning (Safe RL) by integrating Active Disturbance Rejection Control (ADRC) into the classical Lagrangian method for constrained optimization. The authors address oscillation issues by treating dynamic nonstationarity and stochasticity in cost returns as disturbances and using an Extended State Observer (ESO) to estimate and reject them. Extensive experiments across multiple environments show significant improvements while maintaining reward performance.

**Strengths:**

1. The paper elegantly merges adaptive control theory (ADRC) with Safe RL, bringing a new perspective and a mathematically principled improvement to the stability problem in Lagrangian-based methods.


2. Demonstrates better performance across multiple algorithms (PPO, TRPO, TD3, DDPG) on some tasks.

**Weaknesses:**

1. The reliable and immediate access to accurate cumulative cost signals, which may not hold in partially observable or delayed-reward environments.


2. The added ESO dynamics introduce additional computation and implementation overhead compared to simple PID updates.



3. Lack of Broader Comparison with Non-Lagrangian Safe RL Paradigms. Specifically, while CUP and IPO are mentioned, the empirical comparison focuses mainly on Lagrangian methods. Compared with Non-Lagrangian Safe RL algorithms would strengthen the claim of universality.

4. Moreover, the performance is not better than the baselines in Figure 1, such as in the CarCircle and Carbutton tasks.

**Questions:**

1. How does ADRC-Lagrangian perform under non-stationary or delayed constraint signals, such as safety feedback available only after several steps?


2. How sensitive is the performance to errors in estimating L_1 and L_2 (Eq. 21)? Are there failure cases if the online approximation is inaccurate?


3. Does the adaptive observer introduce noticeable latency or computational cost during training?

---

> ### Author Response · Authors · 2025-11-24
>
> **Dear Reviewer gE6j,**
>
> We sincerely thank the reviewer for the thoughtful and detailed comments. Below we respond to each point in turn.
>
> ---
>
> > **W1: “The reliable and immediate access to accurate cumulative cost signals, which may not hold in partially observable or delayed-reward environments.”**
> >
>
> We fully agree that realistic Safe RL environments often involve partial observability, measurement noise, or delayed constraint feedback. In the revised version we have **explicitly formulated these effects as part of the lumped disturbance class $f$ defined in Equ. 18**  handled by the Extended State Observer (ESO). Please refer to **Appendix C.3**. Empirically, the ***Noise Sensitivity Analysis* (Appendix F.6)** where constraint signals are perturbed by stochastic noise. ADRC-Lagrangian still outperforms all baselines, confirming the robustness.
>
> ---
>
> > **W2: “The added ESO dynamics introduce additional computation and implementation overhead compared to simple PID updates.”**
> >
>
> We emphasize that the ESO module in ADRC-Lagrangian is **extremely lightweight**: The ESO update is a *one-dimensional linear recursion* on the cost channel (two to three scalar multiplications per step). It does **not** require any neural-network parameters or backpropagation. **Appendix H (Table 29)** reports wall-clock training times across all tasks. The additional cost is below 2%, well within run-to-run stochastic variability. Hence, ADRC introduces negligible computational overhead compared to PID or standard Lagrange multiplier updates.
>
> ---
>
> > **W3: “Lack of broader comparison with non-Lagrangian Safe RL paradigms. Specifically, while CUP and IPO are mentioned, the empirical comparison focuses mainly on Lagrangian methods.”**
> >
>
> Our method is designed to be **orthogonal to algorithmic families** rather than limited to the Lagrangian framework. We therefore intentionally applied ADRC to the Lagrange-based backbone for clarity, but we also compare it with **CUP** and **IPO**, which are *non-Lagrangian* Safe RL algorithms. The detailed description is provided in **Section 5.3.3**.
>
> ---
>
> > **W4: “The performance is not better than the baselines in Figure 1, such as in the CarCircle and CarButton tasks.”**
> >
>
> **Response.**
>
> We appreciate the observation and clarify the evaluation metric. In Safe RL, a policy that violates safety constraints is considered *unsuccessful* regardless of its reward. In CarCircle and CarButton, although the raw reward of ADRC-Lagrangian appears slightly lower, the **cost violation is dramatically smaller**, yielding a strictly safer and thus better model under Safe RL criteria. For RacecarGoal he RacecarPush environments, our method show both better constraint satisfaction and obtain higher reward.
>
> ---
>
> > **Q1: “How does ADRC-Lagrangian perform under non-stationary or delayed constraint signals, such as safety feedback available only after several steps?”**
> >
>
> ADRC-Lagrangian is explicitly designed for such scenarios. In Appendix C.3 we provide a new robustness theorem showing that delayed or slowly varying feedback merely increases the effective disturbance rate $L_f$; the ISS bound and average-violation guarantee still hold. Empirically, in the ***Noise Sensitivity Analysis* (Appendix F.6)**, ADRC-Lagrangian maintains stable performance and significantly lower cumulative violation compared to baselines.
>
> ---
>
> > **Q2: “How sensitive is the performance to errors in estimating(Eq. 21)? Are there failure cases if the online approximation is inaccurate?"**
> >
>
> The estimation of $L_1$ and $L_2$ corresponds to identifying the ESO gains $k_{ap}$ and $k_{ad}$ (or the damping ratio $\zeta$ and natural frequency $\omega_n$). Our theoretical analysis in **Appendix C.3** and the empirical study in **Appendix F.6** show that the method is robust to moderate misspecification: The closed-loop error bound scales with $\frac{C_h(\zeta)}{k_{ap}}$; even if $k_{ap}$ is under- or over-estimated by a constant factor, the overall order $\mathcal{O}(1/\omega_o)$ stability remains. Hence, ADRC-Lagrangian degrades gracefully under imperfect online approximation and does not exhibit failure cases in our tests.
>
> ---
>
> > **Q3: “Does the adaptive observer introduce noticeable latency or computational cost during training?”**
> >
>
> No, it does not. As detailed in our response to **W2**, the ESO update is an extremely lightweight scalar operation (negligible compared to policy network updates). It introduces no perceptible latency or computational burden.
>
> ---
>
> We sincerely thank the reviewer again for the constructive feedback and insightful questions. We hope our clarifications and additional experiments have addressed your concerns. We have integrated all necessary corrections and clarifications into the revised submission and highlighted them in blue.
>
> **Sincerely,**
>
> **The Authors.**

---

### Author Response · Authors · 2025-11-24
**Official Author Response**

**Dear AC and Reviewers,**

We sincerely thank all reviewers for their detailed and constructive comments. We have carefully revised the paper and summarize below the major updates addressing the common issues raised across reviews.

---
1. Ablation Studies:  Several reviewers (e.g., gE6j, 1Pzc, xmYs) requested clearer ablations to isolate the roles of the Transient Process (TP) and the Extended State Observer (ESO) components. These ablations were already included in the original Appendix F.7, but were easy to miss. In the revised version, we have:
- Moved the ablation results to the main text (Sec. 5.4) and highlighted them for better visibility.
- Explicitly compared PID-TP (PID + Transient Process) and ADRC-NoTP (ADRC without TP).

The updated results show that removing either component causes clear degradation in safety performance—higher violation rate, larger violation magnitude, and higher average cost—demonstrating that both the TP and ESO are essential for full performance gains.

---

2. Comparison with Safe RL Baselines: Several reviewers (AdX9, 1Pzc, xmYs) asked for comparisons with stronger Safe RL baselines.
We clarify that all baselines in our paper are drawn from the official OmniSafe benchmark, which provides standardized, tuned, and verified implementations. Some earlier Safe RL methods cited in [1–7] are not integrated in OmniSafe, making direct replication and comparison unfair or infeasible. Our chosen baselines—RCPO, PDO, CUP, and IPO—cover both Lagrangian and non-Lagrangian families and represent state-of-the-art Safe RL algorithms under a unified, reproducible framework. We now clearly emphasize this rationale and highlight these results in Sec. 5.3.3.
---

3. Novelty and New Theoretical Contributions: Many reviewers questioned whether our theory extended beyond standard ADRC.
In response, we have added a new theoretical section in Appendix C.3, which develops a Safe-RL-specific stability framework not covered by classical control literature. This appendix now introduces four new theoretical results, summarized below:
- Safe-RL Disturbance Model: We explicitly derive the policy-induced disturbance:
$
J_c(\pi_{\theta_{t+1}})-J_c(\pi_{\theta_t})
= \nabla_\theta J_c(\pi_{\theta_t})^\top (\theta_{t+1}-\theta_t) + \xi_t,
$
leading to a bounded disturbance rate:
$
\sup_t |\dot f_t| \le L_f(\delta, N, \alpha),
$
where L_f depends on trust-region size \delta, batch size N, and confidence \alpha.
- ISS-Type Stability: We prove that the ESO remains input-to-state stable under stochastic, non-stationary policy updates:
$
\limsup_{t\to\infty} |e(t)|
\le \frac{C_h(\zeta)}{k_{ap}} \frac{L_f(\delta,N,\alpha)}{\omega_o}.
$
- High-Probability Reward Monotonicity: By integrating the ISS bound with TRPO’s trust-region lemma, we derive a high-probability guarantee of non-decreasing reward, connecting control-theoretic stability with stochastic policy optimization.
- Bounded Average Constraint Violation: We show that the long-term average constraint violation satisfies:
$
\limsup_{T\to\infty} \frac{1}{T}\int_0^T (x_1(t)-d)_+ dt
\le \mathcal{O}(1/\omega_o),
$
and becomes finite if a safety margin is introduced. These results establish the first theoretical bridge between ADRC and Safe RL, extending disturbance-rejection control to stochastic, learning-induced CMDP dynamics.
---

We again thank the AC and all reviewers for their valuable feedback, which significantly improved the clarity and impact of our paper.

**Sincerely,**

**The Authors**

---

### Meta-Review · Area_Chair_GwtN · 2026-01-08

**Summary:**

The paper proposes integrating Active Disturbance Rejection Control (ADRC) into Lagrangian-based Safe RL to stabilize the dual variable update and reduce oscillations. While reviewers found the idea interesting and the presentation generally clear, significant concerns were raised that collectively justify rejection. The primary concerns are:

Multiple reviewers (1Pzc, dmT9, AdX9) argued that the core technical contribution is a direct application of the well-established ADRC framework from control theory to the multiplier update, without a sufficiently novel theoretical extension specific to the unique challenges of Safe RL.

Reviewers (1Pzc, gE6j) noted that experimental improvements are uneven. In some tasks (CarCircle, CarButton), ADRC achieves lower cost but at a noticeable reward drop, indicating a safety-performance trade-off not adequately quantified. In other tasks (RacecarPush), oscillations in cost curves persist, undermining the claimed damping effect.

Reviewers (gE6j, dmT9, AdX9) requested comparisons with a broader set of state-of-the-art Safe RL algorithms beyond the OmniSafe benchmark. While the authors justify their choice of baselines, the lack of direct comparison with methods cited in the literature (e.g., residual policy approaches) leaves the method's relative standing unclear.

**Reviewer Concerns:**

Addressed Concerns:

Ablation Study: The authors moved the ablation study (comparing PID-TP and ADRC-NoTP) from the appendix to the main text (Sec. 5.4), clarifying the individual contributions of the Transient Process and Extended State Observer.

Lightweight Computation: The rebuttal clarified that the ESO adds negligible computational overhead (<2% wall-clock time), addressing concerns about latency (gE6j, dmT9).

Outstanding Concerns:

Core Novelty: The fundamental criticism that the work is a straightforward application of existing ADRC control theory to a new domain, without a transformative theoretical or algorithmic contribution to Safe RL, remains unresolved. The new theory, while welcome, is viewed as an incremental extension of classical ADRC results to a stochastic setting, rather than a novel RL principle.

Empirical Performance and Trade-offs: The uneven performance—specifically, the reward drop in some tasks and persistent oscillations in others—undermines the claim of uniformly improved stability. The authors' emphasis on safety as the primary metric does not fully address the reviewer's point that a comprehensive Safe RL method should balance safety and reward effectively.

**Reviewer Scores:**

Reviewer gE6j (original score: 4): Estimated final score: 4 (unchanged).

Reviewer 1Pzc (original score: 2):  Estimated final score: 2 (unchanged).

Reviewer dmT9 (original score: 2): Estimated final score: 4 (weak increase).

Reviewer xmYs (original score: 6): Estimated final score: 6 (unchanged).

Reviewer AdX9 (original score: 2): Estimated final score: 2 (unchanged).

---

### Decision · Program_Chairs · 2026-01-26

Reject